

# Regional evapotranspiration from image-based implementation of the Surface Temperature Initiated Closure (STIC1.2) model and its validation across an aridity gradient in the conterminous United States

Nishan Bhattarai[1], Kaniska Mallick[2], Nathaniel A. Brunsell[3], Ge Sun[4], Meha Jain[1]

[1] School for Environment and Sustainability, University of Michigan, Ann Arbor, MI 48109, USA
[2] Remote Sensing and Ecohydrological Modeling, Dept. ERIN, Luxembourg Institute of Science and Technology (LIST), L4422, Belvaux, Luxembourg
[3] Geography and Atmospheric Science, University of Kansas, Lawrence, KS 66045, USA
[4] Eastern Forest Environmental Threat Assessment Center, Southern Research Station, U.S. Department of Agriculture Forest Service, Raleigh, 27606, NC, USA

*Correspondence to*: Nishan Bhattarai (nbhattar@umich.edu)

**Abstract.** Recent studies have highlighted the need for improved characterizations of aerodynamic conductance and temperature ($g_A$ and $T_0$) in thermal remote sensing-based surface energy balance (SEB) models to reduce uncertainties in

regional-scale evapotranspiration (ET) mapping. By integrating radiometric surface temperature ($T_R$) into the Penman-Monteith (PM) equation and finding analytical solutions of $g_A$ and $T_0$, this need was recently addressed by the Surface Temperature Initiated Closure (STIC) model. However, previous implementations of STIC were confined to the ecosystem-scale using flux tower observations of infrared temperature. This study demonstrates the first regional-scale implementation of the most recent version of the STIC model (STIC1.2) that physically integrates Moderate Resolution Imaging

Spectroradiometer (MODIS)-derived $T_R$ and ancillary land surface variables in conjunction with NLDAS (North American Land Data Assimilation System) atmospheric variables into a combined structure of the PM and Shuttleworth-Wallace framework for estimating ET at 1 km × 1 km spatial resolution. Evaluation of STIC1.2 at thirteen core AmeriFlux sites covering a broad spectrum of climates and biomes across an aridity gradient in the conterminous US suggests that STIC1.2 can provide spatially explicit ET maps with reliable accuracies from dry to wet extremes. When observed ET from one wet, one dry, and

one normal precipitation year from all sites were combined, STIC1.2 explained 66 % of the variability in observed 8-day cumulative ET with a root mean square error (RMSE) of 7.4 mm/8-day, mean absolute error (MAE) of 5 mm/8-day, and percent bias (PBIAS) of -4 %. These error statistics show higher accuracies than a widely-used SEB-based Surface Energy

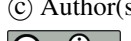



Balance System (SEBS) and PM-based MOD16 ET, which were found to overestimate (PBIAS = 28 %) and underestimate
ET (PBIAS = -26 %), respectively. The performance of STIC1.2 was better in forest and grassland ecosystems as compared
to cropland (20 % underestimation) and woody savanna (40 % overestimation). Model inter-comparison suggested that ET
differences between the models are robustly correlated with $g_A$ and associated roughness length estimation uncertainties which

are intrinsically connected to $T_R$ uncertainties, vapour pressure deficit ($D_A$), and vegetation cover. A consistent performance
of STIC1.2 in a broad range of hydrological and biome categories as well as the capacity to capture spatio-temporal ET
signatures across an aridity gradient points to its potential for near real time ET mapping from regional to continental scales.

## 1 Introduction

Evapotranspiration (ET) is highly variable in space and time and plays a fundamental role in hydrology and land-atmosphere

interactions. Over the past few decades, the use of satellite data to map spatially-explicit and regional-scale ET has advanced
considerably. This is due to the advancements in ET modeling as well as progress in thermal remote sensing satellite missions,
and our ability to retrieve the land surface temperature (LST) or radiometric surface temperature ($T_R$) that is highly sensitive
to evaporative cooling and surface moisture variations. Because LST governs the land surface energy budget (Kustas and
Norman, 1996;Kustas and Anderson, 2009), thermal ET models principally focus on the surface energy balance (SEB)

approach in which $T_R$ represents the lower boundary condition to constrain energy-water fluxes (Anderson et al., 2012).
Contemporary SEB models emphasize on estimating aerodynamic conductance ($g_A$) and sensible heat flux ($H$) while solving
ET (i.e., latent heat flux, $\lambda E$) as a residual SEB component. Despite the advancements in mapping spatially-distributed ET,
some fundamental challenges remain in existing SEB algorithms including, (a) the inequality between $T_R$ and the aerodynamic
temperature ($T_0$) which is essentially responsible for the exchanges of $H$ and $\lambda E$ (Chávez et al., 2010;Boulet et al., 2012), (b)

a non-unique relationship between $T_0$ and $T_R$ due to differences between the roughness lengths (i.e., effective source/sink
heights) for momentum ($z_{0M}$) and heat ($z_{0H}$) within vegetation canopy and substrate complex (Troufleau et al., 1997;Paul et al.,
2014;van Dijk et al., 2015b), (c) the unavailability of a universally agreed model to estimate $T_0$ (Colaizzi et al., 2004), and (d)
the lack of a physically-based preeminent $g_A$ model. To overcome these challenges, we implement the current version of a
recently developed analytical ET model, Surface Temperature Initiated Closure (STIC, version 1.2 (Mallick et al.,

2014;Mallick et al., 2015;Mallick et al., 2016), using Moderate Resolution Imaging Spectro-Radiometer (MODIS) data to
develop spatially-distributed ET maps.

The STIC formulation provides analytical solutions to $g_A$, $T_0$, and canopy (or surface) conductance ($g_C$), and simultaneously
captures the critical feedbacks between $g_A$, $g_C$, $T_0$, and vapour pressure deficit surrounding the evaporating surface ($D_0$) thereby
obtaining a 'closure' of the surface energy balance. In state-of-the-art SEB models, an emphasis on estimating $g_A$ and $H$ is

motivated due to the perception of the broad applicability of the Monin-Obukhov Similarity Theory (MOST) or Richardson



Number (Ri) criteria, and the requirement of minimum inputs for determining these variables. However, these approaches created further problems, particularly in accommodating $T_0$ versus $T_R$ inequalities, as well as adapting the differences between $z_{0M}$ and $z_{0H}$ (Paul et al., 2014). Compensating these temperature and roughness length disparities consequently led to the inception of the $kB^{-1}$ term as a fitting parameter (Verhoef et al., 1997a), and later the progress of the two-source ET model

(Kustas and Norman, 1997;Norman et al., 1995;Anderson et al., 2011). Although useful, the above approaches still rely on empirical response functions of roughness components to characterize $g_A$ that has an uncertain transferability in space and time (Holwerda et al., 2012b;van Dijk et al., 2015b). In contemporary SEB modeling, $g_A$ sub-models are stand-alone and lack the necessary physical feedbacks between the conductances, $T_0$, and $D_0$ to 'close' the surface energy balance. The feedback of $g_A$ on $g_C$ and $D_0$ is critical in semiarid and arid ecosystems (Kustas et al., 2016), where soil moisture stress and sparse vegetation

can cause substantial disparities between $T_R$ versus $T_0$ (Kustas et al., 2016;Paul et al., 2014;Timmermans et al., 2013;Gokmen et al., 2012). Therefore, thermal-based ET modeling needs explicit consideration of these important biophysical feedbacks to overcome the existing uncertainties in regional-scale ET mapping (Kustas et al., 2016). Hence, a genuine question in regional ET mapping is: *How can state-of-the-art SEB models overcome the existing challenges in regional evapotranspiration mapping that arise due to uncertain conductance parameterizations, and can analytical models help this verification process?*

The prime focus of STIC (Mallick et al., 2014;Mallick et al., 2015;Mallick et al., 2016) is based on physical integration of $T_R$ into the Penman-Monteith (PM) equation, which is fundamentally constrained to account for the necessary feedbacks between ET, $T_R$, $D_A$, $g_A$, and $g_C$ (Monteith, 1965). Monteith (1981) highlighted the fact that the biophysical conductances (i.e., $g_A$ and $g_C$) regulating ET are heavily temperature dependent, after which a stream of research demonstrated the dominant control of $T_R$ into $g_C$ and associated canopy-scale aerodynamics (Moffett and Gorelick, 2012;Blonquist et al., 2009). Somewhat

surprisingly, the idea of integrating $T_R$ into the PM model was never attempted because of complexities associated with $g_C$ parameterization (Bell et al., 2015;Matheny et al., 2014), until the concept of STIC was formulated (Mallick et al., 2014;Mallick et al., 2015). The recent version of STIC, STIC1.2, combines PM with the Shuttleworth-Wallace (SW) model (Shuttleworth and Wallace, 1985) to estimate the source/sink height temperature and vapour pressure ($T_0$ and $e_0$) (Mallick et al., 2016). By algebraic reorganization of aerodynamic equations of $H$ and $\lambda E$, Bowen ratio evaporative fraction hypothesis

(Bowen, 1926) and modified advection-aridity hypothesis (Brutsaert and Stricker, 1979), STIC1.2 formulates multiple state equations where the state equations were constrained with an aggregated moisture availability factor ($M$). Through physically linking $M$ with $T_R$ and the source/sink height dew point temperature ($T_{SD}$), STIC1.2 established a direct feedback between $T_R$ and ET, while simultaneously overcoming the empirical uncertainties in conductances and $T_0$ estimations.

Despite providing analytical solutions for the key conductances in PM-based ET modeling, the STIC 1.2 model has yet to

gain a profound interest among the thermal remote sensing community and those interested in regional-scale ET modeling. This could largely be attributed to the fact that the model is only used for understanding ecosystem-scale ET partitioning and their biophysical controls at the eddy covariance (EC) footprints (Mallick et al., 2015;Mallick et al., 2016), where all the necessary forcing variables were measured at the flux tower sites. In this paper, we present the first ever implementation of the STIC1.2 model using MODIS LST and associated land surface products, and its validation in thirteen core AmeriFlux sites





across an aridity gradient in the conterminous US in three different precipitation conditions representing dry, normal, and wet years, respectively. ET estimates from STIC1.2 are also compared against two parametric ET models, namely SEBS (Surface Energy Balance System) (Su, 2002) and MOD16 (Mu et al., 2007;Mu et al., 2011). Through the implementation and validation of the STIC1.2 model at a regional-scale, the current study addresses the following research questions:

1. What is the performance of STIC1.2 when applied at the regional-scale across an aridity gradient and during contrasting rainfall years in the conterminous US?

2. How does STIC1.2-derived ET compare against other global ET models that are driven by $T_R$ and relative humidity (RH)?

3. Under which conditions do the models agree and which factors cause their differences?

4. How well do the models capture spatio-temporal ET variability across an aridity gradient?

A description of methods including models, study sites, dataset, and data processing is given in section 2, followed by the results in section 3. An extended discussion of the results and potential of the method in thermal remote sensing applications is elaborated in sections 4 and 5, respectively. Symbols used for variables in this study are listed in the Appendix in Table A1.

## 2 Methods

### 2.1 Model Descriptions

Most surface energy balance models consist of several modules for estimating net radiation ($R_N$), ground heat flux ($G$), and partitioning of available energy ($\phi = R_N - G$) into $H$ and $\lambda E$ through the derivation of evaporative fraction ($\Lambda$). $\Lambda$ is defined as the ratio of $\lambda E$ to $\phi$. In this paper, we used the widely-used net radiation balance equation (Eq. (1)) to compute $R_N$ (Allen et al., 2007;Allen et al., 2011) and the formulation of Bastiaanssen (2000) to compute $G$ (Eq. 4) in SEBS and STIC1.2.

$$R_N = R_S (1 - \alpha_o) + \varepsilon_o R_{ld} - R_{lu} \tag{1}$$

$$G = R_N (T_R - 273.15)/\alpha_o (0.0038\alpha_o + 0.0074\alpha_o^2)(1 - 0.98NDVI^4) \tag{2}$$

$$H = (1 - \Lambda) \times (R_N - G) \tag{3}$$

$$\lambda E = \Lambda \times (R_N - G) \tag{4}$$

where $R_S$ is the incoming shortwave radiation, $\alpha_o$ is the surface albedo, $\varepsilon_o$ is the surface emissivity, NDVI is the normalized difference vegetation index, $\lambda$ is the latent heat of vaporization, and $R_{ld}$ and $R_{lu}$ are incoming and outgoing longwave radiation, respectively. Using the formulation of Allen et al. (2007) and Bastiaanssen (2000) for estimating $R_N$ and $G$, respectively, we found that the estimated eight-day mean $R_N$ and $G$ during the terra overpass time were within 14 % of the observed $R_N$ and $G$ at the flux sites (Fig. S1).

While the derivation of $H$ in SEBS is based on aerodynamic equation (Su, 2002), SEBS estimates $\lambda E$ as the residual of the surface energy balance (i.e., $\lambda E = R_N - G - H$). On the contrary, STIC1.2 directly estimates $H$ and $\lambda E$ through the PM equation (Mallick et al., 2016) by solving state equations for the conductances. MOD16 estimates $\lambda E$ directly using a modified PM framework (Mu et al., 2007;Mu et al., 2011), where the conductances are estimated based on a biome property look up table





(BPLUT) and meteorological scaling functions. As discussed in section 1, there exist some fundamental differences among STIC1.2, SEBS, and MOD16. However, since the primary focus of the paper is the regional-scale implementation and evaluation of the STIC1.2 model, we only provide detailed descriptions of STIC1.2 and suggest readers follow associated literature for detailed descriptions of the other two models (see subsections 2.1.2 and 2.1.3). The key model structures of SEBS
and MOD16 are briefly explained in subsections 2.1.2 and 2.1.3.

### 2.1.1 STIC1.2

STIC1.2 is the most recent version of the original STIC formulation (Mallick et al., 2014;Mallick et al., 2015), which is a one-dimensional physically-based SEB model that treats the vegetation-substrate complex as a single unit (Fig. 1). The fundamental assumption in STIC1.2 is the first order dependency of $g_A$ and $g_C$ on $T_0$ and soil moisture through $T_R$. Such an assumption
allows a direct integration of $T_R$ in the PM equation (Mallick et al., 2016). The common expression for $\lambda E$ in the PM equation is,

$$\lambda E = \frac{s\phi + \rho_A c_P g_A D_A}{s + \gamma\left(1 + \frac{g_A}{g_C}\right)} \qquad (5)$$

where $\rho_A$ is the air density (kg m$^{-3}$), $c_P$ is the specific heat of air (J kg$^{-1}$ K$^{-1}$), $\gamma$ is the psychrometric constant (hPa K$^{-1}$), $s$ is the slope of the saturation vapour pressure versus $T_A$ (hPa K$^{-1}$), $D_A$ is the saturation deficit of the air (hPa) at the reference level, and $\phi$ is the net available energy (i.e., $R_N$ - $G$). The units for all the surface fluxes and conductances are W m$^{-2}$ and m s$^{-1}$,
respectively.

In Eq. (5), the two biophysical conductances ($g_A$ and $g_C$) are unknown and the STIC1.2 methodology is based on finding analytical solutions for the two unknown conductances to directly estimate ET (Mallick et al., 2014;Mallick et al., 2015). The need for such analytical estimation of these conductances is motivated by the fact that $g_A$ and $g_C$ can neither be measured at the canopy or larger spatial scales, and there is not an appropriate model of $g_A$ and $g_C$ that currently exists (Matheny et al.,
2014;van Dijk et al., 2015b). By integrating $T_R$ with standard SEB theory and vegetation biophysical principles, STIC1.2 formulates multiple state equations (Eqs. (7)-(10) below) in order to eliminate the need for empirical parameterization for $g_A$, $g_C$, and $T_0$. The state equations for the conductances and $T_0$ were expressed as a function of those variables that can be estimated by remote sensing observations. In the state equations, a direct connection of $T_R$ is established by estimating an aggregated moisture availability index ($M$). The information of $M$ is subsequently used in the state equations of $g_A$, $g_C$, $T_0$, and evaporative
fraction ($\Lambda$) (Eqs. (7)-(10) below), which is eventually propagated into their analytical solutions. $M$ is a unitless quantity, which describes the relative wetness of the surface and also controls the transition from potential to actual evaporation. Therefore, $M$ is critical for providing a constraint against which the conductances can be estimated. Since $T_R$ is extremely sensitive to the surface water content variations, it is extensively used for estimating $M$ in a physical retrieval scheme (detail in Appendix A3) (also in Mallick et al., 2016). We hypothesize that linking $M$ with the biophysical conductances will simultaneously integrate
the information of $T_R$ into the PM equation (Eq. (5)) in the framework of STIC1.2.


In STIC1.2, the estimation of $M$ is based on Venturini et al. (2008), where $M$ is expressed as the ratio of the vapour pressure difference between the source/sink height and air to the vapour pressure deficit between source/sink height to the atmosphere as follows.

$$M = \frac{(e_0 - e_A)}{(e_0^* - e_A)} = \frac{(e_0 - e_A)}{\kappa(e_S^* - e_A)} = \frac{s_1(T_{SD} - T_D)}{\kappa s_2(T_R - T_D)} \tag{6}$$

Where $e_0$ and $e_0^*$ are the actual and saturation vapour pressure at the source/sink height; $e_A$ is the atmospheric vapour pressure; $e_S^*$ is the saturation vapour pressure at the surface; $T_D$ is the air dewpoint temperature; $s_1$ and $s_2$ are the psychrometric slopes of the saturation vapour pressure and temperature between $(T_{SD} - T_D)$ versus $(e_0 - e_A)$ and $(T_R - T_D)$ versus $(e_S^* - e_A)$ relationship (Venturini et al., 2008); and $\kappa$ is the ratio between $(e_0^* - e_A)$ and $(e_S^* - e_A)$. Despite $T_0$ driving the sensible heat flux, the comprehensive dry-wet signature of the underlying surface due to aggregated moisture variability is directly reflected in $T_R$ (Kustas and Anderson, 2009). Therefore, using $T_R$ in the denominator of Eq. (6) tends to give a direct signature of the surface moisture availability. In Eq. (6), both $s_1$ and $T_{SD}$ are unknowns, and an initial estimate of $T_{SD}$ is obtained using Eq. (6) of Venturini et al. (2008) where $s_1$ was approximated in $T_D$. From the initial estimates of $T_{SD}$, an initial estimate of $M$ is obtained as $M = s_1 (T_{SD} - T_D)/s_2 (T_R - T_D)$. However, since $T_{SD}$ also depends on $\lambda E$, an iterative updating of $T_{SD}$ (and $M$) is carried out by expressing $T_{SD}$ as a function of $\lambda E$ which is described in detail in Appendix A3 (also in Mallick et al., 2016).

The state equations of STIC1.2 are provided below and their detailed descriptions are available in Mallick et al. (2014; 2015; 2016).

$$g_A = \frac{\phi}{\rho_A c_P \left[ (T_0 - T_A) + \left( \frac{e_0 - e_A}{\gamma} \right) \right]} \tag{7}$$

$$g_C = g_A \frac{(e_0 - e_A)}{(e_0^* - e_0)} \tag{8}$$

$$T_0 = T_A + \left( \frac{e_0 - e_A}{\gamma} \right) \left( \frac{1 - \Lambda}{\Lambda} \right) \tag{9}$$

$$\Lambda = \frac{2\alpha s}{2s + 2\gamma + \gamma \frac{g_A}{g_C}(1 + M)} \tag{10}$$

Here $\alpha$ is the Priestley-Taylor coefficient (unitless) (Priestley and Taylor, 1972). In Eq. (10), $\alpha$ appeared due to using the Advection-Aridity (AA) hypothesis (Brutsaert and Stricker, 1979) for deriving the state equation of $\Lambda$ (Mallick et al., 2016;Mallick et al., 2015). However, instead of optimising it as a 'fixed parameter', $\alpha$ is dynamically estimated by constraining it as a function of $M$, conductances, source/sink height vapour pressure, and temperature (Mallick et al., 2016). The derivation of the equation for $\alpha$ is described in Appendix A3.

Given values of $M$, $R_N$, $G$, $T_A$, and RH or $e_A$, the four state equations (Eqs. (7)-(10)) can be solved simultaneously to derive analytical solutions for the four unobserved state variables and to simultaneously produce a 'closure' of the PM model that is independent of empirical parameterizations for both $g_A$ and $g_C$ (Appendix A2). However, the analytical solutions to the four state equations contain three accompanying unknowns; $e_0$, $e_0^*$, and $\alpha$, and as a result there are four equations with seven





unknowns. Consequently, an iterative solution must be found to determine the three unknown variables (Appendix A3) (also in Mallick et al., 2016).

In STIC1.2, the key modifications to the original STIC formulation (Mallick et al., 2014) include estimation of the source/sink height vapour pressures by combining PM and Eq. (8) of Shuttleworth-Wallace (Shuttleworth and Wallace, 1985), as detailed in Appendix A3 (also in Mallick et al., 2016). STIC1.2 consists of a feedback loop describing the relationship between $T_R$ and $\lambda E$, coupled with canopy-atmosphere components relating $\lambda E$ to $T_0$ and $e_0$ (Mallick et al., 2016). Upon finding analytical solution of $g_A$ and $g_C$, both the variables are returned into Eq. (5) to directly estimate $\lambda E$. For the image-based implementation of STIC1.2, we make a key adjustment to the original ecosystem-scale STIC1.2 version (Mallick et al., 2016) to apply the model at an instantaneous scale (i.e. MODIS image acquisition time) by removing the calculation of hysteresis occurrence using hourly data (Mallick et al., 2015). Such adjustment was necessary to adapt the model to single time-of-day $T_R$ data from MODIS acquisition.

### 2.1.2 SEBS

SEBS formulation uses an empirical model for estimating $z_{0M}$, the Bulk Atmospheric Similarity Theory for planetary boundary layer scaling, and the Monin-Obukhov atmospheric surface layer similarity for surface layer scaling for the estimation of surface fluxes from thermal remote sensing data (Su, 2002;Su et al., 2001). To estimate $H$, SEBS solves the similarity relationships for the profile wind speed ($u$) and the mean difference between potential temperatures ($\Delta\theta$; K) at the surface and reference height ($z$)):

$$u = \frac{u_*}{k}\left[\ln\left(\frac{z-d_0}{z_{0M}}\right) - \psi_M\left(\frac{z-d_0}{L}\right) + \psi_M\left(\frac{z_{0M}}{L}\right)\right] \quad (11)$$

$$\Delta\theta = \frac{H}{ku_*\rho c_p}\left[\ln\left(\frac{z-d_0}{z_{0H}}\right) - \psi_H\left(\frac{z-d_0}{L}\right) + \psi_H\left(\frac{z_{0H}}{L}\right)\right] \quad (12)$$

$$L = -\frac{\rho_A c_p u_*^3 \theta_v}{kgH} \quad (13)$$

Here $L$ is the Monin-Obukhov length (m), $\theta_v$ is virtual potential temperature (K) near the surface (Brutsaert, 2005), k is the Von Karman Constant (0.41), $u_*$ is the friction velocity (m s$^{-1}$), and $g$ is the acceleration due to gravity (9.8 m s$^{-2}$). $\Psi_M$ and $\Psi_H$ are the stability corrections for momentum and heat transport, respectively.

One of the key characteristics of the SEBS model is the use of a semi-physical adjustment factor ($kB^{-1}$) to compensate for the differences between $z_{0M}$ and $z_{0H}$ (Su et al., 2001):

$$z_{0H} = z_{0M}/\exp(kB^{-1}) \quad (14)$$

The pixel-level energy balance at a dry limit ($\lambda E = 0$ or $H = \phi$) and a wet limit (potential ET, $E_p$, rate based on Penman equation) is used in SEBS to estimate relative evaporation ($\Lambda_R$, the ratio of actual to the maximum evaporation rates) to further compute $\Lambda$ (Su, 2002).



$$\Lambda_\mathrm{r} = 1 - \frac{H - H_\mathrm{wet}}{H_\mathrm{dry} - H_\mathrm{wet}} \qquad (15)$$

$$\Lambda = \frac{\Lambda_\mathrm{R} \times \lambda E_\mathrm{wet}}{R_\mathrm{N} - G} \qquad (16)$$

where $H_\mathrm{wet}$ and $H_\mathrm{dry}$ are $H$ under the wet and dry limiting conditions, respectively. $\lambda E_\mathrm{wet}$ is the $\lambda E$ at the wet limit.

### 2.1.3 MOD16 algorithm

The MOD16 algorithm is based on the PM equation (Eq. (5)) and is designed to estimate ET by summing wet soil evaporation, interception evaporation from the wet canopy, and transpiration through canopy over vegetated land surfaces. The original PM equation was modified by Mu et al. (2007, 2011) for estimating global ET components and is primarily driven by MODIS-derived vegetation variables (leaf area index, fractional vegetation cover) and daily meteorological inputs including $R_\mathrm{S}$, $T_\mathrm{A}$, and $D_\mathrm{A}$.

Key inputs in the MOD16 ET product include the global 1 km × 1 km MODIS collections, including annual land cover (MOD12Q1), 8-day LAI/FPAR (MOD15A2), 8-day albedo (MCD43B2 and MCD43B3 products), and the global GMAO daily meteorological reanalysis data (1.00° × 1.25° resolution). The MODIS 8-day albedo products and daily surface downwelling shortwave radiation and air temperature from daily meteorological reanalysis data are used to calculate $R_\mathrm{N}$. The vegetation cover fraction from the MODIS 8-day FPAR products is used to allocate the $R_\mathrm{N}$ between soil and vegetation. Daily $T_\mathrm{A}$, $D_\mathrm{A}$ and RH, and 8-day MODIS LAI information are used to estimate individual resistances from soil and canopy, and soil heat flux, respectively. A Biome-Property-Lookup-Table (BPLUT) is used to assign minimum and maximum resistances for all land cover categories, and the biome-specific resistances are constrained through different environmental scalars. Readers are referred to Mu et al. (2011) for a detailed description of the derivation of key ET components and the parameters used in the MOD16 algorithm for estimating ET.

### 2.2 Study sites

For validating the STIC1.2 model, we selected thirteen core AmeriFlux sites covering a broad spectrum of biomes which also represent a wide range of climatic, elevation (5 to 3050 m), precipitation (P; 380 to 1320 mm year$^{-1}$), temperature (1.50 to 17.92 °C), and aridity gradients across the conterminous United States (Fig. 2; Table 1). AmeriFlux is a subnetwork of FLUXNET which is a global micrometeorological eddy covariance (EC) network for measuring carbon, water vapour, and energy exchanges between the biosphere and atmosphere (Baldocchi and Wilson, 2001). AmeriFlux core sites are the EC flux tower sites that deliver high-quality continuous data to the AmeriFlux database (http://ameriflux.lbl.gov). Currently, there are 44 core sites distributed in 12 clusters. We selected 13 out of 44 sites, which also represent the primary EC sites of the selected clusters. These sites also cover a broad class of aridity index (AI) (Food and Agriculture Organization, FAO, 2015): arid (AI<0.30), semiarid (0.50>AI>0.30), subhumid (0.65>AI>0.50), and humid (AI>0.65). Each of these four AI categories contained at least two validation sites. Four MODIS subsets (Fig. 2) covering at least two validation sites within each region



(labeled as East (E), Midwest 1 (MW1), Midwest 2 (MW2), and West (W), from the east to west, respectively) were used for image processing to implement the STIC1.2 model. For the regional-scale intercomparison of ET models, similar MODIS subsets were used.

## 2.3 Datasets

Key remotely sensed data for model implementation were obtained from the MODIS Terra 8-day composites. Meteorological inputs were obtained from hourly NLDAS-2 (North American Land Data Assimilation System - 2) forcing data (Xia et al., 2012). Daily meteorological variables, which were derived from hourly NLDAS and PRISM (Parameter-elevation Relationships on Independent Slopes Model; PRISM Climate Group, Oregon State University, http://prism.oregonstate.edu) data, were obtained from the University of Idaho (http://climate.nkn.uidaho.edu/METDATA/). A list of datasets used in the present analyses is given in Table 2. The PRISM precipitation dataset was used to select dry, wet, and normal years for each site.

## 2.4 Data processing

### 2.4.1 Selection of dry, wet and normal rainfall years

Dry, wet, and normal years were selected based on 30-year (1980-2010) precipitation from PRISM data. For each site, we selected the driest (dry), wettest (wet), and closest to the 30-year mean (normal) years based on PRISM precipitation data (Fig. 3).

### 2.4.2 MODIS-based variables: surface albedo, NDVI, LST, surface emissivity, and LAI

Broadband surface albedo was estimated using all the narrow band surface reflectances from MOD09A1, and NDVI was computed using near infrared and red band surface reflectance MOD09A1 products. $T_R$ information was obtained from the MOD11A2 LST products for the study years. For estimating surface emissivity, we took mean emissivity from band 31 and 32 (Bisht et al., 2005) from the MOD11A2 products. While the information of LAI from MOD15A2 and MCD15A2 products (mean of the two) were used for computing the extra resistance parameter ($kB^{-1}$) (Su, 2002;Su et al., 2001), NDVI was used for estimating $z_{0M}$ and $d_0$ (van der Kwast et al., 2009) in SEBS.

### 2.4.3 Meteorological Variables at the Satellite Overpass: RH, $T_A$, $u$, and $R_S$

Half-hourly gridded meteorological datasets from the North American Land Data Assimilation System (NLDAS-2) at 4 km × 4 km spatial resolution were used as inputs in the STIC1.2 ($R_S$, $T_A$, and RH) and SEBS ($u$, $R_S$, $T_A$, and RH) models. Because RH was not explicitly available in the NLDAS-2 dataset, we derived RH from surface pressure (Pa) and specific humidity (kg kg$^{-1}$) information using the method developed by McIntosh and Thom (1978). The half-hourly meteorological variables at the





time of MODIS Terra overpass during every 8-day period were averaged to ensure that the weather dataset is well representative of all the corresponding 8 days within each MODIS 8-day period.

Additional inputs of daily meteorology ($R_S$, $T_A$, $u$, and RH) required for computing 8-day ET were obtained from the University of Idaho (http://climate.nkn.uidaho.edu/METDATA/), and these data products were derived from hourly NLDAS and PRISM datasets. Daily weather data were also aggregated to the corresponding MODIS 8-day periods.

### 2.4.4 Derivation of regional-scale eight-day and annual ET maps (STIC1.2 and SEBS)

Net available energy ($\phi = R_N - G$, W m$^{-2}$) at MODIS Terra overpass time was partitioned into $H$ and $\lambda E$ by both models as explained in sections 2.1.1 and 2.1.2. Instantaneous $\Lambda$ was then computed as the ratio of $\lambda E$ to $\phi$. For the extrapolation of instantaneous $\lambda E$ to daily ET under clear sky conditions, the instantaneous $\Lambda$ is assumed to be constant for the day (Brutsaert and Sugita, 1992;Crago and Brutsaert, 1996) and 8-day cumulative ET (5-day for doy 361) was estimated as follows:

$$ET_8 = \frac{86400 \times 10^3 \times \Lambda \times R_{N24\text{-}8day} \times n}{\lambda p_w} \tag{17}$$

where $R_{N24\text{-}8day}$ is the 8-day net radiation; and $n$ = number of days in the 8-day period (8; $n$ = 5 for doy 361) computed using the ASCE standardized PM equation using daily weather inputs (ASCE-EWRI, 2005). Combining all the sites, the estimated $R_{N24\text{-}8day}$ from MODIS was within 10 % (i.e., 9 % overestimation) of mean observed 8-day net radiation at the flux sites (coefficient of determination, $R^2$, = 0.89, root mean squared error, RMSE, = 20 W m$^{-2}$, Fig. S1).

Annual ET maps were derived by summing all the corresponding 8-day ET maps within a given year. To fill the missing 8-day ET values, $\Lambda$ values from up to the two nearest 8-day periods were used (i.e. mean $\Lambda$ values of $n$ prior and after 8-day period, where $n$ = 1 or 2). The filled $\Lambda$ values were then used in Eq. (17)) ($R_{N24\text{-}8day}$ from the current 8-day period is used) to fill the missing 8-day ET values. Since there were missing daily flux data in some years, we filled missing values using linear interpolation between available days. For the statistical analysis, we retained those annual ET values when observed $\lambda E$ was available for at least 300 days at each flux tower site. Similarly, annual ET from the models was only compared when at least 38 (out of the 46) 8-day cumulative ET values were available.

### 2.4.5 Regional-scale eight-day and annual ET maps from MOD16 ET

The MOD16 ET product provides global 8-day (MOD16A2), monthly, and annual (MOD16A3) terrestrial ecosystem evapotranspiration datasets at 1 km × 1 km spatial resolution over 109.03 million km2 of global vegetated land areas. The dataset is currently available for the period of 2000-2014 and will be updated for years beyond 2014 in the future. The 8-day and annual MOD16 ET products were acquired from the Numerical Terradynamic Simulation group (ftp://ftp.ntsg.umt.edu/pub/MODIS/NTSG_Products/MOD16/MOD16A2.105_MERRAGMAO/) of the University of Montana. ET values of the corresponding flux sites for every 8-day period within each dry, wet, and normal year were extracted for model intercomparison. The annual ET maps from MOD16 products (MOD16A3) were used for regional-scale model intercomparison of annual ET estimates from STIC1.2 and SEBS.



### 2.4.6 Preparation of validation datasets

We used half-hourly surface energy balance flux data from the thirteen core EC sites of the AmeriFlux network that covers an aridity gradient (from arid to humid), and a wide range of elevation and biome types in the conterminous US (Table 1). A Bowen ratio (Bowen, 1926) based surface energy balance closure method (Chávez et al., 2005; Twine et al., 2000) was used to force the SEB closure at half-hour time scales. The half-hourly $\lambda E$ (W m$^{-2}$) was converted into ET (mm hr$^{-1}$) using the proportionality parameter between energy and equivalent water depth unit of ET (Mu et al., 2007; Velpuri et al., 2013).

$$ET = \lambda E/\lambda \tag{18}$$

Here $\lambda$ is the latent heat of vaporization of water.

Half-hourly ET data was then aggregated to hourly, daily, and eight-day temporal scales corresponding to the MODIS 8-day periods. The 8-day sum of ET was used for validating ET estimates from MOD16, SEBS, and STIC1.2 only when flux data were available for the entire 8-day period. Daytime fluxes ($H$, $\lambda E$, $R_N$, and $G$) close to MODIS Terra overpass time were also averaged over the 8-day periods corresponding to MODIS 8-day DOYs.

### 2.4.7 Statistical Analysis

The three ET models were evaluated based on their ability to estimate 8-day cumulative ET at the flux tower sites during dry, normal, and wet years. Widely used statistical metrics, such as RMSE, $R^2$, mean absolute error (MAE), and percent bias error (PBIAS) were used for evaluating the model performances. The location information of the AmeriFlux sites (Table 1 was used to extract the pixel values of ET (outputs from STIC1.2, SEBS, and MOD16 products) and other biophysical variables (Table 2) for the statistical analysis.

Comparisons were made for the 8-day periods when flux data were available for all 8 days corresponding to each MODIS 8-day period, and when MODIS inputs for STIC1.2 and SEBS, and MOD16 ET data were available. Overall, the data availability for statistical analysis ranged from 43 % (59 out of 138 MODIS 8-day periods) at the US-kon site to 93 % (128 out of 138 MODIS 8-day periods) at the US-Wkg site with an average of 65 % (SM, Table S1).

## 3 Results

### 3.1 What is the performance of STIC1.2 at the regional-scale across an aridity gradient and during contrasting rainfall years in the conterminous US?

Combining results from thirteen core AmeriFlux sites, it is apparent that STIC1.2 captured 66 % of the observed variability ($R^2 = 0.66$) in 8-day cumulative ET (Table 3) with an overall RMSE, MAE and PBIAS of 7.5 mm, 5.4 mm, and -3 %, respectively. Consistent performance of STIC1.2 was noted throughout dry, wet, and normal rainfall years, explaining about 64-69 % of the variability in 8-day cumulative ET (Fig. 4), with a slight overestimation in dry years (PBIAS 7 %) and an underestimation in wet years (PBIAS -11 %; Fig. 4). Biome-specific analysis revealed relatively better performance of STIC1.2





in forests as compared to non-forest sites (Fig. 5). STIC1.2 explained 73 % - 89 % variability in ET from ENF (evergreen needleleaf forests) and DBF (deciduous broadleaf forests) with an RMSE of 5.2 - 6.4 mm. Among the non-forest sites, although STIC1.2 explained 60 % - 70 % of the observed ET variability in CRO (croplands) and GRA (grasslands) (RMSE 7.2 - 9.9 mm/8-day), it explained only 23 % of the observed ET variability in WSA with a PBIAS of 44 % (Figure 5).

At the CRO sites, STIC1.2 underestimated ET by about 20 %. At the GRA sites, a better performance of STIC1.2 was noted in the dry year as compared to the wet and normal years (Fig. S2-S4). Regardless of vegetation type, STIC1.2 had a tendency to underestimate ET under high wetness conditions.

Performance evaluation of STIC1.2 across an aridity gradient suggests the better predictive capacity of STIC1.2 in subhumid and humid sites as compared to arid and semiarid sites (Figure 6). As seen in Figure 6, 41 % - 45 % of the variability in 8-day
ET was explained in arid and semiarid ecosystems (RMSE 5 - 7.5 mm/8-day and MAE 4.8 - 5.1 mm/8-day), which increased to 61 % -77 % in the humid and subhumid ecosystems (with RMSE 7 - 10 mm/8-day and MAE 5 - 7.5 mm/8-day). The key reason is that STIC1.2 does not effectively capture very low ET values in the semiarid and arid sites, particularly in woody savannas (Figure 5).

### 3.2 Comparison of STIC1.2 against other global ET models that are constrained by $T_R$ and RH

STIC1.2 showed relatively high accuracy when independently compared against observed ET at thirteen AmeriFlux sites than did SEBS and MOD16. Combining all sites, the predictive capability of STIC1.2 was found to be 7 % - 17 % better than SEBS and MOD16, which explained about 53 % and 59 % of the variability in observed 8-day ET, respectively (Table 3). As evident from PBIAS, SEBS has a tendency to overestimate and MOD16 has a tendency to underestimate 8-day cumulative ET by over 20% (28% from SEBS and -27 % from MOD16), while STIC1.2 has a small tendency to underestimate (-3 %) (Table 3). In
addition to a high RMSE (9.6 - 10.2 mm for SEBS, 8.5 - 9.4 mm for MOD16), an overestimation tendency of SEBS (PBIAS 13% - 44%) and underestimation tendency of MOD16 (PBIAS -25% to -32%) were consistent throughout dry, wet, and normal years (Figure 4).

The biome-specific performance intercomparison revealed that STIC1.2 produced a substantially lower RMSE than SEBS and MOD16 in ENF (12 % - 17 % less RMSE), GRA (18 % - 29 % less RMSE), and DBF (7 % - 37 % less RMSE) in 8-day
cumulative ET with better or tantamount skill in capturing the observed ET variability as compared to the two other models (Figure 5). While MOD16 was found to produce relatively lower RMSE in WSA (16 % less than STIC1.2 and 49 % less than SEBS), SEBS performed relatively better in CRO (5 % and 33 % less RMSE than STIC1.2 and MOD16, respectively).

Statistical intercomparison of the predictive capacity of STIC1.2 with respect to SEBS and MOD16 across an aridity gradient revealed notable differences in RMSE and MAE between the models (Fig. 6), despite general agreement on the capabilities of
individual models to explain the variability in observed ET ($R^2 = 0.34$-$0.77$). STIC1.2 was found to produce the lowest RMSE in 8-day cumulative ET in arid (31 % and 43 % lower than MOD16 and SEBS, respectively), semiarid (5 % and 32 % lower than MOD16 and SEBS, respectively), and humid (3 % and 19 % lower than MOD16 and SEBS, respectively) ecosystems (Fig. 6). In the subhumid ecosystem, the performance of STIC1.2 was comparable with SEBS (PBIAS from STIC1.2 and





SEBS were -20 % and 2 %, other error statistics were comparable) and substantially better than MOD16 (PBIAS=-48 %). A consistent overestimation (underestimation) tendency of SEBS (MOD16) in arid and semiarid ecosystems is reflected in positive (negative) PBIAS (58 % to 84 % for SEBS; -67 to -37 % for MOD16) in these two aridity classes.

### 3.3 Factors affecting agreements/disagreements between ET models

The residual differences in 8-day ET between STIC1.2 versus SEBS ($dET_{STIC1.2-SEBS}$ = $ET_{STIC1.2}$ - $ET_{SEBS}$) as well as SEBS versus observed ET ($dET_{SEBS-obs}$ = $ET_{SEBS}$ - $ET_{obs}$) were found to be significantly associated with $T_R$ ($r$ = -0.301 to 0.38, $p$-value < 0.005) and $D_A$ ($r$ = -0.30 to 0.46, $p$-value < 0.005) (Fig. 7a-7b). Negative $dET_{STIC1.2-SEBS}$ (positive $dET_{SEBS-obs}$) was found with increasing $T_R$ and $D_A$ above 290 K and 2 kPa, whereas ET differences were narrowed down below these limits (Fig. 7). A logarithmic pattern was found between $dET_{STIC1.2-SEBS}$ ($dET_{SEBS-obs}$) and NDVI, with a correlation of 0.31 and 0.35,

respectively. Major ET differences (both $dET_{STIC1.2-SEBS}$ and $dET_{SEBS-obs}$) (±20 mm) were found in the NDVI range of 0.15 - 0.35, whereas ET differences were diminished within ±10 mm above NDVI of 0.5.

A similar analysis of ET differences between STIC1.2 and MOD16 ($dET_{STIC1.2-MOD16}$ = $ET_{STIC1.2}$ - $ET_{MOD16}$) and between MOD16 and the observed ET ($dET_{MOD16-obs}$ = $ET_{MOD16}$ - $ET_{obs}$) also revealed a significant correlation with $T_R$ and $D_A$ (Fig. 7d-7e and inset) ($r$ = -0.30 to 0.66, $p$-value < 0.005), but the direction of these correlations are opposite to those found with the

ET differences between STIC1.2 and SEBS. $dET_{MOD16-obs}$ was found to have no significant relationship ($p$-value > 0.15) with NDVI, while $dET_{STIC1.2-MOD16}$ appear to have a significant negative relationship with NDVI, which was also opposite of what found with ET differences between STIC1.2 and SEBS.

To examine the relative importance of the meteorological and land surface variables in explaining the variances in $dET_{STIC1.2-SEBS}$ and $dET_{STIC1.2-MOD16}$, a random forest analysis (Liaw and Wiener, 2002) was performed between the residual ET differences

and seven climatic/land surface variables (NDVI, $D_A$, P, $u$, observed soil moisture [SM], $T_A$, and $T_R$) as predictors (Fig. S5). Overall, these variables explained 41 % and 57 % variances in $dET_{STIC1.2-SEBS}$ and $dET_{STIC1.2-MOD16}$, respectively. The most important variables for explaining variance in $dET_{STIC1.2-SEBS}$ were $T_A$ and NDVI. These two variables would lead to about 25-40 % increase in mean residual errors (MSEs) if they are permuted in the random forest model. For $dET_{STIC1.2-MOD16}$, all the variables expect $u$ appeared to be important in determining the variance of ET difference, as each variable would lead to about

17 %-22 % increase in MSEs if they are permuted in the random forest model.

### 3.4 Regional-scale intercomparison of STIC1.2 versus SEBS and MOD16 ET

Annual ETs from STIC1.2 for the driest, wettest, and normal precipitation years for each of four study zones during the period 2001-2014 were compared against those derived from SEBS and the MOD16A3 annual ET products. Because the study years were selected based on the spatial mean of precipitation across 4 km x 4 km PRISM grids, the study years (Table 4) do not

necessarily match with those considered for ET analysis over the flux sites as presented in sections 3.1, 3.2, and 3.3.

Figs. 8-11 present annual ET maps for the driest, wettest, and normal years for each of the four study zones covering all thirteen study sites and a distinct positive relationship was found between annual ET computed from the three models.




However, the magnitude of annual ET from the three models varied widely, particularly in the relatively dry zones of the Midwestern US (MW1 and MW2). Such differences in annual ET could be attributed to the systematic differences in 8-day cumulative ET among the three models (i.e., overestimation from SEBS and underestimation from MOD16).

The mean percent difference (and standard deviation) in ET between STIC1.2 vs. SEBS and MOD16 (Table 5) from all pixels within the bounding box of four study zones during the contrasting rainfall years (as in Fig. 8 - 11) showed noteworthy disagreements in arid and semiarid (W and MW2) zones, where annual ET from SEBS (MOD16) were 66-85 % more (11-55 % less) than STIC1.2. Conversely, major agreements between the models were found in the humid (E) zone where SEBS and MOD16 annual ET estimates were within 13 % of STIC1.2 ET.

We further compared annual ET estimates from the models against the flux tower estimates for the years listed in Table 5 and annual ET maps corresponding to Figs. 8-11. Comparison of annual ET at the core AmeriFlux sites revealed a consistent overestimation and underestimation from SEBS (PBIAS 23 %) and MOD16 (PBIAS -30 %) (Fig. 12). Despite the uncertainties due to linear interpolation for missing days or the use of neighbouring 8-day $\Lambda$ in computing annual ET (as mentioned in section 0), STIC1.2 produced the lowest RMSE (175 mm) and MAE (134 mm) as compared to SEBS (RMSE 239 mm, MAE 188 mm) and MOD16 (RMSE 261 mm, MAE 228 mm) (Fig. 12).

Figure 12 provides evidence that errors in 8-day cumulative ET from SEBS and MOD16 were largely additive, as indicated by the consistent overestimation or underestimation from the models at different sites. Notably, MOD16 estimates were particularly poor in the MW2 zones, while SEBS was found to be poor both in the MW1 and MW2 zones. Apart from that, differences between STIC1.2 and the other two models were also noticed in other zones.

To further investigate the role of biomes on ET differences between STIC1.2 and other models, we computed the mean percent ET difference (standard deviation) (similar to Table 5) on the five vegetation types, corresponding to those represented by the core AmeriFlux sites. The differences in annual ET between STIC1.2 vs. SEBS and STIC1.2 vs. MOD16 were mostly evident in all five vegetation classes, particularly in the W and MW2 spatial domains, with the maximum ET differences in grasslands (-135 % to 44 %) (Table 6). For almost all of the five vegetation types, ET differences between the models decreased across the aridity gradient from arid to humid ecosystems from western to the eastern US (±20 %).

In order to quantify the relative contribution of these three categorical variables [e.g., (1) zones (W, MW2, MW2, E), (2) land cover types (five land cover classes), and 3) precipitation extremes (dry, wet, and normal years)] to variations in residual ET differences (annual) between STIC1.2 and the other two models, we performed a random forest analysis (Fig. S6). The three categories together explain 45 % to 60 % of the variances in the residual ET difference between STIC1.2 vs. MOD16 and STIC1.2 vs. SEBS. However, study zone increases 51 % - 65 % of mean residual errors (MSEs) in ET if this group is permuted in the random forest model, thus appearing to be the most important factor among the three categorical variables. This finding is also consistent with the results in presented in Tables 5 and 6 that the residual ET differences between the models progressively reduced across an aridity gradient from arid to humid ecosystems. The precipitation extremes appeared to have no effect on the residual ET difference between STIC1.2 and SEBS, similar to the land cover effect on the residual difference between STIC1.2 and MOD16.



## 4 Discussion

Overall, STIC1.2 performed reasonably well across an aridity gradient and a wide range of biomes in the conterminous US. One noticeable weakness of STIC1.2 appears to be its tendency to underestimate ET in the grassland and cropland biomes (Fig. 4 and 5). These biases could be attributed to the nature of the MODIS LST product that aggregates sub-grid heterogeneity

in $T_R$, vegetation cover, and radiation at 1 km × 1 km area. Due to the relatively low tower heights in CRO and GRA sites (3 - 10 m), the EC towers aggregate fluxes at scales of approximately 0.009 - 0.10 km$^2$. Such a critical mismatch of the scales between MODIS pixels and the flux tower footprint could be a potential source of disagreement between STIC1.2 and tower-observed ET (Stoy et al., 2013). Another source of error could be the presence of widely varied dry and wet patches within one MODIS 1 km × 1 km pixel as well as around the flux towers. For example, if more than 50 % of the area falling within a

1 km × 1 km MODIS pixel is predominantly dry, the lumped $T_R$ signal in MODIS LST product will be biased due to the dryness of the landscape (Stoy et al., 2013;Mallick et al., 2014;Mallick et al., 2015) and the resultant ET will be underestimated. The overestimation tendency in WSA is mainly due to the poor performance of STIC1.2 in the Tonzi Ranch site, which could be associated with the emissivity correction uncertainties and systematic underestimation of MODIS LST in arid and semiarid ecosystems (Wan and Li, 2008;Jin and Liang, 2006;Hulley et al., 2012). Since $T_R$ plays an important role in constraining the

conductances in STIC1.2, an underestimation of $T_R$ would ultimately result in an overestimation of $g_C$ and underestimation of $g_A$, which would result in overestimation of ET. The differences between STIC1.2 versus observed ET in WSA may also largely be attributed to the Bowen ratio energy balance closure correction of EC $\lambda E$ observations (Chávez et al., 2005;Twine et al., 2000). Although the Bowen ratio correction forces SEB closure, in arid and semiarid ecosystems major corrections are generally observed in sensible heat flux, whereas $\lambda E$ is negligibly corrected (Chávez et al., 2005;Mallick et al., In Review).

Besides, direct water vapour adsorption on the land surface occurs in arid and semiarid ecosystems when air close to the surface is drier than the overlying air (McHugh et al., 2015;Agam and Berliner, 2006), and this source of moisture is unaccounted for in the EC measurements. This will automatically result in disagreement between STIC1.2 and observed ET. Nevertheless, the performance of STIC1.2 in forest ecosystems is encouraging, given the uncertainties associated with more complex SEB models that use MOST to parameterize the turbulent mixing in tall canopies (Finnigan et al., 2009;Garratt, 1978;Harman and

Finnigan, 2007) that could induce substantial biases in estimated fluxes (Wagle et al., 2017;Numata et al., 2017;Bhattarai et al., 2016)

Performance intercomparison of STIC1.2 with SEBS and MOD16 indicated overall low statistical errors for STIC1.2, and better agreement than SEBS and MOD16 with observed ET values. The principal differences between STIC1.2 and SEBS (as evident from Fig. 7a and Fig. 8 to 13), in particular, the overestimation of ET through SEBS, is in cases of high $T_R$ and $D_A$

with low vegetation cover (i.e., low NDVI), a characteristic feature of arid and semiarid ecosystems. In these water limited ecosystems, $T_R$ induced water stress and the diminishing ET rate leads to high atmospheric dryness (i.e., high $D_A$), increased evaporative potential, and very high sensible heat flux. This leads to substantial differences between $T_R$ and $T_0$, and the role of radiometric roughness length ($z_{0H}$) becomes critical, which is estimated empirically through the adjustment factor $kB^{-1}$ (Paul



et al., 2014). Although there is a first order dependence of $kB^{-1}$ on $T_R$, radiation, and meteorological variables (Verhoef et al., 1997b), no physical model of $kB^{-1}$ is available (Paul et al., 2014). Therefore, uncertainties in $kB^{-1}$ estimation are propagated into $z_{0H}$. Overestimation (or underestimation) of $z_{0H}$ would lead to underestimation (overestimation) of $g_A$ in SEBS, which is mirrored in ET differences between SEBS vs. observations ($dET_{SEBS-obs}$) (Zhou et al., 2012). This is also evident when a logarithmic pattern was found between $dET_{SEBS-obs}$ and $kB^{-1}$, with a correlation of 0.39 (*p-value* < 0.005) (Fig. 13a). Major ET differences were found (±20 mm) within a $kB^{-1}$ range of 2-6 (arid, semiarid, heterogeneous vegetation), whereas ET differences were diminished within ±10 mm above $kB^{-1}$ of 6 (subhumid, humid, homogeneous vegetation). Apart from $z_{0H}$, empirical parameterization of $z_{0M}$ and a resultant ±50 % uncertainties in $z_{0M}$ can also lead to 25 % errors in $g_A$ estimation (Liu et al., 2007;Verhoef et al., 1997a), which will lead to more than 30 % uncertainty in ET estimates. This is also evident from the exponential scatter between $z_{0M}$ and $dET_{SEBS-obs}$ (Fig. 13b) that showed a significant negative correlation between $z_{0M}$ and the residual ET error ($r = -0.40$, *p*-value < 0.005).

It is important to emphasize that the momentum transfer equation for estimating $g_A$ in SEBS is based on the semi-empirical MOST approach that mainly holds for extended, uniform, and flat surfaces (Foken, 2006;Verhoef et al., 1997c). MOST tends to become uncertain on rough surfaces due to a breakdown of the similarity relationships for heat and water vapour transfer in the roughness sub-layer, which results in an underestimation of the 'true' $g_A$ by a factor 1-3 (Holwerda et al., 2012a;van Dijk et al., 2015a;Simpson et al., 1998). Since $g_A$ is the main anchor in SEBS, an underestimation of $g_A$ would lead to an underestimation of sensible heat flux and an overestimation of ET (Gokmen et al., 2012;Paul et al., 2014). Also, due to the priority of estimating $g_A$ and $H$, SEBS appears to ignore the important feedbacks between $g_C$, $D_A$, $\phi$ and transpiration (which are included in STIC1.2) which consequently led to differences between STIC1.2 and SEBS. Relatively better performance of SEBS at croplands, as well as in wet years could be attributed to the ability of the model to perform well in predominantly homogeneous vegetation and under wet conditions where the differences between $T_R$ and $T_0$ are not critical.

The wide use of the global MOD16 ET product for calculating regional water and energy balances should be evaluated on a case-by-case basis as one could come to different conclusions using ET outputs from the other two models considered in this study. A significant underestimation of actual ET by the MOD16 ET products, particularly in arid and semiarid conditions has already been reported (Hu et al., 2015;Ramoelo et al., 2014;Feng et al., 2012). Conversely, others have reported better performance of MOD16 ET products in humid climates (Hu et al., 2015) and forest ecosystems (Kim et al., 2012), consistent with the performance of the model in the two flux sites in NC in our study (Table S1). Underestimation of ET by the MOD16 ET products in croplands has also been reported (Velpuri et al., 2013;Kim et al., 2012), though not to the same degree as we found in this study. This consistent underestimation of MOD16 ET in croplands and grasslands, and associated statistical errors in other biomes could be associated with both aerodynamic and canopy conductance parameterizations and use of biome property look-up tables for assigning biome-specific minimum and maximum conductances. Additionally the empirical scaling functions used for constraining the conductances and the spatial scale mismatch between MODIS and flux towers could also introduce additional uncertainties in MOD16 ET. Similarly, our results suggests that caution should be taken when applying SEBS under extreme dry condition, and also for grasslands, savannas, and deciduous broadleaf forests. The overestimation of





grassland ET from SEBS is consistent with a recent study (Bhattarai et al., 2016), which could be attributed to uncertain characterization of $z_{OH}$ (Gokmen et al., 2012). However, the performance of SEBS was relatively better under wet conditions, and in homogeneous croplands and evergreen needleleaf forests (Fig. 5, Table S1). However, for regional-scale ET modelling in heterogeneous landscapes, STIC1.2 appears to be better than the other two models given its consistency across a wide range of biomes and aridity conditions.

## 5 Conclusions

This paper establishes the first ever regional-scale implementation of a simplified thermal remote sensing based model, Surface Temperature Initiated Closure (STIC1.2) for spatially explicit ET mapping, which is independent of any empirical parameterization of aerodynamic/surface conductances and aerodynamic temperature. By combining MODIS land surface temperature, surface reflectances, and gridded weather data, we demonstrate the promise of STIC1.2 to generate regional ET at 1 km × 1 km spatial resolution in the conterminous US. Independent validation of STIC1.2 using observed flux data from a dry, wet, and normal precipitation years at thirteen core AmeriFlux sites covering a wide range of climatic, biome, and aridity gradients in the US led us to the following conclusions.

(i) Overall, STIC1.2 explained significant variability in the observed 8-day cumulative ET with a root mean square error (RMSE) of less than 1 mm/day and was robust throughout dry, wet, and normal years. Biome-wise evaluation of STIC1.2 suggests the smallest errors in forest ecosystems, followed by grassland, cropland, and woody savannas. Underestimation of ET in croplands is mainly attributed to the spatial scale mismatch between a MODIS pixel and the flux tower footprint in croplands, and an overestimation of ET in woody savannas is mainly attributed to the large uncertainties in the MODIS LST product in savannas, and surface energy balance closure correction of eddy covariance ET observations.

(ii) STIC1.2 performed substantially better or comparable to SEBS and MOD16 in a broad spectrum of aridity, biome, and dry-wet extremes. Model evaluation in different aridity conditions suggests that all three models performed better under sub-humid and humid conditions as compared to arid or semi-arid conditions.

(iii) The principal difference between STIC1.2 and SEBS ET appears to be associated with the differences in aerodynamic conductance estimation between the two models. Empirical characterization of $z_{0M}$ and $kB^{-1}$ in SEBS are found to be the major factors creating uncertainties in aerodynamic conductance and ET estimations in SEBS, which is eventually responsible for large ET differences between the two models. Similarly, the differences in aerodynamic and surface conductance estimation between STIC1.2 and MOD16 could also be responsible for ET differences between the two models.

(iv) STIC1.2 is highly sensitive to uncertainties in $T_R$ and hence accurate $T_R$ maps are needed for reliable ET estimates, which are currently missing in arid and semiarid ecosystems. However with the improved emissivity corrected $T_R$ from new the MODIS LST product (MOD21; Hulley et al., 2014;Hulley et al., 2016), an improved performance of STIC1.2 is expected in woody savannas. Alternatively, the use of time difference $T_R$ from MODIS Terra Aqua can also help diminish STIC1.2 errors in woody savannas. Besides, gridded weather inputs (air temperature, RH, solar radiation), ideally at the resolution of $T_R$, are




required for STIC1.2 implementation and hence any errors associated with the weather inputs will create biased model outputs. These insights should provide guidance for future implementations of STIC1.2 in the US and other regions.

**Acknowledgements**

The study was partly supported by NSF SEES Fellowship (Award # 1415436) and NASA New Investigator Award
(NNH15ZDA001N) to Meha Jain. Kaniska Mallick was supported by Luxembourg Institute of Science and Technology (LIST) through the project BIOTRANS (grant number, 00001145) and CAOS-2 project grant (INTER/DFG/14/02) funded by FNR (Fonds National de la Recherche) - DFG (German Science Foundation). This project also contributes to HiWET consortium funded by Belgian Science Policy (BELSPO) - FNR under the programme STEREOIII (INTER/STEREOIII/13/03/HiWET; CONTRACT NR SR/00/301).

Funding for AmeriFlux core site data was provided by the U.S. Department of Energy's Office of Science. The authors would like to thank all the principal investigators: Dr. Russell Scott, USDA- ARS (US-Wkg, US-SRM, and US-SRG); Asko Noormets, North Carolina State University (US-NC1 and US-NC2); Dr. Sebastien Biraud, Lawrence Berkeley National Lab (US-ARM); Dr. Dennis Baldocchi, University of California, Berkeley (US-Ton); Dr. Nathaniel A. Brunsell (NAB), University of Kansas (US-KFS and US-Kon); Dr. Kim Novick, Indiana University (US-MMS); Dr. Peter Blanken, University of
Colorado (US-NR1); Andy Suyker, University of Nebraska, Lincoln (US-Ne1); and Dr. Bev Law, Oregon State University (US-Me2) for maintaining and providing access to the flux data for free. NAB acknowledges fundings for the US-Kon site through the NSF Long Term Ecological Research grant to the Konza Prairie (DEB-0823341), and for the US-Kon and US-KFS sites through AmeriFlux core site funding from the US Department of Energy under a sub contract from DE-AC02-05CH11231 and additional funding support through the USDA-AFRI 2014-67003-22070. The authors would also like to thank
Dr. Bob Su and Dr. Xuelong Chen from the University of Twente, Netherlands for answering queries related to the SEBS model. Authors declare no conflict of interest.

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



**Table 1. An overview of the thirteen core AmeriFlux sites used for the validation of the STIC1.2 model.**

| Site Name | Latitude | Longitude | Elevation | Biome[*] | Average $T_A$(C) | Average Annual P (mm) | Climate[**] | Aridity Index (AI[***]) | Reference |
|---|---|---|---|---|---|---|---|---|---|
| US-Me2 | 44.4523 | -121.557 | 1253 | ENF | 6.28 | 523 | M | 1.004 | Thomas et al. (2009) |
| US-Ton | 38.4316 | -120.966 | 177 | WSA | 15.8 | 559 | M | 0.440 | Baldocchi et al. (2004) |
| US-SRM | 31.8200 | -110.8700 | 1120 | WSA | 17.92 | 380 | ASC | 0.258 | Scott (2016a) |
| US-SRG | 31.7894 | -110.828 | 1291 | GRA | 17 | 420 | ASC | 0.317 | Scott (2016b) |
| US-Wkg | 31.7365 | -109.942 | 1531 | GRA | 15.64 | 407 | ASC | 0.225 | Scott (2016c) |
| US-NR1 | 40.0329 | -105.546 | 3050 | ENF | 1.5 | 800 | SA | 0.478 | Monson et al. (2005) |
| US-Kon | 39.0824 | -96.5603 | 330 | GRA | 12.77 | 867 | HS | 0.674 | Logan and Brunsell (2015) |
| US-KFS | 39.0561 | -95.1907 | 310 | GRA | 12 | 1014 | HS | 0.807 | |
| US-ARM | 36.6058 | -97.4888 | 314 | CRO | 14.76 | 843 | HS | 0.551 | Biraud (2002) |
| US-Ne1 | 41.1651 | -96.4766 | 361 | CRO | 10.07 | 790 | HC | 0.645 | Suyker (2016) |
| US-MMS | 39.3232 | -86.4131 | 275 | DBF | 10.85 | 1032 | HS | 0.984 | Philip and Novick (2016) |
| US-NC1 | 35.8118 | -76.7119 | 5 | ENF | 16.6 | 1320 | HS | 1.031 | Domec et al. (2015), Sun et al. (2010) |
| US-NC2 | 35.8030 | -76.6685 | 5 | ENF | 16.6 | 1320 | HS | 1.031 | Domec et al. (2015), Sun et al. (2010) |

* WSA = Woody savanna, GRA = Grassland. ENF = Evergreen needleleaf forest, DBF = Deciduous broadleaf forest, CRO=croplands

**M = Mediterranean, ASC= Arid steppe cold, SA= sub artic, HS = Humid subtropical, HC = Humid continental

5 *** AI = $P/E_P$ (Food and Agriculture Organization, FAO, 2015). We categorized the sites into arid (AI<0.30), semiarid (0.50>AI>0.30), subhumid (0.65>AI>0.50), and humid (AI>0.65) zones, such that each AI category contained at least two validation sites.



**Table 2.Descriptions of MODIS and meteorological datasets used in this study.**

| Dataset Name | Variables | Spatial Resolution | Temporal | Source |
|---|---|---|---|---|
| MOD11A2 | Land surface Temperature, emissivity | 1 km × 1 km | 8-day | Wan et al. (2015) |
| MOD09A1 | Surface reflectance, Albedo, NDVI | 1 km × 1 km | 8-day | Vermote (2015) |
| MOD15A2/MCD15A2 | LAI | 1 km × 1 km | 8-day | Myneni et al. (2002) |
| NLDAS | $T_A$, RH, $R_S$, $u$ | 4 km × 4 km | hourly | Mitchell et al. (2004); Xia et al. (2009) |
| University of Idaho Gridded Surface Meteorological Data | $T_A$, RH, $R_S$, $u$ | 4 km × 4 km | Daily | Abatzoglou (2013) |
| PRSIM | Precipitation | 4 km × 4 km | Daily | PRISM Climate Group, Oregon State University |





**Table 3. Evaluation of 8-day cumulative ET from STIC1.2, SEBS, and MOD16 against observed ET from thirteen core**
5 **AmeriFlux sites in the US combining data from one dry, one wet, and one normal year.**

| Model | $R^2$ | RMSE (mm) | MAE (mm) | PBIAS (%) |
|---|---|---|---|---|
| STIC1.2 | 0.66 | 7.5 | 5.4 | -3 |
| SEBS | 0.53 | 9.8 | 7.3 | 28 |
| MOD16 | 0.59 | 8.9 | 6 | -27 |

**Table 4. Study years considered for regional-scale intercomparison of annual ETs from STIC1.2, SEBS, and MOD16**

| Zone (2001-2014 mean annual P, mm) | Dry year (annual P, mm) | Wet year (annual P, mm) | Normal year (annual P, mm) |
|---|---|---|---|
| W (838) | 2013 (397) | 2010 (1021) | 2014 (856) |
| MW2 (403) | 2012 (259) | 2010 (428) | 2005 (403) |
| MW1 (1037) | 2012 (786) | 2008 (1313) | 2014 (1023) |
| E (1210) | 2007 (915) | 2003 (1643) | 2010 (1220) |

**Table 5. Mean percentage difference in annual ET (standard deviation) between STIC1.2 vs. SEBS and MOD16 from all pixels within the bounding box of four study zones during dry, wet, and normal years.**

| Years | West | | Mid-West 2 | | Mid-West 1 | | East | |
|---|---|---|---|---|---|---|---|---|
| | STIC 1.2 - SEBS | STIC1.2- MOD16 | STIC 1.2 - SEBS | STIC1.2- MOD16 | STIC 1.2 - SEBS | STIC1.2- MOD16 | STIC1.2- SEBS | STIC1.2- MOD16 |
| Dry | -69 | 15 | -85 | 55 | -22 | 26 | -12 | 11 |
| | (58) | (23) | (37) | (23) | (9) | (13) | (7) | (17) |
| Wet | -66 | 11 | -73 | 43 | -33 | -8 | -13 | -1 |
| | (53) | (20) | (34) | (23) | (13) | (14) | (7) | (14) |
| Normal | -72 | 21 | -78 | 43 | -25 | 6 | -13 | 6 |
| | (58) | (21) | (34) | (24) | (8) | (12) | (7) | (15) |

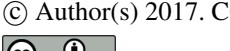



**Table 6. Mean percent difference in annual ET (standard deviation) between STIC1.2 vs SEBS and MOD16 within the bounding box of the four study zones considering all pixels and five vegetation types based on MCD12Q1 products (Friedl et al., 2010).**

| Zones | STIC1.2 - SEBS | | | | | | STIC1.2 – MOD16 | | | | | |
|---|---|---|---|---|---|---|---|---|---|---|---|---|
| | All | ENF | DBF | WSA | GRA | CRO | All | ENF | DBF | WSA | GRA | CRO |
| | -93 | -59 | NA | -55 | **-135** | -49 | 24 | 26 | NA | 16 | 30 | 18 |
| W | (62) | (34) | (NA) | (35) | (60) | (19) | (23) | (24) | (NA) | (23) | (22) | (17) |
| | -86 | -49 | -47 | -61 | -94 | -58 | 42 | 32 | 13 | 44 | **44** | 25 |
| MW2 | (34) | (24) | (30) | (23) | (31) | (31) | (23) | (20) | (35) | (19) | (23) | (36) |
| | -29 | -17 | -13 | -22 | -31 | -31 | 15 | 25 | -1 | 6 | 18 | 17 |
| MW1 | (11) | (11) | (9) | (10) | (9) | (9) | (10) | (17) | (17) | (22) | (20) | (19) |
| | -15 | -9 | -11 | -15 | -17 | -19 | 7 | 5 | -3 | 5 | 19 | 18 |
| E | (7) | (9) | (7) | (7) | (6) | (6) | (6) | (21) | (12) | (15) | (14) | (12) |





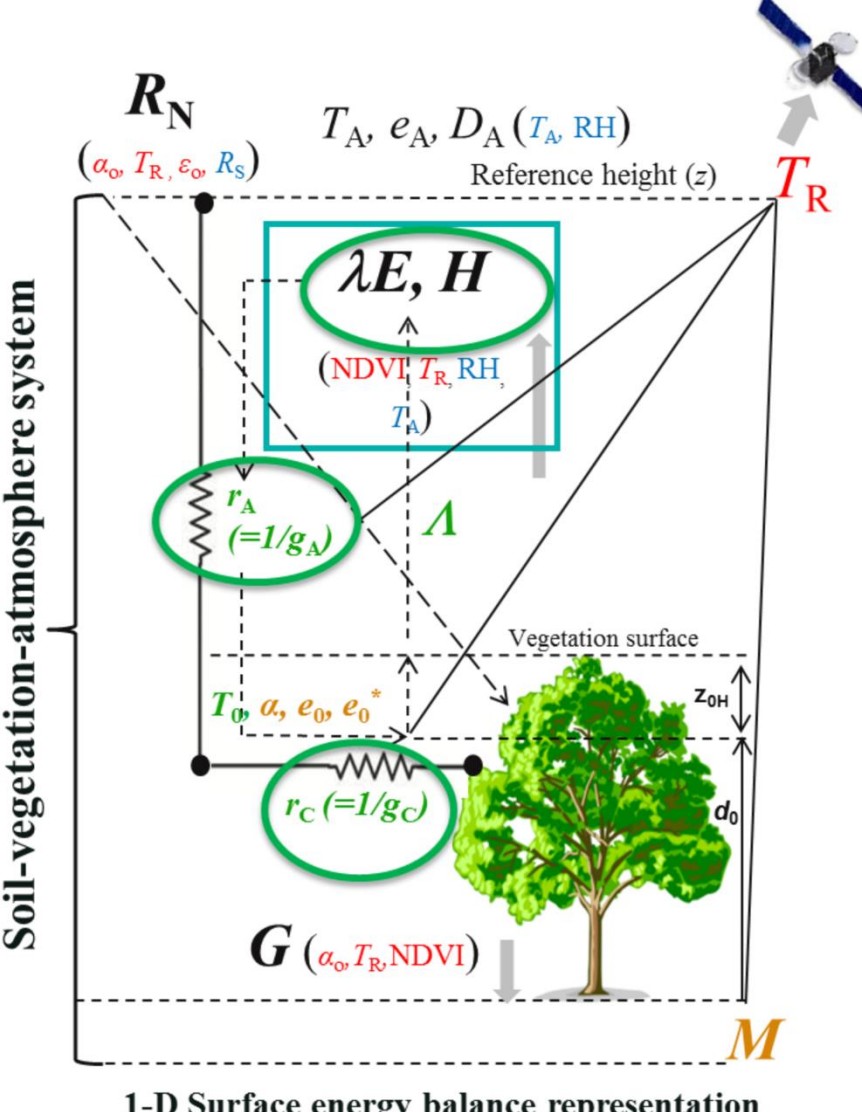

**Figure 1: Schematic representation of one-dimensional description of STIC1.2 showing how a feedback is established between the surface layer evaporative fluxes and source/sink height mixing and coupling (dotted arrows between $e_0$, $e_0^*$, $g_A$ and $g_C$, and $\lambda E$). Here $r_A$ and $r_C$ ($g_A$ and $g_C$) are the aerodynamic and canopy resistances (conductances); $e_0^*$ is the saturation vapour pressure at the source/sink height; $T_0$ is the source/sink height temperature (i.e. aerodynamic temperature) that is responsible for transferring the sensible heat ($H$); $e_0$ and $e_S$ are the vapour pressure at the source/sink height and the surface, respectively; $z_{0H}$ is the roughness length for heat transfer, $d_0$ is the displacement height; $T_R$ is the radiometric surface temperature; $M$ is the surface moisture availability or evaporation coefficient; $R_N$ and $G$ are net radiation and ground heat flux; $T_A$, $e_A$, and $D_A$ are temperature, vapour pressure, and vapour pressure deficit at the reference height ($z$); and $\lambda E$ and $H$ are latent and sensible heat fluxes, respectively. Inputs from MODIS land surface products and gridded weather datasets for the regional implementation of STIC1.2 in this paper are shown in red and blue fonts, respectively. Texts in green font represent the state variables for which analytical solution was obtained by solving the 'state equations' (Eqs. (7)-(10)). Texts in burnt orange are the variables that were obtained iteratively along with the state variables.**

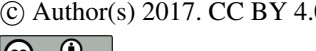


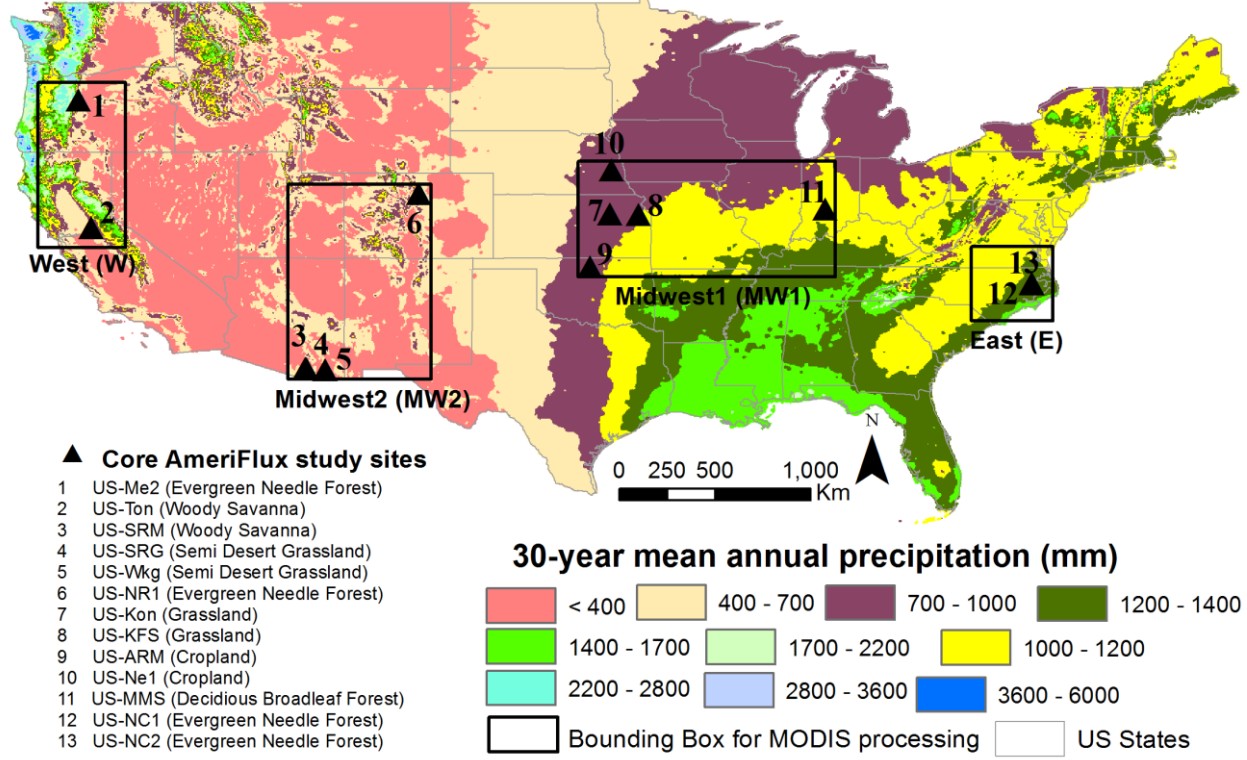

**Figure 2: Distribution of core AmeriFlux sites (13) used in this study shown over 30-year (1980-2010) mean annual precipitation of the US and the processing grids (MODIS subsets) used to estimate regional-scale ET from MODIS datasets.**





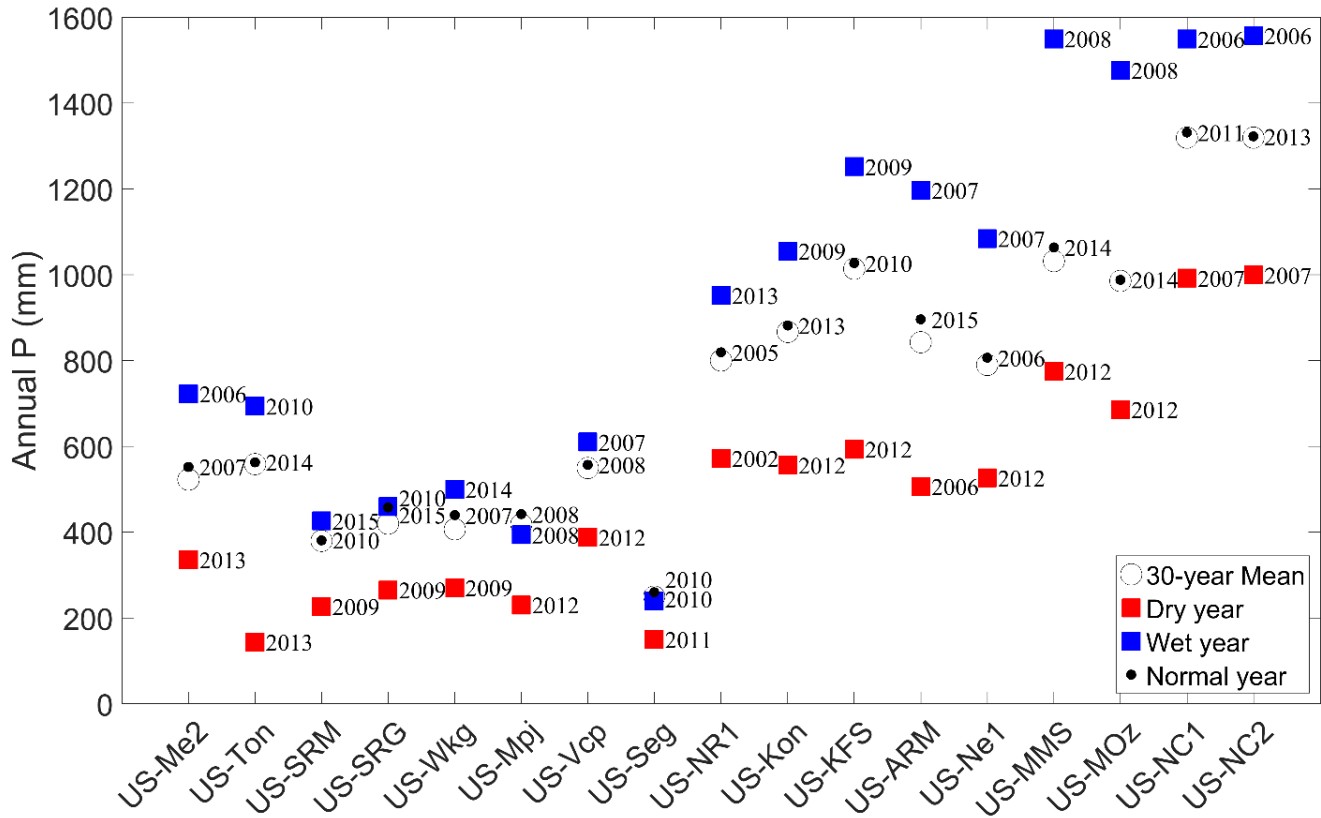

**Figure 3: Distribution of annual precipitation during dry, wet, and normal years considered for ET evaluation at each site corresponding to its 30-year mean annual precipitation from the PRISM data.**





**Figure 4: Evaluation of 8-day cumulative ET from STIC1.2, SEBS, and MOD16 against observed ET from thirteen core AmeriFlux sites in the US during dry, wet, and normal years.**





**Figure 5: Validation of 8-day cumulative ET from STIC1.2, SEBS, and MOD16 for each biome type.**





**Figure 6: Evaluation of 8-day cumulative ET from STIC1.2, SEBS, and MOD16 for each long-term aridity index (AI) category.**





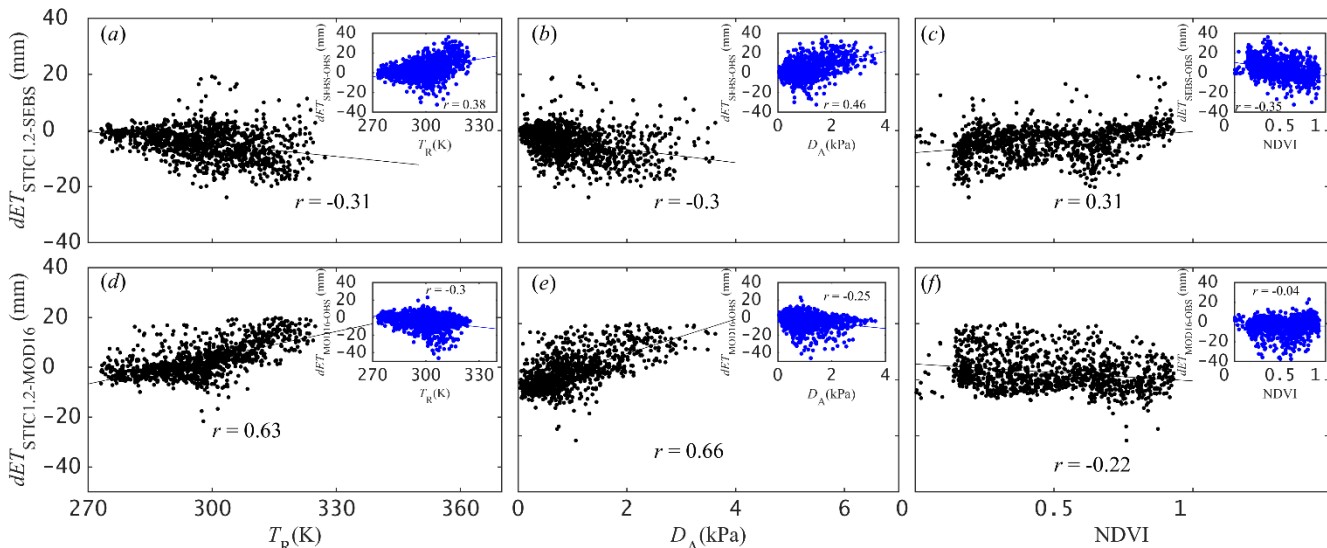

**Figure 7: Scatter plots of differences in STIC1.2 and (top row) SEBS and (bottom row) MOD16 ET estimates against input land surface variables used in these models ($T_R$, $D_A$, and NDVI). The pearson correlation coefficient, $r$ ($p$-value was < 0.005 for all cases except $dET_{MOD16-obs}$ vs. NDVI relationship), is also shown in each plot.**





**Figure 8: Annual ET (mm) maps for the dry, wet, and normal years derived from STIC1.2, SEBS, and MOD16 for the western (W) bounding box covering US-Ton and US-Me2 flux sites (Fig. 1). Scatterplots between annual ET estimates from STIC1.2 vs. SEBS and MOD16 are shown on the right.**





**Figure 9: Annual ET (mm) maps for the dry, wet, and normal years derived from STIC1.2, SEBS, and MOD16 for the mid-western 2 (MW2) bounding box covering US-ARM, US-SRG, US-Wkg, and US-NR1 flux sites (Fig. 1). Scatterplots between annual ET estimates from STIC1.2 vs. SEBS and MOD16 are shown on the right.**





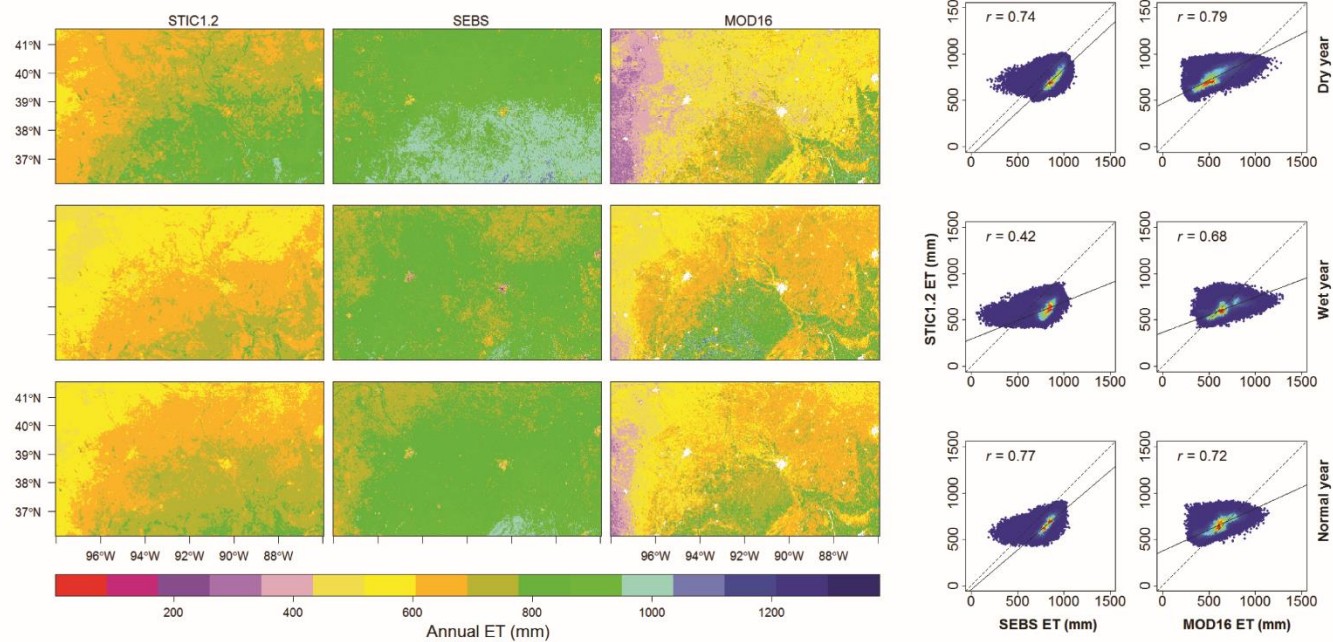

**Figure 10: Annual ET (mm) maps for the dry, wet, and normal years derived from STIC1.2, SEBS, and MOD16 for mid-western 1 (MW1) bounding box covering US-Kon, US-KFS, US-ARM, US-Ne1, and US-MMS flux sites (Fig. 1). Scatterplots between annual ET estimates from STIC1.2 vs. SEBS and MOD16 are shown on the right.**





**Figure 11: Annual ET (mm) maps for the dry, wet, and normal years derived from STIC1.2, SEBS, and MOD16 for the eastern (E) bounding box covering US-NC1 and US-NC2 flux sites (Fig. 1). Scatterplots between annual ET estimates from STIC1.2 vs. SEBS and MOD16 are shown on the right.**





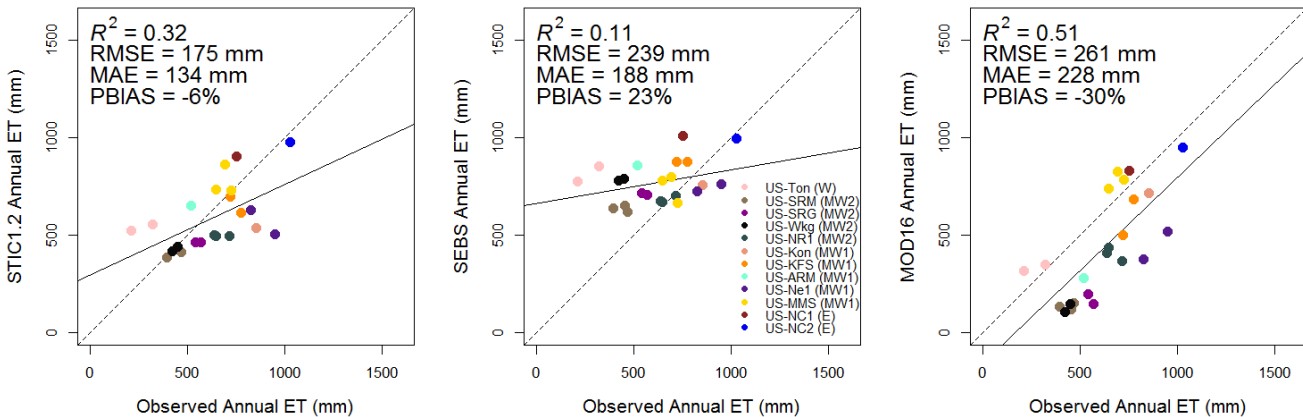

**Figure 12: Comparison of annual ET from STIC1.2, SEBS, and MOD16 against observed annual ET from the core AmeriFlux sites. Missing daily observed ET at the flux sites were filled using linear interpolation between available days. Missing 8-day cumulative ET from STIC1.2 and SEBS were filled using constant EF approach. Annual ET from the models and flux sites are compared when at least 38 (out of 46) 8-day cumulative ET were available for computation of annual ET and at least 300 days of observed $\lambda E$ were available at the flux tower sites.**




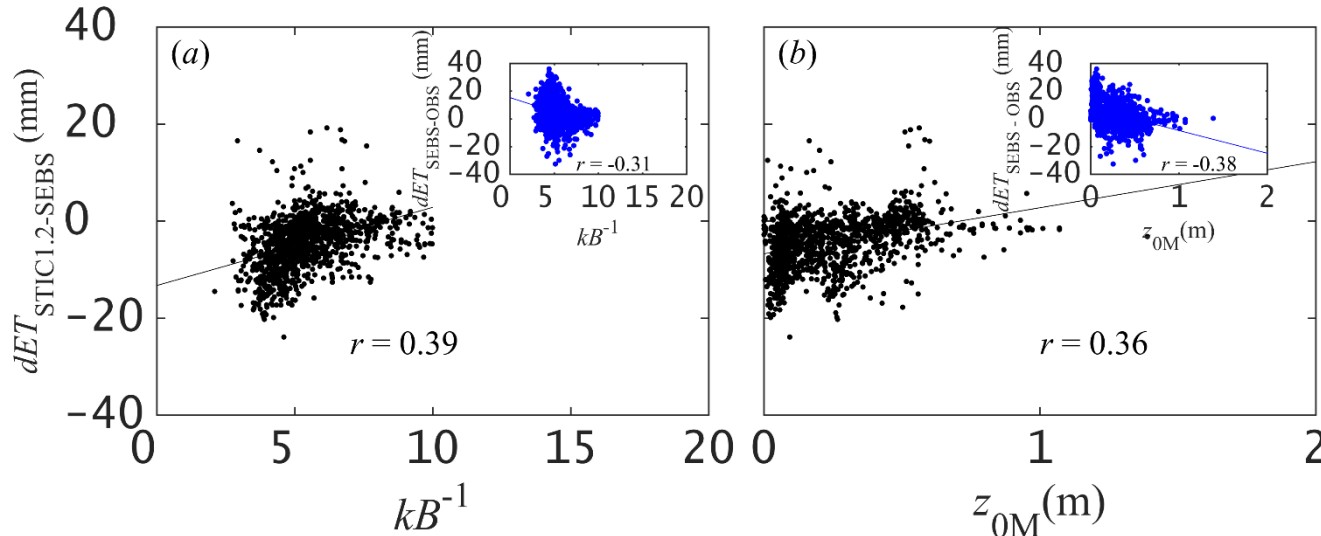

**Figure 13: Scatterplots of the residual differences in cumulative 8-day ET estimates from STIC1.2 and SEBS and the residual errors from SEBS (versus the observations) against $kB^{-1}$ and $z_{0M}$. Pearson correlation coefficient, $r$ ($p$-value was < 0.005 for all relationships shown above), are also shown in each plot.**



# APPENDIX

**Appendix A:**

A1. Table of symbols and their description used in the study.

| Symbol | Description |
| --- | --- |
| $\lambda$ | Latent heat of vaporization of water (J kg$^{-1}$ K$^{-1}$) |
| $H$ | Sensible heat flux (W m$^{-2}$) |
| $R_N$ | Net radiation (W m$^{-2}$) |
| $R_S$ | Shortwave radiation (W m$^{-2}$) |
| $R_{ld}$ | Incoming longwave radiation (W m$^{-2}$) |
| $R_{lu}$ | Outgoing longwave radiation (W m$^{-2}$) |
| $G$ | Ground heat flux (W m$^{-2}$) |
| $\phi$ | Available energy (W m$^{-2}$) |
| ET | Evapotranspiration (evaporation + transpiration) as depth of water (mm) |
| $\lambda E$ | Latent heat flux (W m$^{-2}$) |
| $E_p$ | Potential evaporation as depth of water (mm) |
| $g_A$ | Aerodynamic conductance (m s$^{-1}$) |
| $g_C$ | Canopy (or surface) conductance (m s$^{-1}$) |
| $r_A$ | Aerodynamic resistance (s m$^{-1}$) |
| $r_C$ | Canopy (or surface) resistance (s m$^{-1}$) |
| $M$ | Aggregated surface moisture availability (0–1) |
| $T_A$ | Air temperature (°C) |
| $T_D$ | Dewpoint temperature of the air (°C) |
| $T_R$ | Radiometric surface temperature (°C) |
| $T_{SD}$ | Dew point temperature at the source/sink height (°C) |
| $T_0$ | Aerodynamic surface temperature (°C) |
| RH | Relative humidity (%) |
| $e_A$ | Atmospheric vapour pressure (hPa) at the level of $T_A$ measurement |
| $D_A$ | Atmospheric vapour pressure deficit (hPa) at the level of $T_A$ measurement |
| $e_S$ | Vapour pressure at the surface (hPa) |
| $e_S^*$ | Saturation vapour pressure at surface (hPa) |





| $e_0^*$ | Saturation vapour pressure at the source/sink height (hPa) |
|---|---|
| $e_0$ | Saturation vapour pressure at the source/sink height (hPa) |
| $s$ | Slope of saturation vapour pressure versus temperature curve (hPa K$^{-1}$) |
| $s_1$ | Slope of saturation vapour pressure and temperature between ($T_{SD} - T_D$) versus ($e_0 - e_A$), approximated at $T_D$ (hPa K$^{-1}$) |
| $s_2$ | Slope of saturation vapour pressure and temperature between ($T_R - T_D$) versus ($e_S^* - e_A$), estimated according to Mallick et al. (2015) (hPa K$^{-1}$) |
| $\gamma$ | Psychrometric constant (hPa K$^{-1}$) |
| $\rho_A$ | Density of air (kg m$^{-3}$) |
| $c_p$ | Specific heat of dry air (MJ kg$^{-1}$ K$^{-1}$) |
| $\Lambda$ | Evaporative fraction |
| $\Lambda_R$ | Relative evaporation (-) |
| $\theta$ | Surface (0–5 cm) soil moisture (m$^3$ m$^{-3}$) |
| LAI | Leaf area index (m$^2$ m$^{-2}$) |
| NDVI | Normalized difference vegetation Index (-) |
| $\beta$ | Bowen ratio (-) |
| $\theta v$ | Virtual potential temperature near the surface (K) |
| $\varepsilon_o$ | Surface emissivity (-) |
| $\alpha_o$ | Surface albedo (-) |
| $u_*$ | Friction velocity (m s$^{-1}$) |
| $R_{N24}$ | Daily net radiation (W m$^{-2}$) |
| $R_{N24,8\text{-day}}$ | 8-day net radiation (W m$^{-2}$) |
| $kB^{-1}$ | Excess resistance to the heat transfer parameter (-) |
| $\lambda E_{wet}$ | $\lambda E$ at wet limits (W m$^{-2}$) |
| $H_{wet}$ | $H$ at wet limits (W m$^{-2}$) |
| $H_{dry}$ | $H$ at dry limits (W m$^{-2}$) |
| $L$ | Monin–Obukhov length (m) |
| $g$ | Acceleration due to gravity (9.8 m s$^{-2}$) |
| $d_0$ | Zero plane displacement height (m) |
| $\Psi_H$ | Atmospheric stability correction for heat transport (-) |
| $\Psi_M$ | Atmospheric stability correction for momentum transfer (-) |
| $z_{0M}$ | Roughness length for momentum transfer (m) |





| $z_{0H}$ | Roughness length for heat transfer (m) |
| $z$ | Reference height (m) |
| $z_b$ | Blending Height (m) |
| k | von Karman constant (-) |

## A2. Derivation of 'state equations' in STIC 1.2

After neglecting the horizontal advection and energy storage, the surface energy balance equation is written as:

$$\phi = \lambda E + H \tag{A1}$$

While $H$ is controlled by a single aerodynamic resistance ($r_A$) (or $1/g_A$); $\lambda E$ is controlled by two resistances in series, the canopy (or surface) resistance ($r_C$) (or $1/g_C$) and the aerodynamic resistance to vapour transfer ($r_C + r_A$). For simplicity, it is

5   implicitly assumed that the aerodynamic resistance of water vapour and heat are equal (Raupach, 1998), and both the fluxes are transported from the same level from near surface to the atmosphere. The sensible and latent heat flux can be expressed in the form of aerodynamic transfer equations (Boegh et al., 2002;Boegh and Soegaard, 2004) as follows:

$$H = \rho_A c_P g_A (T_0 - T_A) \tag{A2}$$

$$\lambda E = \frac{\rho_A c_P}{\gamma} g_A (e_0 - e_A) = \frac{\rho_A c_P}{\gamma} g_C (e_0^* - e_0) \tag{A3}$$

Where $T_0$ and $e_0$ are the air temperature and vapour pressure at the source/sink height (i.e., $T_0$ and vapour pressure) and represent the vapour pressure and temperature of the quasi-laminar boundary layer in the immediate vicinity of the surface

10   level. $T_0$ can be obtained by extrapolating the logarithmic profile of $T_A$ down to $z_{0H}$.

By combining Eqs. (A1), (A2), and (A3) and solving for $g_A$, we get the following equation.

$$g_A = \frac{\phi}{\rho_A c_P \left[ (T_0 - T_A) + \left( \frac{e_0 - e_A}{\gamma} \right) \right]} \tag{A4}$$

Combining the aerodynamic expressions of $\lambda E$ in Eq. (A3) and solving for $g_C$, we can express $g_C$ as a function of $g_A$ and vapour pressure gradients.

$$g_C = g_A \frac{(e_0 - e_A)}{(e_0^* - e_0)} \tag{A5}$$

In Eqs. (A4) and (A5), two more unknown variables ($e_0$ and $T_0$) are introduced resulting into two equations and four

15   unknowns. Hence, two more equations are needed to close the system of equations. An expression for $T_0$ is derived from the Bowen ratio ($\beta$) (Bowen, 1926) and evaporative fraction ($\Lambda$) (Shuttleworth et al., 1989) equation as:

$$\beta = \left( \frac{1 - \Lambda}{\Lambda} \right) = \frac{\gamma (T_0 - T_A)}{(e_0 - e_A)} \tag{A6}$$

$$T_0 = T_A + \left( \frac{e_0 - e_A}{\gamma} \right) \left( \frac{1 - \Lambda}{\Lambda} \right) \tag{A7}$$





The expression for $T_0$ introduces another new variable ($\Lambda$); therefore, one more equation that describes the dependence of $\Lambda$ on the conductances ($g_A$ and $g_C$) is needed to close the system of equations. In order to express $\Lambda$ in terms of $g_A$ and $g_C$, STIC1.2 adopts the advection – aridity (AA) hypothesis (Brutsaert and Stricker, 1979) with a modification introduced by Mallick et al. (2015). The AA hypothesis is based on a complementary connection between the potential evaporation ($E_P$),

sensible heat flux ($H$), and ET; and leads to an assumed link between $g_A$ and $T_0$. However, the effects of surface moisture (or water stress) were not explicit in the AA equation and Mallick et al. (2015) implemented a moisture constraint in the original advection-aridity hypothesis while deriving a 'state equation' of $\Lambda$ (Eq. A8). A detailed derivation of the 'state equation' for $\Lambda$ is described in Mallick et al. (2014, 2015, and 2016).

$$\Lambda = \frac{2\alpha s}{2s + 2\gamma + \gamma \frac{g_A}{g_C}(1+M)} \tag{A8}$$

### A3. Estimating $e_0$, $e_0^*$, $M$, and $\alpha$ in STIC 1.2

In the early versions of STIC (Mallick et al., 2014; Mallick et al., 2015), no distinction was made between the surface and source/sink height vapour pressures and hence $e_0^*$ was approximated as the saturation vapour pressure at $T_R$. $e_0$ was estimated from $M$ with an assumption that the vapour pressure at the source/sink height scales between extreme wet–dry surface conditions. However, the level of $e_0^*$ and $e_0$ should be consistent with the level of $T_0$ from which the sensible heat flux is transferred (Lhomme and Montes, 2014). To use the PM equation predictively, it is imperative to consider the feedback

between the surface layer evaporative fluxes and source/sink height mixing and coupling (McNaughton and Jarvis, 1984). Therefore, STIC1.2 uses physical expressions for estimating $e_0^*$ and $e_0$ followed by estimating $T_{SD}$ and $M$ as described below.

An estimate of $e_0^*$ is obtained by inverting the aerodynamic transfer equation of $\lambda E$.

$$e_0^* = e_A + \left[\frac{\gamma \lambda E (g_A + g_C)}{\rho_A c_P g_A g_C}\right] \tag{A9}$$

Following Shuttleworth and Wallace (1985) (SW), the vapour pressure deficit ($D_0$) ($=e_0^* - e_0$) and $e_0$ at the source/sink

height are expressed as follows.

$$D_0 = D_A + \left[\frac{\{s\phi - (s+\gamma)\lambda E\}}{\rho_A c_P g_A}\right] \tag{A10}$$

$$e_0 = e_0^* - D_0 \tag{A11}$$

A physical equation of $\alpha$ is derived by expressing $\Lambda$ as function of the aerodynamic equations $H$ and $\lambda E$.

$$\Lambda = \frac{\lambda E}{H + \lambda E} \tag{A12}$$

$$\Lambda = \frac{\frac{\rho_A c_P}{\gamma} \frac{g_A g_C}{g_A + g_C}(e_0^* - e_A)}{\rho_A c_P g_A (T_0 - T_A) + \frac{\rho_A c_P}{\gamma} \frac{g_A g_C}{g_A + g_C}(e_0^* - e_A)} \tag{A13}$$




$$\Lambda = \frac{g_C(e_0^* - e_A)}{[\gamma(T_0 - T_A)(g_A + g_C) + g_C(e_0^* - e_A)]} \tag{A14}$$

Combining Eqs. (A14) and (A8) (eliminating $\Lambda$), $\alpha$ can be expressed as:

$$\alpha = \frac{g_C(e_0^* - e_A)\left[2s + 2\gamma + \gamma \frac{g_A}{g_C}(1 + M)\right]}{2s[\gamma(T_0 - T_A)(g_A + g_C) + g_C(e_0^* - e_A)]} \tag{A15}$$

Following Venturini et al. (2008), and the theory of psychrometric slope of saturation vapour pressure versus temperatures, $M$ is expressed as the ratio of the dewpoint temperature difference between the source/sink height and air to the temperature difference between $T_R$ and dewpoint temperature of the air ($T_D$).

$$M = \frac{s_1(T_{SD} - T_D)}{\kappa s_2(T_R - T_D)} \tag{A16}$$

Where $T_{SD}$ is the dewpoint temperature at the source/sink height; $s_1$ and $s_2$ are the psychrometric slopes of the saturation vapour pressure and temperature between $(T_{SD} - T_D)$ versus $(e_0 - e_A)$ and $(T_R - T_D)$ versus $(e_S^* - e_A)$ relationship (Venturini et al., 2008); and $\kappa$ is the ratio between $(e_0^* - e_A)$ and $(e_S^* - e_A)$. Despite $T_0$ drives the sensible heat flux, the comprehensive dry-wet signature of underlying surface due to soil moisture variations is directly reflected in $T_R$ (Kustas and Anderson, 2009). Therefore, using $T_R$ in the denominator of Eq. (A16) tends to give a direct signature of the surface moisture availability ($M$).

In Eq. (A16), both $s_1$ and $T_{SD}$ are unknowns, and an initial estimate of $T_{SD}$ is obtained using Eq. (6) of Venturini et al. (2008) where $s_1$ was approximated in $T_D$. From the initial estimates of $T_{SD}$, an initial estimate of $M$ is obtained as $M = s_1(T_{SD} - T_D)/s_2(T_R - T_D)$. However, since $T_{SD}$ also depends on $\lambda E$, an iterative updating of $T_{SD}$ (and $M$) is carried out by expressing $T_{SD}$ as a function of $\lambda E$ as described below (also in Mallick et al., 2016). By decomposing the aerodynamic equation of $\lambda E$, $T_{SD}$ can be expressed as follows.

$$\lambda E = \frac{\rho_A c_P}{\gamma} g_A(e_0 - e_A) = \frac{\rho_A c_P}{\gamma} g_A s_1(T_{SD} - T_D) \tag{A17}$$

$$T_{SD} = T_D + \frac{\gamma \lambda E}{\rho_A c_P g_A s_1} \tag{A18}$$

An initial value of $\alpha$ is assigned as 1.26 and initial estimates of $e_0^*$ and $e_0$ are obtained from $T_R$ and $M$ as $e_0^* = 6.13753 e^{\frac{17.27 T_R}{(T_R + 237.3)}}$ and $e_0 = e_A + M(e_0^* - e_A)$. Initial $T_{SD}$ and $M$ were estimated from Eq. (6) of Venturini et al. (2008) and Eq. (A16), respectively. With the initial estimates of these variables; initial estimate of the conductances, $T_0$, $\Lambda$, and $\lambda E$ are obtained. This process is then iterated by updating $e_0^*$ (using Eq. A9), $D_0$ (using Eq. A10), $e_0$ (using Eq. A11), $T_{SD}$ (using Eq. A18 with $s_1$ estimated at $T_D$), $M$ (using Eq. A16), and $\alpha$ (using Eq. A15), with the initial estimates of $g_C$, $g_A$, and $\lambda E$, and

recomputing $g_C$, $g_A$, $T_0$, $\Lambda$, and $\lambda E$ in the subsequent iterations with the previous estimates of $e_0^*$, $e_0$, $T_{SD}$, $M$, and $\alpha$ until the convergence $\lambda E$ is achieved. Stable values of $\lambda E$, $e_0^*$, $e_0$, $T_{SD}$, $M$, and $\alpha$ are obtained within ~25 iterations.