# Peer review of "Regional evapotranspiration from image-based implementation of the Surface Temperature Initiated Closure (STIC1.2) model and its validation across an aridity gradient in the conterminous United States"

_Hydrology and Earth System Sciences, 2017_

## Referee Comment (RC1) · Anonymous Referee #1 · 29 Nov 2017

This paper has enlightened ET remote sensing community about the importance of aerodynamic conductance/resistance. Currently, most of the ET algorithms do not take into account the diurnal variation in this resistance. The authors have implemented STIC model at regional scale. STIC model integrates remote sensed surface temperature into Penman-Monteith equation to derive an analytical solution for the resistance and use the resolved resistance to calculate surface heat fluxes/ET. They also compare its performance with other two ET algorithms, SEBS and MOD16. SEBS model

provide direct solution for surface and boundary layer conductance/resistance from momentum/heat roughness and stability. However, MOD16 uses a kind of constant resistance in its ET calculation, which explains its worst performance among the three methods. STIC use an energy balance and meteorological information to inversely retrieve the surface and boundary layer conductance. The results are sufficient to support their conclusions. The paper address very relevant scientific questions within the scope of HESS. Thus I suggest an acceptance for publication.

Figure 5 shows that SEBS has a similar performance as STIC, and MOD16 for CRO, DBF, and ENF, but worse result at WSA and GRA. Please check if this is due to inaccuracy of satellite input data.

Fig. 8, 9 10 shows that SEBS ET maps have higher ET than STIC and MOD16, this is due to sensible heat flux is low-estimated, because of high kB_1. Please check the reference of Chen et al. 2013.

Which method or model is used to calculate kB_1 and z0m in figure 13? Or kB_1 and z0m is derived from flux tower measurement?

Figure. 12, please have more discussion about the higher SEBS annual ET, is this due to the method in annual accumulation or SEBS model. Fig. 4 and 5 does not show SEBS ET has different performance over different land covers, at least does not always show high ET estimation.

Special report to the authors: Sorry for the late report due to other heavy dateline.

---

## Author Comment (AC1) · 3 Jan 2018

Response to Referee's (#1) comments

Comment: This paper has enlightened ET remote sensing community about the importance of aerodynamic conductance/resistance. Currently, most of the ET algorithms do not take into account the diurnal variation in this resistance. The authors have implemented STIC model at regional scale. STIC model integrates remote sensed surface

temperature into Penman-Monteith equation to derive an analytical solution for the resistance and use the resolved resistance to calculate surface heat fluxes/ET. They also compare its performance with other two ET algorithms, SEBS and MOD16. SEBS model provide direct solution for surface and boundary layer conductance/resistance from momentum/heat roughness and stability. However, MOD16 uses a kind of constant resistance in its ET calculation, which explains its worst performance among the three methods. STIC use an energy balance and meteorological information to inversely retrieve the surface and boundary layer conductance. The results are sufficient to support their conclusions. The paper address very relevant scientific questions within the scope of HESS. Thus I suggest an acceptance for publication.

Response: We thank the reviewer for appreciating our work and considering the manuscript to be interesting and relevant to the HESS community.
* * *
Comment: Figure 5 shows that SEBS has a similar performance as STIC, and MOD16 for CRO, DBF, and ENF, but worse result at WSA and GRA. Please check if this is due to inaccuracy of satellite input data.

Response: Fig. 5 suggests significant overestimation of SEBS ET in WSA and GRA sites. Specifically, at one of the WSA sites (e.g., US-Ton), all of the three models performed poorly, where observed ET values were extremely low (0.6 mm per day; Fig. 5). In this specific site, 10% of the observed LST pixels were within 2-3 K LST errors with surface emissivity errors of 0.01 to 0.03, based on the MODIS QA/QC data. Similarly, in another WSA site (i.e. US-SRM), 37% of observed LST pixels were within a similar error range. The literature also suggests high emissivity correction uncertainties and systematic underestimation of MODIS LST in arid and semiarid ecosystems (Wan and Li, 2008; Jin and Liang, 2006; Hulley et al., 2012). The significantly poor performance of SEBS in the US-SRM site could also be attributed to these uncertainties. We have added text in the discussion section about the performance of STIC1.2 at the US-Ton site (Page 15, Lines 13). It is also important to note that uncertainties also exist in

the Bowen ratio energy balance closure correction of energy balance at the arid and semi-arid sites, which is also discussed on Page 15, Lines 16-20. Regarding the poor performance of SEBS in the GRA sites, the errors are mostly due to large differences in SEBS ET and observed ET in the two semi-arid desert grasslands sites (US-SRG and US-Wkg) (Supplementary Table S1). In the two other grassland sites (US-Kon and US-KFS), SEBS performed relatively better and was comparable with STIC1.2 (Supplementary Table S1). In those two semi-arid desert grasslands sites (US-SRG and US-Wkg), the MODIS QC/QA bin suggested that 27% (US-Wkg) and 4% (US-SRG) of MODIS LST were within 2-3 K errors with emissivity errors within the 0.01 to 0.02 range, based on the MODIS QA/QC data. These errors are however more predominant in SEBS, and, as in the semi-arid and arid conditions, substantial differences exist between TR and T0 due to strong soil water limitations. Such water-limited conditions may not have been properly characterized in the kB-1 parameterization, which could lead to large differences between modeled and observed ET. We discussed these potential limitations on Page 15, Line 27 to Page 16, Line 10. We will add additional discussion of the performances of SEBS and other models in GRA and WSA sites and how the model performed differently in two GSA sites in two different climates in the revised paper as following in Page 15 after Line 26:

"The overall performance metrics from the three models may be slightly biased due to their strikingly poor performances at some specific sites (Table S1). For example, SEBS overestimated ET by over 64% in the two semi-arid WSA (US-Ton, US-SRM) and GRA (US-SRG and US-Wkg) sites (Table S1); however, its performance in US-Ne1 (CRO), US-Kon (GRA), US-KFS (GRA), US-NR1 (ENF) were better or comparable than the other two models. Interestingly, the performance of SEBS in the two wet GRA sites (i.e. US-Kon and US-KFS) was found to be significantly better compared to its performance at the two semi-arid GRA sites. This could be due to the inability of the kB-1 parameterization scheme in SEBS to account for the substantial differences between TR and T0 due to strong soil water limitations. MOD16 underestimated ET from all but three sites (US-Ton, US-MMS, US-NC1) and underestimated mean ET by over 50% in

US-Ne1 (CRO), US-SRM (WSA), US-SRG (GRA), and US-Wkg (GRA) sites. STIC1.2 appears to be most consistent among the three models, as the mean bias errors were within 20% for all but three sites (US-Ton, US-Kon, US-Ne1)."
* * *
Comment: Fig. 8, 9, 10 shows that SEBS ET maps have higher ET than STIC and MOD16, this is due to sensible heat flux is low-estimated, because of high kB_1. Please check the reference of Chen et al. 2013.

Response: This is a correct statement. The underestimation of sensible heat flux (H) by SEBS (nearly 41% underestimation) was mostly seen in arid and semi-arid sites, which eventually led to overestimation of ET in these sites. Chen et al. (2013) provided an extensive discussion on how H is underestimated by the original SEBS model and proposed an improved way of estimating roughness length for heat transfer through a new parametrization of kB-1adopted from Yang et al. (2002) for bare soil and snow surfaces. In this paper, we incorporated the same kB-1 formulation from Yang et al. (2002) (source code obtained from Abouali et al., 2013); however, the new kB-1 formulation needs substantial modification in arid and semi-arid conditions. Fig. 13 suggests that ET biases (SEBS ET-Observed ET) were typically random and large when kB-1 values were within 5 (r = -0.03) and slightly negative when kB-1 values were within 5 and 8 (r = -0.16). However, for kB-1 values > 8, a linear trend in ET bias was evident (underestimation of H) with an increase in kB-1 (r = 0.24). We will add some discussions on how uncertainties in kB-1 parametrization could lead to biases in estimated fluxes in the revised version (Page 16, after Line 11) as:

"Overestimation of SEBS is mostly associated with the underestimation of sensible heat flux (H) in the arid and semi-arid sites (nearly 41% underestimation in this study). Such underestimation of H by SEBS is highlighted by Chen et al. (2013), who proposed an improved way of estimating roughness length for heat transfer through a new parametrization of kB-1adopted from Yang et al. (2002) for bare soil and snow surfaces. Our study adopted the same kB-1 parametrization. Fig. 13 suggests that ET
biases (SEBS ET-Observed ET) were typically random and large when kB-1 values were within 5 (r = -0.03) and slightly negative when kB-1 values were within 5 and 8 (r = -0.16). However, for kB-1 values > 8, a linear trend in ET bias was evident (i.e. under-estimation of H) with an increase in kB-1 (r = 0.24). Our study suggests that substantial modification in kB-1 parametrization is still needed in arid and semi-arid conditions for improving SEBS accuracies."

References:

Yang, K., et al. "Improvement of surface flux parametrizations with a turbulence‐re-lated length." Quarterly Journal of the Royal Meteorological Society 128.584 (2002): 2073-2087.

Chen, X." et al. "An improvement of roughness height parameterization of the Sur-face Energy Balance System (SEBS) over the Tibetan Plateau." Journal of Applied Meteorology and Climatology 52.3 (2013): 607-622.

Abouali, Mohammad, et al. "A high performance GPU implementation of Surface En-ergy Balance System (SEBS) based on CUDA-C." Environmental modelling & software 41 (2013): 134-138.
* * *
Comment: Which method or model is used to calculate kB_1 and z0m in figure 13? Or kB_1 and z0m is derived from flux tower measurement?

Response: In this paper, the roughness height for heat transfer from Yang et al. (2002) was used to parametrize kB-1 . This approach provides a relatively better estimate of kB-1 as compared to other kB-1 formulations (Su et al., 2001; Su, 2002) over bare soil and low canopies as demonstrated by Chen et al. (2013). z0M was derived using a simple empirical relationship between the roughness length of momentum transfer, z0M, and NDVI, as suggested by Van der Kwast et al. (2009) [Page 9, Lines 23-24]. Most sub-models of SEBS were either adapted or modified from Abouali et al.

(2013). We will add these details with appropriate citations in the revised version of the manuscript (After Page 9, Line 24) as:

"z0M was derived using a simple empirical relationship between the roughness length of momentum transfer, z0M, and NDVI, as suggested by Van der Kwast et al. (2009). The roughness height for heat transfer proposed by Yang et al. (2002), was used to parametrize kB-1. This new parametrization of kB-1 was designed to improve the SEBS model performances on bare soil, low canopies, and snow surfaces as was proposed by Chen et al. (2013)." In Page 10, Line 6, We will add the following sentences to provide information on source codes of SEBS and STIC1.

"The source codes for different sub-models within SEBS were either adapted or modified from Abouali et al. (2013). The STIC1.2 source code was modified from the original STIC1.2 code (Mallick et al. 2016) in Matlab (Mathworks Inc, Natick, USA)."

References:

Su, Z. (2002). The Surface Energy Balance System (SEBS) for estimation of turbulent heat fluxes. Hydrology and earth system sciences, 6(1), 85-100.

Su, Z., Schmugge, T., Kustas, W. P., & Massman, W. J. (2001). An evaluation of two models for estimation of the roughness height for heat transfer between the land surface and the atmosphere. Journal of applied meteorology, 40(11), 1933-1951.

Yang, K., et al. "Improvement of surface flux parametrizations with a turbulence‐related length." Quarterly Journal of the Royal Meteorological Society 128.584 (2002): 2073-2087.

Chen, X., et al. "An improvement of roughness height parameterization of the Surface Energy Balance System (SEBS) over the Tibetan Plateau." Journal of Applied Meteorology and Climatology 52.3 (2013): 607-622.

Van der Kwast, J., et al. "Evaluation of the Surface Energy Balance System (SEBS) applied to ASTER imagery with flux-measurements at the SPARC 2004 site (Barrax,

Spain)." Hydrology and Earth System Sciences Discussions 6.1 (2009): 1165-1196.

Abouali, M., et al. "A high performance GPU implementation of Surface Energy Balance System (SEBS) based on CUDA-C." Environmental modelling & software 41 (2013): 134-138.

Mallick, K., Trebs, I., Boegh, E., Giustarini, L., Schlerf, M., Drewry, D. T., ... & Saleska, S. (2016). Canopy-scale biophysical controls of transpiration and evaporation in the Amazon Basin. Hydrology and Earth System Sciences, 20(10), 4237-4264.
* * *
Comment: Figure. 12, please have more discussion about the higher SEBS annual ET, is this due to the method in annual accumulation or SEBS model.

Response: We have briefly discussed that this overestimation is mostly due to consistent overestimation of 8-day ET by the SEBS model (Page 14, lines 10-16). In addition, the 8-day average net radiation was also overestimated by 9% (Supplementary: Fig. 1), which could also add some positive biases by SEBS (and also STIC). In the two cropland sites (US-ARM and US-Ne1), SEBS annual ET estimates were within 2% of observed annual ET, which is better than the performance of STIC (22% underestimation) and MOD16 (49% underestimation). SEBS mostly overestimated annual ET from the arid and semi-arid sites (47%). We will add these additional discussions in section 4 of the revised manuscript (After Page 14, Line 16) as:

"In addition, the 8-day average net radiation was also overestimated by about 9% (Supplementary: Fig. 1). SEBS overestimation of annual ET was mostly observed in the arid and semi-arid sites (47%). In the two cropland sites (US-ARM and US-Ne1), SEBS annual ET estimates were within 2% of observed annual ET, which is better than the performance of STIC (22% underestimation) and MOD16 (49% underestimation)."
* * *
Comment: Fig. 4 and 5 does not show SEBS ET has different performance over

different land covers, at least does not always show high ET estimation.

Response: Here we disagree. According to Fig. 4, RMSE and MAE of SEBS is similar to MOD16 across different precipitation conditions, but STIC1.2 performed better overall. According to Fig. 5, we noticed that SEBS performed best in CRO sites among all the models and its performance was similar to MOD16 in ENF sites compared to its performances in other biomes (GRA and WSA). We agree with the reviewer that SEBS does not overestimate ET all the time; as seen in Fig. 4 and 5, there are several instances when SEBS ET was lower than the observed ET. However, the overestimation tendency of ET by SEBS was predominant during the dry year (Fig. 4). The term "overestimation" refers to the mean ET observed at the flux sites. Notably, SEBS ET estimates were within 3%, 8%, and 17% of observed ET at croplands, ENF, and DBF sites, respectively, which were comparable or sometimes better than the other two models. We have briefly stated these in section 4 (Page 16, Line 18-20 and Page 16, Lines 30-35, Page 17, Lines 1-4). In page 17, between Line 2 and Line 3, we will add the following sentences:

"It should be noted that the performance of SEBS was not entirely poor. The overestimation tendency of ET by SEBS was predominant during the dry year (Fig. 4 and Fig. S2). Notably, SEBS ET estimates were within 3%, 8%, and 17% of observed ET at croplands, ENF, and DBF sites, respectively, which were comparable or sometimes better than the other two models (Fig. 5 and Table S1). In addition, the performance of SEBS was relatively good in cropland (Fig. 5)."

---

## Referee Comment (RC2) · Anonymous Referee #2 · 4 Jan 2018

This manuscript provides an evaluation of STIC1.2 in estimating actual evapotranspiration at the spatial scale by combining the model with satellite remote sensing data. In addition, the authors also compare the performance of STIC1.2 with other two existing remote sensing algorithm (i.e., SEBS and MOD16). In general, the topic of this MS is of interest to the HESS' readership and the manuscript is well written. However, there are several major issues in this study, which introduce additional uncertainties and preclude a focused evaluation of the models themselves (as described below). In

this light, a MAJOR REVISION is needed. Major: 1. My largest criticism lies in the use of MOD11A2, where LST is reported as the average values of clear-sky LSTs during an 8-day period. As there is no information about which day (or days) out of each 8-days contributes to the final 8-day averages, this 8-day average LST is highly likely to not correspond to the 8-day averages of meteorological variables (i.e., air temperature, VPD, etc). For example, the 8-day LST might only be a result of day-1 LST, or the average of day-1 and day-7 LSTs. Even they correspond well with each other, using 8-day averages may still lead to additional uncertainties due to differences in the temporal variability between, say, daily LST and air temperature. For example, H_day1+H_day2…+H_day8 does not equal to 1/8 * (H_calculated using 8-day average LST and meteorological inputs), as all the responses are non-linear. To focus on evaluating the model itself, it is recommended to work on the instantaneous scale rather than 8-day averages. 2. Page11 (Line 8-17) Again, validation should be carried out at an instantaneous scale but not daily (or 8-day averages), as upscaling can introduce additional uncertainties. In my opinion, upscaling from satellite overpass to longer time scales is another scientific question. 3. Page11 (L29-30): Any explanation of this model performance: overestimation in dry year and underestimation in wet years? Additionally, according to your Figure 6, it seems that this "overestimation in dry and underestimation in wet" pattern persists across sites (i.e., spatially). This may suggest some systematic uncertainty of the model. Given this, I do not agree with the statement given in Page 15 (Line 4-12). First, does any previous study support this wet/dry patches around the studied sites? If not, this is just your speculation. Second, the footprint issue could lead to random errors rather than a systematic underestimation. Finally, it is the authors' responsibility to ensure the footprint of a flux site corresponds (or encompasses) the MODIS footprint so that to eliminate data uncertainties and to allow a focused evaluation of the model. Minor: 4. Page7(L7): Delete "the" between "both" and "variables"; 5. Page9(L20): Please specify the equation for NDVI and/or provide references. 6. Page 16 (Discussion on MOD16): It is worthwhile reading Yang et al. (2016, WRR; doi: 10.1002/2014WR015619) for a more physical explanation on

the MOD16 uncertainty.

---

## Author Comment (AC2) · 23 Jan 2018

**Response to Referee # 2**

*This manuscript provides an evaluation of STIC1.2 in estimating actual evapotranspiration at the spatial scale by combining the model with satellite remote sensing data. In addition, the authors also compare the performance of STIC1.2 with other two existing remote sensing algorithm (i.e., SEBS and MOD16). In general, the topic of this MS is of interest to the HESS' readership and the manuscript is well written. However, there are several major issues in this study, which introduce additional uncertainties and preclude a focused evaluation of the models themselves (as described below). In this light, a MAJOR REVISION is needed.*

We thank Referee #2 (R#2 hereafter) for finding our work interesting to the HESS community and providing useful criticism, comments, and suggestions. We also acknowledge R #2 for pointing out potential uncertainties associated with the use of input data. Uncertainties arising from the temporal and spatial mismatches of input datasets from different sources are common in remote sensing based studies. Acknowledging this fact, we made best possible efforts to minimize those. First, it is important to clarify that our study aims to provide a scientific basis for the operational use of STIC1.2 model towards estimating regional ET using remotely sensed data. Since the MODIS daily products suffer greatly due to cloud cover, the 8-day MODIS products are more applicable for regional ET model implementation and hence the validation was done at this temporal scale, which is a common practice (Ichii et al., 2009;Zhang et al., 2010;Yang et al., 2006;Xiong et al., 2015). The MOD11A2 product was considered to provide a better temporal and spatial representation (compared to the daily MODIS products) of regions (i.e. a wide range of biome and aridity conditions in the US). Nevertheless, we have now implemented STIC1.2 and SEBS at instantaneous (at point scale using daily MODIS and instantaneous weather data) and presented main validation results in this response. However, this implementation is not the core part of our paper and hence will only be added to the supplementary (1 figure and 1 table) and with a brief discussion in the main text (highlighted in red font, Page 6).

In this response, we not only justify our approach (e.g. use of MODIS 8-day LST) with additional analysis and references but also show results from the evaluation of instantaneous ET from SEBS and STIC1.2 models using instantaneous/daily MODIS products (and weather inputs of the same period). Results suggest that the core findings and conclusions of this study will remain the same regardless of whether instantaneous or 8-day products are used. However, additional knowledge on the potential sensitivity of SEBS to input meteorological forcing and the applicability of the STIC1.2 model across different time scales were gained. In the revised version, we propose to add additional descriptions (highlighted in red font, see pages 6 and 12) on potential uncertainties associated with the application of models at 8-day scales, use of 8-day average (daytime) weather inputs, and the MODIS 16 ET products. We believe that the evidence we provided and detailed answers to your concerns with appropriate references will sufficiently address R #2's major concerns that are mostly associated with the use of input data and uncertainties.

*Major: 1. My largest criticism lies in the use of MOD11A2, where LST is reported as the average values of clear-sky LSTs during an 8-day period. As there is no information about*

*which day (or days) out of each 8-days contributes to the final 8-day averages, this 8-day average LST is highly likely to not correspond to the 8-day averages of meteorological variables (i.e., air temperature, VPD, etc). For example, the 8-day LST might only be a result of day-1 LST, or the average of day-1 and day-7 LSTs. Even they correspond well with each other, using 8-day averages may still lead to additional uncertainties due to differences in the temporal variability between, say, daily LST and air temperature. For example, H_day1+H_day2. . .+H_day8 does not equal to 1/8 * (H_calculated using 8-day average LST and meteorological inputs), as all the responses are non-linear. To focus on evaluating the model itself, it is recommended to work on the instantaneous scale rather than 8-day averages.*

Response: The scientific basis for using MOD11A2 comes from an abundance of studies (for e.g. Ichii et al., 2009;Jin et al., 2011;Tian et al., 2013;Garcia et al., 2014;Xiong et al., 2015) that have also used this 8-day product in ET modeling. In our study, SEBS and STIC1.2 models were run at 8-day average scale corresponding to the MODIS daytime overpass time using MOD11A2 (and ancillary MODIS data; Table 2, Page 27) and 8-day average meteorological data corresponding at the MODIS Terra LST overview time (not the entire 8-day average; Page 9, Line 26-Page 10, Line 3 in the MC). The 8-day averaged meteorological data (that considers all hours) are only used in extrapolation of daytime ET to 8-day ET using a constant ET approach (Brutsaert and Sugita, 1992;Crago, 1996;Chávez et al., 2008). We evaluated 8-day cumulative ET to compare model performances against those from available MOD16 8-day ET products (Mu et al., 2011).We provide further justifications for using 8-day MODIS products as below:

**Comparison of 8-day vs. daily LST (or $T_R$), air temperature ($T_A$), and $T_R - T_A$**

We find that the 8-day average LST (or $T_R$), air temperature ($T_A$), and difference between $T_R$ and $T_A$ ($T_R - T_A$) were good representative of the corresponding instantaneous values during each of the 8-days within the corresponding MODIS 8-day period ($R^2$= 0.80-0.92, PBIAS within 3%) (Fig. AR1).

[Figure]

Fig AR1. Scatter plots of 8-day average LST vs. instantaneous LST, 8-day average daytime $T_A$ vs. instantaneous $T_A$, and 8-day average daytime $T_R - T_A$ vs. instantaneous $T_R - T_A$.

**Comparison of 8-day vs. daily ancillary meteorological variables**

When 8-day vs. daily ancillary meteorological variables were compared, solar radiation ($R^2$ = 0.82, PBIAS = -5%) and relative humidity ($R^2$ = 0.78, PBIAS= 6%) were also found to be in a similar range (Table AR1). However, we noted that 8-day average daily wind speed, which is highly variable with time and space, was not well representative of daytime conditions ($R^2$ = 0.36). Note that this uncertainty in wind speed could affect instantaneous ET values from the SEBS model, which uses wind speed to determine sensible heat flux ($H$) and estimate latent heat flux ($\lambda E$) as a residual of surface energy balance. In addition, this would also slightly affect 8-day average net radiation that used the FAO-based Penman-Monteith (PM) equation (Allen et al., 1998;ASCE-EWRI, 2005) that is used to upscale instantaneous ET to 8-day cumulative ET. Hence, we think the uncertainties associated with differences in actual meteorological conditions will have more of an effect on SEBS than STIC1.2, which is also supported by results from model evaluation at the instantaneous scale (i.e., MODIS TERRA daytime overpass time).

Table AR1. Comparison of 8-day average daytime meteorological and radiative inputs vs. instantaneous inputs to assess how representative the 8-day average values were of each day within the 8-day period.

| Variable | $R^2$ | RMSE | MAE | PBIAS (%) |
|---|---|---|---|---|
| $T_R$ (K) | 0.92 | 3.53 | 2.74 | 0.1 |
| $T_A$ (K) | 0.900 | 3.04 | 2.34 | 0 |
| $T_R$ - $T_A$ (K) | 0.80 | 3.16 | 2.42 | 3.8 |
| RH (%) | 0.78 | 10 | 8 | 6 |
| Wind speed (m s$^{-1}$) | 0.36 | 1.61 | 1.19 | 2 |
| Incoming shortwave radiation (W m$^{-2}$) | 0.82 | 69 | 49 | -5% |

**Model implementation and validation at instantaneous scale showed similar results**

STIC1.2 and SEBS were implemented using all the available daily MODIS products (MOD11A1, MOD09GA, etc. and instantaneous weather data from NLDAS-2 forcing data[1]) during the study years (Fig. 3, page 32) and evaluated at the instantaneous scale. We noticed similar overestimation and slight underestimation tendencies of SEBS and STIC1.2, respectively (Table AR2 and Fig. AR2). Overall, model performances at the instantaneous scale during dry, normal, and wet years were also consistent with those observed when validation was done at the 8-day scale (see figure 4, Page 33 in the manuscript); there was a slight underestimation tendency of STIC1.2 and overestimation of SEBS (better under wetter conditions). The additional 11% overestimation of SEBS (17% vs. 28%) could be attributed to uncertainties associated with widely varied wind speed and other meteorological variables within the 8 days of a given 8-day period as well as the slightly overestimated 8-day average net radiation (Fig.AR3, also in the supplementary, Figure S1). The uncertainty associated with wind speed should not affect STIC1.2, which does not use wind speed; therefore, STIC1.2 performances were more consistent compared to SEBS. These results suggest that regardless of whether 8-day or daily MODIS products are used, the key findings of our research would largely remain the same.
* * *
[1] Note: This dataset is 12.5° (~12.5 km) dataset (https://ldas.gsfc.nasa.gov/nldas/NLDAS2forcing.php). The typo in Table2 (page 27) and Page 26, Line 10, where it was written as 4 km will be corrected in the revised version.

Table AR2.Evaluation of instantaneous ET and 8-day cumulative ET (Table 3, Page 28) from STIC1.2 and SEBS against observed ET from thirteen core AmeriFlux sites in the US combining data from one dry, one wet, and one normal year. Note: the 8-day ET estimates are derived from 8-day MODIS products (MOD11A2, MOD09A2, etc.) and 8-day average weather data (both at the satellite overview time and the 8-day average values; Table AR2).

| Scales | Model | $R^2$ | RMSE (mm hr$^{-1}$ or mm 8-day$^{-1}$) | MAE (mm) (mm hr$^{-1}$ or mm 8-day$^{-1}$) | PBIAS (%) |
|---|---|---|---|---|---|
| Instantaneous | STIC1.2 | 0.61 | 0.12 | 0.09 | -5 |
| | SEBS | 0.53 | 0.14 | 0.10 | 17 |
| 8-day | STIC1.2 | 0.66 | 7.5 | 5.4 | -3 |
| | SEBS | 0.53 | 9.8 | 7.3 | 28 |

[Figure]

Fig. AR2. Evaluation of ET estimates from the STIC and SEBS models against flux data at the instantaneous scale using daily MODIS products (MOD11A1) during the dry, normal and wet years (see Fig. 4, Page 33).

**Good correlation between estimated and observed 8-day average available energy ($\phi$)**

The 8-day mean (considering all hours) available energy ($\phi = R_\text{n}$-G) that was used to upscale 8-day average daytime instantaneous ET to 8-day total ET (Eq. 17, page 10) was found to be strongly correlated with the observed 8-day mean $\phi$ at the flux towers. While the random noises in instantaneous $\phi$ (Figure AR3), which was also found to be well correlated with observed values ($R^2 = 0.65$), are removed in the 8-day average $\phi$, a small positive bias (i.e. overestimation of 9%) (Fig. AR3, also in the supplementary, Figure S1) was also added. The residual difference in estimated 8-day average $\phi$ and observed 8-day average $\phi$ was positively correlated with the residual difference in estimated and observed 8-day cumulative ET ($r = 0.27$ for SEBS and $r = 0.18$ for STIC1.2, $p$-value $< 0.001$). This positive bias in estimated 8-day mean $\phi$ could have

reduced biases from STIC1.2 from -5% to -3% (positive shift) and led to further increases in SEBS biases (i.e. 28%, Table AR2).

[Figure]

Fig. AR3. Scatter plot of estimated vs. observed available energy ($\phi$) at instantaneous (using daily MODIS products) and 8-day averaged conditions (using 8-day products).

**Models performed better in areas that are mostly affected by clouds**

In the humid regions, all three models performed better than in the arid and semi-arid regions with similar accuracies despite high cloud cover (resulting in a fewer number of cloud-free days in each MODIS 8-day cycle). This is partly because vegetation (forests) is mostly energy-limited in this region and because estimated average 8-day $\phi$ from the meteorological inputs from the NLDAS-2 forcing (12.5°) were well correlated with observations ($R^2$=0.91) and with 9% error. In the arid sites, the estimated and observed $\phi$ relationship was also similar ($R^2 = 0.96$ and PBIAS within 6%). In our opinion, the effect of cloud cover is smaller in the arid and semi-arid regions compared to the humid and sub-humid regions and hence most of the differences in model performances could be attributed to the physical differences among the models.

**MOD11A2 and aggregated meteorological are commonly used in ET modeling**

The model implementation and validation scheme used in this study (i.e. use of MODIS aggregated datasets and $n$-day averaged meteorological inputs) have been applied in several other studies (Ichii et al., 2009;Senay et al., 2013;Wu et al., 2010;Xiong et al., 2016). In addition, comparison of daily ET, 8-day average ET or the 8-day with respective flux ET has become a common practice in ET model evaluation studies (Yang et al., 2006;Ichii et al., 2009;Senay et al., 2013;Biggs et al., 2016;Xiong et al., 2015;Ryu et al., 2012) and particularly those that use MODIS 8-day datasets.

**Proposed changes in the revised manuscript**

While we have provided evidence and justifications for using MOD11A2 and aggregated weather information, we do acknowledge that there are uncertainties associated with the use of 8-day average LST and aggregated meteorological variables which should be mentioned in the

manuscript. We propose to add the following text (in section 4, after page 17, Line 5) in the revised manuscript and provide tables and figures in the supplementary (Table AR1-AR2 and Figs. AR1-AR3).

"One of the key sources of uncertainty in the implementation of STIC1.2 and SEBS at the 8-day timescale (using MOD11A2) could be the use of average 8-day daily time meteorological inputs that may not well correspond with LST observation days within each MODIS 8-day cycle. We found all 8-day daytime averaged meteorological variables (those used in SEBS or STIC1.2 models) except wind speed to be well representative of instantaneous measurements within the 8-day period (Supplementary Table S2[2]). This could be a source of additional uncertainty in SEBS since it uses wind speed to parameterize the aerodynamic conductance using MOST theory. Model implementation at an instantaneous scale (i.e. MODIS overpass time and using daily MODIS products including MOD11A1 datasets) showed that the performance of STIC1.2 ($R^2$ = 0.61, PBIAS = -5%) was similar to its performance at the 8-day scale. However, for SEBS ($R^2$ = 0.53) the performance was slightly better with a PBIAS of 17% (Table S3[3]). In addition to the wind speed, the slight overestimation of 8-day average $\phi$ (PBIAS = 9%), and variations in $T_R$, $T_A$, $T_R$ -$T_A$, and other meteorological variables during days within the corresponding 8-day period could have added positive biases to SEBS (increase from 17% to 28%), when evaluated at the 8-day scale. SEBS is sensitive to the meteorological input especially the temperature gradient and its performance is expected to degrade with the use of gridded forcing data (Ershadi et al., 2013;McCabe et al., 2016;van der Kwast et al., 2009). Conversely, the overestimation in $\phi$ could have slightly reduced STIC biases (increase from -5% to -3%). Overall, the application of STIC1.2 and SEBS at the instantaneous scale showed similar model predictive capability and potential model strengths and weaknesses (e.g. better under wetter conditions). STIC1.2 appears to be slightly more consistent through time, which could be because the PM equation is designed to be applied at different time scales and STIC1.2 does not rely on wind speed to solve for $G_A$ and $G_C$. Results also suggest that biases from SEBS could be kept well under 20% if uncertainties associated with meteorological and radiative forcing are reduced."

***2. Page11 (Line 8-17) Again, validation should be carried out at an instantaneous scale but not daily (or 8-day averages), as upscaling can introduce additional uncertainties. In my opinion, upscaling from satellite overpass to longer time scales is another scientific question.***

Response: For the most part, please refer to our response to the previous comment. We have added few more points below.

**Should an ET model always be validated at the instantaneous scale only?**

While we agree that the validation of ET at the instantaneous scale could help reduce uncertainties associated with upscaling and overall meteorological representation, we also disagree that the validation should always be conducted this way. ET is a hydrological process and like precipitation and runoff, these processes (and the errors) are perceived better when reported at daily or seasonal scales (e.g. 0.01 mm hr$^{-1}$vs 1 mm/day or 1000 mm year$^{-1}$). For
* * *
[2] Table AR1
[3] Table AR2

example, ET at daily or seasonal scales is more meaningful for hydrologists who manage water resources (Tang et al., 2015;Cammalleri et al., 2014;Colaizzi et al., 2006) and for comparing with accumulated precipitation (Baldocchi and Ryu, 2011). There are host of studies (for e.g.Fisher et al., 2008;Senay et al., 2013;Velpuri et al., 2013;Jiang and Ryu, 2016;Bunting et al., 2014) that have conducted ET validation at much larger temporal scales (e.g., 8-day, monthly, seasonal, annual) than the instantaneous scale.

The 8-day cumulative or mean ET corresponds with the cycle of MODIS global coverage (Masuoka et al., 1998) and one of the most widely used forms of ET model implementation and validation (Ryu et al., 2012). The 8-day or other multiday MODIS composites are designed to deal with cloud cover to provide a more routine measurement of Earth's surface than the daily MODIS data. The cloud effect on ET estimation or understanding other physical processes is greatly reduced when a composite 8-day LST product is used (Yang et al., 2013;Hu and Brunsell, 2013).

**STIC1.2 has already been validated at an instantaneous scale**

STIC1.2 model has been extensively validated at a half-hourly scale using flux tower data (Mallick et al., 2014;Mallick et al., 2015;Mallick et al., 2016), which is a better evaluation than using remotely sensed data, as any bias associated with the spatial and temporal mismatches between input meteorological and land surface variables are distinctly identified. The strength of the present study is to test the ET mapping potential of STIC1.2 at the regional scale using purely remote sensed data and gridded climate data. Therefore, we performed validation of the STIC1.2 model at the 8-day scale and compared our resulting ET estimates with readily available products such as MOD16 or the widely used SEBS model. In addition, our results are consistent with instantaneous scale validation (Table AR2 and Fig. AR2 in this response and Table 3, page 28 and Fig. 4, page 33 in original MS).

**Upscaling errors (instantaneous to 8-day ET) are minimized through reliable estimates of 8-day average $\phi$**

We agree that the upscaling of ET from satellite overpass time to longer time scales (e.g. 8-day, as done in this study) is a different scientific question, as there are several uncertainties associated with it. The approach (Page 10, Equation 17) we used in this paper is a well-established method (Allen et al., 2007;Colaizzi et al., 2006;Ryu et al., 2012;Shuttleworth et al., 1989;Gentine et al., 2007;Chávez et al., 2008). In addition, the estimated 8-day $\phi$ (the key driving force of ET) was strongly correlated with the 8-day average $\phi$ at flux towers ($R^2$ =0.89, RMSE = 20 W m$^{-2}$, PBIAS = 9%, Fig. AR3, also in the supplementary, Figure S1). Hence, the errors associated with model evaluation at the 8-day scale by upscaling of instantaneous to 8-day cumulative ET should be within 9%. If we had directly used the observed $\phi$ at the towers, PBIAS from SEBS would have reduced to 18% (from 28%) and STIC biases would have been -10% (from -3%) with increased $R^2$ (STIC = 0.69 and SEBS= 0.58) and slightly reduced RMSEs (STIC= 6.8 mm and SEBS = 8.6 mm) from both models. However, it should be noted that the evaporative fraction was derived during the image time obtained using the weather information from gridded data, not the flux tower data itself. Hence there are some uncertainties with the use

of meteorological data from multiple sources during instantaneous and multiple scales, as data from the same source its typically used to extrapolate instantaneous to daily or other scales (Allen et al., 2007;Chávez et al., 2008;Allen et al., 1998). Overall, we find that the upscaling errors are within 10% for both models.

*Major Point 3. Page11 (L29-30): Any explanation of this model performance: overestimation in dry year and underestimation in wet years? Additionally, according to your Figure 6, it seems that this "overestimation in dry and underestimation in wet" pattern persists across sites (i.e., spatially). This may suggest some systematic uncertainty of the model. Given this, I do not agree with the statement given in Page 15 (Line 4-12). First, does any previous study support this wet/dry patches around the studied sites? If not, this is just your speculation. Second, the footprint issue could lead to random errors rather than a systematic underestimation. Finally, it is the authors' responsibility to ensure the footprint of a flux site corresponds (or encompasses) the MODIS footprint so that to eliminate data uncertanties and to allow a focused evaluation of the model.*

Response: Overestimation of SEBS could be due to uncertainties associated with the $kB^{-1}$ parameter (Chen et al., 2013), as well as the positive biases in 8-day average $\phi$ (as discussed earlier). Underestimation of ET from STIC1.2 could be due to an excessive moisture constraint applied during initialization of soil moisture availability ($M$) using $T_R$ and dew point temperature at source/sink and reference heights. In addition, in the dry years, overestimation errors of $\phi$ ($R^2$ = 0.88, RMSE = 22 W m$^{-2}$, PBIAS = 12%) was slightly more than in the wet year ($R^2$ = 0.91, RMSE = 18 W m$^{-2}$, PBIAS = 9%). At the instantaneous scale, we noticed that STIC1.2 did not overestimate ET during the dry year and the SEBS overestimation was within 24% (PBIAS = 24%), which could be due to uncertain conductance parameterization, as in case of 8-day evaluation. We have discussed potential uncertainties in detail in section 4 (Figs. 7 and 13). The manuscript is about first ever regional scale implementation of the STIC1.2 model using remotely sensed data and hence have focused more on the initial validation as well as the comparison with two other commonly used models.

Please check figures AR4-AR7, where annual ET maps from three global ET products: 1) The Global Land Evaporation Amsterdam Model (GLEAM; 0.25 ° spatial resolution) (Martens et al., 2017;Miralles et al., 2011); 2) MPI-BGC or Fluxnet: MTE (0.5° spatial resolution) (Jung et al., 2010;Jung et al., 2011)3) SSEBOp (1km spatial resolution; https://earlywarning.usgs.gov/fews/datadownloads) (Senay et al., 2013;Velpuri et al., 2013) are added. Here, the annual ET map from one of three study years (the year when datasets from the other three models were also available) is shown. While the first two datasets (GLEAM and MPI) are at relatively coarser spatial resolution, most of these maps clearly show a similar spatial pattern of ET as STIC1.2. Hence, the spatial patterns of ET produced by our model seem to be reasonable and not linked with any systematic uncertainty of the model. In addition, scatter plots of estimated vs. observed ET (both instantaneous and 8-day cumulative; Figure AR2, Figure 3 and 4) show points spread uniformly around the 1:1 line.

[Figure]

Figure AR4. Annual ET map derived from STIC1.2, SEBS, MOD16, GLEAM, MTE, and SSEBop for the western (W) bounding box covering US-Ton and US-Me2 flux sites (Fig. 2, Page 31).

[Figure]

Figure AR5. Annual ET map derived from STIC1.2, SEBS, MOD16, GLEAM, MTE, and SSEBop for the mid-western 2 (MW2) bounding box covering US-ARM, US-SRG, US-Wkg, and US-NR1 flux sites (Fig. 2, Page 31).

[Figure]

Figure AR6. Annual ET map derived from STIC1.2, SEBS, MOD16, GLEAM, MTE, and SSEBop for mid-western 1 (MW1) bounding box covering US-Kon, US-KFS, US-ARM, US-Ne1, and US-MMS flux sites (Fig. 2, Page 31).

[Figure]

Figure AR7. Annual ET map derived from STIC1.2, SEBS, MOD16, GLEAM, MTE, and SSEBop for the eastern (E) bounding box covering US-NC1 and US-NC2 flux sites (Fig. 2, Page 31).

As demonstrated and discussed in Mallick et al. (2014), although towers are often installed in relatively homogenous terrain, rarely can this be assumed for heterogeneous landscapes in arid and semi-arid environment. The slope of the regression between the observed and estimated λE of individual biome category was significantly related to the average variance of LST surrounding the tower sites (Mallick et al., 2014). The slope of regression varied systematically with the landscape heterogeneity. Similar results are also shown by Stoy et al. (2013), who also found a systematic relationship between the surface energy balance closure, soil moisture variability, and landscape heterogeneity over 173 FLUXNET tower sites.

Currently, there is no consensus on which MODIS footprint size to use to represent fluxes from a flux site and hence any size or method used is subjected to debate. However, most flux sites (other than the arid and some semi-arid sites) used in this study cover vegetated area that is large enough for a $1 \times 1$ km$^2$ MODIS pixel to be represented as a homogenous pixel, which was also verified in Google Earth Engine. Typically, a pixel-to-footprint match is considered adequate if the vegetation and environmental characteristics within the footprint are good representatives of the surrounding area contained by the MODIS pixels (Yuan et al., 2010). The US-Ton site, however, may not be as homogenous as other sites in terms of vegetation type, as the site is dominated by deciduous blue oaks (*Quercusdouglasii sp.*) and the understory and open grassland are mainly cool-season C3 annual species(Ma et al., 2007). This could lead to dry and wet patches of LST, as briefly discussed on Page 15, Line 5-10. Blue oaks and grasses have distinct phenology and MODIS is not sensitive to understory canopy (Ma et al., 2007;Xiao et al., 2010). In addition, the US-Wkg site was classified as either open shrublands or grasslands in different years on MCD12Q1 datasets and was not homogenous beyond a 3×3 neighborhood (i.e. one class was surrounded by pixels of another class).

Nonetheless, the sites considered in our study have been widely used in validation of ET as well as other land surface variables and is currently considered the state of the art observation datasets that can be used as benchmark to assess the performance of the remote sensing based models (Running et al., 2004;Yang et al., 2007;Jung et al., 2010) and several common approaches to extract representative MODIS pixel values include single tower pixel (Yuan et al., 2010;Ryu et al., 2011;Jiang and Ryu, 2016), 3×3 mean value with center pixel as the coordinates of flux towers (Sims et al., 2008;Xiao et al., 2008;Yang et al., 2013), and some foot print analysis (Vinukollu et al., 2011). In this study, we extracted ET values from a single tower pixel located closest to the MODIS pixel, but when a 3×3 mean value of estimated ET with the center pixel as the coordinates of flux towers was considered, only negligible changes in model performances was noticed (Table AR2). In addition, the mean values of 8-day cumulative ET from each model were not significantly different when a single pixel or a 3×3 neighborhood was considered (*p*-value > 0.75). Hence, we think that in this study, footprint uncertainties are minimized by selecting homogenous core AmeriFlux sites and this method is consistent with what has been done in previous studies.

**Table AR2.** Evaluation of 8-day cumulative ET (Table 3, Page 28) from STIC1.2 and SEBS against observed ET from thirteen core AmeriFlux sites in the US combining data from one dry, one wet, and one normal year when pixel values of estimated ET were taken as a mean of 3×3 neighborhood with center

pixel as the coordinates of flux towers. No significant differences in model performance were noticed when a single tower pixel was considered (Table 3, Page 28).

| Model | $R^2$ | RMSE (mm 8-day$^{-1}$) | MAE (mm) (mm 8-day$^{-1}$) | PBIAS (%) |
|---|---|---|---|---|
| STIC1.2 | 0.66 | 7.3 | 5.3 | -4 |
| SEBS | 0.54 | 9.7 | 7.2 | 27 |
| MOD16 | 0.58 | 9 | 6.3 | -27 |

Minor: 4. Page7(L7): Delete "the" between "both" and "variables"

Response: Necessary change will be incorporated.

Minor: 5. Page9(L20): Please specify the equation for NDVI and/or provide references.

Response: We will provide the following reference for NDVI in page 9 line 24-25.

Tucker, C. J. (1979). Red and photographic infrared linear combinations for monitoring vegetation. *Remote sensing of Environment*, *8*(2), 127-150.

*Minor 6. Page 16 (Discussion on MOD16): It is worthwhile reading Yang et al. (2016, WRR; doi: 10.1002/2014WR015619) for a more physical explanation on the MOD16 uncertainty.*

Response: Thanks for referring this paper and will add the following description in the revised manuscript (After Page 16, Line 30)

"Other studies have also found MOD16 to underestimate λE or ET significantly (Yang et al., 2015;Biggs et al., 2016). Yang et al. (2015) highlighted four key uncertainties associated with the MOD16 algorithm (Mu et al., 2011), which could explain the relatively poor performance of MOD16 in this study. First, the dependency of the MOD16 algorithm on meteorological forcing (and not the $T_R$) to account for the soil moisture restriction on evaporation and transpiration processes results in a slower response of variations in energy and heat fluxes (Long and Singh, 2010). Second, underestimation of transpiration in MOD16 could occur due to overestimation of environmental stresses on canopy conductance that is expressed as the potential canopy conductance multiplied by two scaling factors that represent influences from $T_A$ and VPD deficit (Yang et al., 2013). Third, the empirical nature of the soil moisture constraint function(Fisher et al., 2008) based on the complementary hypothesis (Bouchet, 1963) using VPD and RH leads to large uncertainties in evaporation from the unsaturated soil. Finally, the coarse resolution meteorological data (1° × 1.25°) used in MOD16 may not be well representative of surfaces with high moisture availability."

References

Allen, R. G., Pereira, L. S., Raes, D., and Smith, M.: Crop evapotranspiration-Guidelines for computing crop water requirements-FAO Irrigation and drainage paper 56, FAO, Rome, 300, D05109, 1998.

Allen, R. G., Tasumi, M., and Trezza, R.: Satellite-based energy balance for mapping evapotranspiration with internalized calibration (METRIC) - Model, J Irrig Drain Eng, 133, 380-394, 2007.

ASCE-EWRI: The ASCE standardized reference evapotranspiration equation: ASCE-EWRI Standardization of Reference Evapotranspiration Task Committe Report, 2005,

Baldocchi, D. D., and Ryu, Y.: A Synthesis of Forest Evaporation Fluxes – from Days to Years – as Measured with Eddy Covariance, in: Forest Hydrology and Biogeochemistry: Synthesis of Past Research and Future Directions, edited by: Levia, D. F., Carlyle-Moses, D., and Tanaka, T., Springer Netherlands, Dordrecht, 101-116, 2011.

Biggs, T. W., Marshall, M., and Messina, A.: Mapping daily and seasonal evapotranspiration from irrigated crops using global climate grids and satellite imagery: Automation and methods comparison, Water Resour Res, 52, 7311-7326, 10.1002/2016WR019107, 2016.

Bouchet, R.: Evapotranspiration reelle, evapotranspiration potentielle, et production agricole, Annales Agronomiques, 1963, 743-824,

Brutsaert, W., and Sugita, M.: Application of self-preservation in the diurnal evolution of the surface energy budget to determine daily evaporation, Journal of Geophysical Research: Atmospheres, 97, 18377-18382, 10.1029/92JD00255, 1992.

Bunting, D. P., Kurc, S. A., Glenn, E. P., Nagler, P. L., and Scott, R. L.: Insights for empirically modeling evapotranspiration influenced by riparian and upland vegetation in semiarid regions, Journal of Arid Environments, 111, 42-52, https://doi.org/10.1016/j.jaridenv.2014.06.007, 2014.

Cammalleri, C., Anderson, M., and Kustas, W.: Upscaling of evapotranspiration fluxes from instantaneous to daytime scales for thermal remote sensing applications, Hydrology and Earth System Sciences, 18, 1885-1894, 2014.

Chávez, J. L., Neale, C. M. U., Prueger, J. H., and Kustas, W. P.: Daily evapotranspiration estimates from extrapolating instantaneous airborne remote sensing ET values, Irrigation Sci, 27, 67-81, 10.1007/s00271-008-0122-3, 2008.

Chen, X., Su, Z., Ma, Y., Yang, K., Wen, J., and Zhang, Y.: An Improvement of Roughness Height Parameterization of the Surface Energy Balance System (SEBS) over the Tibetan Plateau, Journal of Applied Meteorology and Climatology, 52, 607-622, 10.1175/jamc-d-12-056.1, 2013.

Colaizzi, P., Evett, S., Howell, T., and Tolk, J.: Comparison of five models to scale daily evapotranspiration from one-time-of-day measurements, Transactions of the ASABE, 49, 1409-1417, 2006.

Crago, R. D.: Conservation and variability of the evaporative fraction during the daytime, Journal of Hydrology, 180, 173-194, 1996.

Ershadi, A., McCabe, M. F., Evans, J. P., Mariethoz, G., and Kavetski, D.: A Bayesian analysis of sensible heat flux estimation: Quantifying uncertainty in meteorological forcing to improve model prediction, Water Resour Res, 49, 2343-2358, 2013.

Fisher, J. B., Tu, K. P., and Baldocchi, D. D.: Global estimates of the land–atmosphere water flux based on monthly AVHRR and ISLSCP-II data, validated at 16 FLUXNET sites, Remote Sensing of Environment, 112, 901-919, https://doi.org/10.1016/j.rse.2007.06.025, 2008.

Garcia, M., Fernández, N., Villagarcía, L., Domingo, F., Puigdefábregas, J., and Sandholt, I.: Accuracy of the Temperature–Vegetation Dryness Index using MODIS under water-limited vs. energy-limited evapotranspiration conditions, Remote Sensing of Environment, 149, 100-117, https://doi.org/10.1016/j.rse.2014.04.002, 2014.

Gentine, P., Entekhabi, D., Chehbouni, A., Boulet, G., and Duchemin, B.: Analysis of evaporative fraction diurnal behaviour, Agricultural and Forest Meteorology, 143, 13-29, https://doi.org/10.1016/j.agrformet.2006.11.002, 2007.

Hu, L., and Brunsell, N. A.: The impact of temporal aggregation of land surface temperature data for surface urban heat island (SUHI) monitoring, Remote Sensing of Environment, 134, 162-174, https://doi.org/10.1016/j.rse.2013.02.022, 2013.

Ichii, K., Wang, W., Hashimoto, H., Yang, F., Votava, P., Michaelis, A. R., and Nemani, R. R.: Refinement of rooting depths using satellite-based evapotranspiration seasonality for ecosystem modeling in California, Agricultural and Forest Meteorology, 149, 1907-1918, https://doi.org/10.1016/j.agrformet.2009.06.019, 2009.

Jiang, C., and Ryu, Y.: Multi-scale evaluation of global gross primary productivity and evapotranspiration products derived from Breathing Earth System Simulator (BESS), Remote Sensing of Environment, 186, 528-547, https://doi.org/10.1016/j.rse.2016.08.030, 2016.

Jin, Y., Randerson, J. T., and Goulden, M. L.: Continental-scale net radiation and evapotranspiration estimated using MODIS satellite observations, Remote Sensing of Environment, 115, 2302-2319, https://doi.org/10.1016/j.rse.2011.04.031, 2011.

Jung, M., Reichstein, M., Ciais, P., Seneviratne, S. I., Sheffield, J., Goulden, M. L., Bonan, G., Cescatti, A., Chen, J., de Jeu, R., Dolman, A. J., Eugster, W., Gerten, D., Gianelle, D., Gobron, N., Heinke, J., Kimball, J., Law, B. E., Montagnani, L., Mu, Q., Mueller, B., Oleson, K., Papale, D., Richardson, A. D., Roupsard, O., Running, S., Tomelleri, E., Viovy, N., Weber, U., Williams, C., Wood, E., Zaehle, S., and Zhang, K.: Recent decline in the global land evapotranspiration trend due to limited moisture supply, Nature, 467, 951, 10.1038/nature09396

https://www.nature.com/articles/nature09396#supplementary-information, 2010.

Jung, M., Reichstein, M., Margolis, H. A., Cescatti, A., Richardson, A. D., Arain, M. A., Arneth, A., Bernhofer, C., Bonal, D., Chen, J., Gianelle, D., Gobron, N., Kiely, G., Kutsch, W., Lasslop, G., Law, B. E., Lindroth, A., Merbold, L., Montagnani, L., Moors, E. J., Papale, D., Sottocornola, M., Vaccari, F., and Williams, C.: Global patterns of land-atmosphere fluxes of carbon dioxide, latent heat, and sensible heat derived from eddy covariance, satellite, and meteorological observations, Journal of Geophysical Research: Biogeosciences, 116, n/a-n/a, 10.1029/2010JG001566, 2011.

Long, D., and Singh, V. P.: Integration of the GG model with SEBAL to produce time series of evapotranspiration of high spatial resolution at watershed scales, Journal of Geophysical Research: Atmospheres, 115, n/a-n/a, 10.1029/2010JD014092, 2010.

Ma, S., Baldocchi, D. D., Xu, L., and Hehn, T.: Inter-annual variability in carbon dioxide exchange of an oak/grass savanna and open grassland in California, Agricultural and Forest Meteorology, 147, 157-171, https://doi.org/10.1016/j.agrformet.2007.07.008, 2007.

Mallick, K., Jarvis, A. J., Boegh, E., Fisher, J. B., Drewry, D. T., Tu, K. P., Hook, S. J., Hulley, G., Ardö, J., Beringer, J., Arain, A., and Niyogi, D.: A Surface Temperature Initiated Closure (STIC) for surface energy balance fluxes, Remote Sensing of Environment, 141, 243-261, http://dx.doi.org/10.1016/j.rse.2013.10.022, 2014.

Mallick, K., Boegh, E., Trebs, I., Alfieri, J. G., Kustas, W. P., Prueger, J. H., Niyogi, D., Das, N., Drewry, D. T., Hoffmann, L., and Jarvis, A. J.: Reintroducing radiometric surface temperature into the Penman-Monteith formulation, Water Resour Res, 51, 6214-6243, 10.1002/2014wr016106, 2015.

Mallick, K., Trebs, I., Boegh, E., Giustarini, L., Schlerf, M., Drewry, D. T., Hoffmann, L., von Randow, C., Kruijt, B., Araùjo, A., Saleska, S., Ehleringer, J. R., Domingues, T. F., Ometto, J. P. H. B., Nobre, A. D., de Moraes, O. L.

L., Hayek, M., Munger, J. W., and Wofsy, S. C.: Canopy-scale biophysical controls of transpiration and evaporation in the Amazon Basin, Hydrol. Earth Syst. Sci., 20, 4237-4264, 10.5194/hess-20-4237-2016, 2016.

Martens, B., Gonzalez Miralles, D., Lievens, H., van der Schalie, R., de Jeu, R. A., Fernández-Prieto, D., Beck, H. E., Dorigo, W., and Verhoest, N.: GLEAM v3: Satellite-based land evaporation and root-zone soil moisture, Geosci Model Dev, 10, 1903-1925, 2017.

Masuoka, E., Fleig, A., Wolfe, R. E., and Patt, F.: Key characteristics of MODIS data products, IEEE Transactions on Geoscience and Remote Sensing, 36, 1313-1323, 10.1109/36.701081, 1998.

McCabe, M. F., Ershadi, A., Jimenez, C., Miralles, D. G., Michel, D., and Wood, E. F.: The GEWEX LandFlux project: evaluation of model evaporation using tower-based and globally gridded forcing data, Geosci Model Dev, 9, 283-305, 10.5194/gmd-9-283-2016, 2016.

Miralles, D., Holmes, T., De Jeu, R., Gash, J., Meesters, A., and Dolman, A.: Global land-surface evaporation estimated from satellite-based observations, Hydrology and Earth System Sciences, 15, 453, 2011.

Mu, Q., Zhao, M., and Running, S. W.: Improvements to a MODIS global terrestrial evapotranspiration algorithm, Remote Sensing of Environment, 115, 1781-1800, http://dx.doi.org/10.1016/j.rse.2011.02.019, 2011.

Running, S. W., Nemani, R. R., Heinsch, F. A., Zhao, M., Reeves, M., and Hashimoto, H.: A Continuous Satellite-Derived Measure of Global Terrestrial Primary Production, BioScience, 54, 547-560, 10.1641/0006-3568(2004)054[0547:ACSMOG]2.0.CO;2, 2004.

Ryu, Y., Baldocchi, D. D., Kobayashi, H., van Ingen, C., Li, J., Black, T. A., Beringer, J., van Gorsel, E., Knohl, A., Law, B. E., and Roupsard, O.: Integration of MODIS land and atmosphere products with a coupled-process model to estimate gross primary productivity and evapotranspiration from 1 km to global scales, Global Biogeochemical Cycles, 25, n/a-n/a, 10.1029/2011GB004053, 2011.

Ryu, Y., Baldocchi, D. D., Black, T. A., Detto, M., Law, B. E., Leuning, R., Miyata, A., Reichstein, M., Vargas, R., Ammann, C., Beringer, J., Flanagan, L. B., Gu, L., Hutley, L. B., Kim, J., McCaughey, H., Moors, E. J., Rambal, S., and Vesala, T.: On the temporal upscaling of evapotranspiration from instantaneous remote sensing measurements to 8-day mean daily-sums, Agricultural and Forest Meteorology, 152, 212-222, https://doi.org/10.1016/j.agrformet.2011.09.010, 2012.

Senay, G. B., Bohms, S., Singh, R. K., Gowda, P. H., Velpuri, N. M., Alemu, H., and Verdin, J. P.: Operational Evapotranspiration Mapping Using Remote Sensing and Weather Datasets: A New Parameterization for the SSEB Approach, J Am Water Resour As, 49, 577-591, 2013.

Shuttleworth, W., Gurney, R., Hsu, A., and Ormsby, J.: FIFE: the variation in energy partition at surface flux sites, IAHS Publ, 186, 1989.

Sims, D. A., Rahman, A. F., Cordova, V. D., El-Masri, B. Z., Baldocchi, D. D., Bolstad, P. V., Flanagan, L. B., Goldstein, A. H., Hollinger, D. Y., Misson, L., Monson, R. K., Oechel, W. C., Schmid, H. P., Wofsy, S. C., and Xu, L.: A new model of gross primary productivity for North American ecosystems based solely on the enhanced vegetation index and land surface temperature from MODIS, Remote Sensing of Environment, 112, 1633-1646, https://doi.org/10.1016/j.rse.2007.08.004, 2008.

Stoy, P. C., Mauder, M., Foken, T., Marcolla, B., Boegh, E., Ibrom, A., Arain, M. A., Arneth, A., Aurela, M., Bernhofer, C., Cescatti, A., Dellwik, E., Duce, P., Gianelle, D., van Gorsel, E., Kiely, G., Knohl, A., Margolis, H., McCaughey, H., Merbold, L., Montagnani, L., Papale, D., Reichstein, M., Saunders, M., Serrano-Ortiz, P., Sottocornola, M., Spano, D., Vaccari, F., and Varlagin, A.: A data-driven analysis of energy balance closure across FLUXNET research sites: The role of landscape scale heterogeneity, Agricultural and Forest Meteorology, 171, 137-152, http://dx.doi.org/10.1016/j.agrformet.2012.11.004, 2013.

Tang, R., Tang, B., Wu, H., and Li, Z.-L.: On the feasibility of temporally upscaling instantaneous evapotranspiration using weather forecast information, Int J Remote Sens, 36, 4918-4935, 10.1080/01431161.2015.1029597, 2015.

Tian, F., Qiu, G., Yang, Y., Lü, Y., and Xiong, Y.: Estimation of evapotranspiration and its partition based on an extended three-temperature model and MODIS products, Journal of Hydrology, 498, 210-220, https://doi.org/10.1016/j.jhydrol.2013.06.038, 2013.

van der Kwast, J., Timmermans, W., Gieske, A., Su, Z., Olioso, A., Jia, L., Elbers, J., Karssenberg, D., and de Jong, S.: Evaluation of the Surface Energy Balance System (SEBS) applied to ASTER imagery with flux-measurements at the SPARC 2004 site (Barrax, Spain), Hydrol. Earth Syst. Sci., 13, 1337-1347, 10.5194/hess-13-1337-2009, 2009.

Velpuri, N. M., Senay, G. B., Singh, R. K., Bohms, S., and Verdin, J. P.: A comprehensive evaluation of two MODIS evapotranspiration products over the conterminous United States: Using point and gridded FLUXNET and water balance ET, Remote Sensing of Environment, 139, 35-49, 2013.

Vinukollu, R. K., Meynadier, R., Sheffield, J., and Wood, E. F.: Multi-model, multi-sensor estimates of global evapotranspiration: climatology, uncertainties and trends, Hydrological Processes, 25, 3993-4010, 2011.

Wu, C., Munger, J. W., Niu, Z., and Kuang, D.: Comparison of multiple models for estimating gross primary production using MODIS and eddy covariance data in Harvard Forest, Remote Sensing of Environment, 114, 2925-2939, https://doi.org/10.1016/j.rse.2010.07.012, 2010.

Xiao, J., Zhuang, Q., Baldocchi, D. D., Law, B. E., Richardson, A. D., Chen, J., Oren, R., Starr, G., Noormets, A., Ma, S., Verma, S. B., Wharton, S., Wofsy, S. C., Bolstad, P. V., Burns, S. P., Cook, D. R., Curtis, P. S., Drake, B. G., Falk, M., Fischer, M. L., Foster, D. R., Gu, L., Hadley, J. L., Hollinger, D. Y., Katul, G. G., Litvak, M., Martin, T. A., Matamala, R., McNulty, S., Meyers, T. P., Monson, R. K., Munger, J. W., Oechel, W. C., Paw U, K. T., Schmid, H. P., Scott, R. L., Sun, G., Suyker, A. E., and Torn, M. S.: Estimation of net ecosystem carbon exchange for the conterminous United States by combining MODIS and AmeriFlux data, Agricultural and Forest Meteorology, 148, 1827-1847, https://doi.org/10.1016/j.agrformet.2008.06.015, 2008.

Xiao, J., Zhuang, Q., Law, B. E., Chen, J., Baldocchi, D. D., Cook, D. R., Oren, R., Richardson, A. D., Wharton, S., Ma, S., Martin, T. A., Verma, S. B., Suyker, A. E., Scott, R. L., Monson, R. K., Litvak, M., Hollinger, D. Y., Sun, G., Davis, K. J., Bolstad, P. V., Burns, S. P., Curtis, P. S., Drake, B. G., Falk, M., Fischer, M. L., Foster, D. R., Gu, L., Hadley, J. L., Katul, G. G., Matamala, R., McNulty, S., Meyers, T. P., Munger, J. W., Noormets, A., Oechel, W. C., Paw U, K. T., Schmid, H. P., Starr, G., Torn, M. S., and Wofsy, S. C.: A continuous measure of gross primary production for the conterminous United States derived from MODIS and AmeriFlux data, Remote Sensing of Environment, 114, 576-591, https://doi.org/10.1016/j.rse.2009.10.013, 2010.

Xiong, Y., Zhao, S., Yin, J., Li, C., and Qiu, G.: Effects of Evapotranspiration on Regional Land Surface Temperature in an Arid Oasis Based on Thermal Remote Sensing, IEEE Geoscience and Remote Sensing Letters, 13, 1885-1889, 10.1109/LGRS.2016.2616409, 2016.

Xiong, Y. J., Zhao, S. H., Tian, F., and Qiu, G. Y.: An evapotranspiration product for arid regions based on the three-temperature model and thermal remote sensing, Journal of Hydrology, 530, 392-404, https://doi.org/10.1016/j.jhydrol.2015.09.050, 2015.

Yang, F., White, M. A., Michaelis, A. R., Ichii, K., Hashimoto, H., Votava, P., Zhu, A. X., and Nemani, R. R.: Prediction of Continental-Scale Evapotranspiration by Combining MODIS and AmeriFlux Data Through Support Vector Machine, IEEE Transactions on Geoscience and Remote Sensing, 44, 3452-3461, 10.1109/TGRS.2006.876297, 2006.

Yang, F., Ichii, K., White, M. A., Hashimoto, H., Michaelis, A. R., Votava, P., Zhu, A. X., Huete, A., Running, S. W., and Nemani, R. R.: Developing a continental-scale measure of gross primary production by combining MODIS

and AmeriFlux data through Support Vector Machine approach, Remote Sensing of Environment, 110, 109-122, https://doi.org/10.1016/j.rse.2007.02.016, 2007.

Yang, Y., Shang, S., Guan, H., and Jiang, L.: A novel algorithm to assess gross primary production for terrestrial ecosystems from MODIS imagery, Journal of Geophysical Research: Biogeosciences, 118, 590-605, 10.1002/jgrg.20056, 2013.

Yang, Y., Long, D., Guan, H., Liang, W., Simmons, C., and Batelaan, O.: Comparison of three dual-source remote sensing evapotranspiration models during the MUSOEXE-12 campaign: Revisit of model physics, Water Resour Res, 51, 3145-3165, 10.1002/2014WR015619, 2015.

Yuan, W., Liu, S., Yu, G., Bonnefond, J.-M., Chen, J., Davis, K., Desai, A. R., Goldstein, A. H., Gianelle, D., Rossi, F., Suyker, A. E., and Verma, S. B.: Global estimates of evapotranspiration and gross primary production based on MODIS and global meteorology data, Remote Sensing of Environment, 114, 1416-1431, https://doi.org/10.1016/j.rse.2010.01.022, 2010.

Zhang, Y., Leuning, R., Hutley, L. B., Beringer, J., McHugh, I., and Walker, J. P.: Using long-term water balances to parameterize surface conductances and calculate evaporation at 0.05° spatial resolution, Water Resour Res, 46, n/a-n/a, 10.1029/2009WR008716, 2010.

---

## Referee Report (RR1)

1. Below table is my result from global daily ET. I did not see overestimation, when I also use satellite data and SEBS model to calculate ET. You need be careful about your input data. I suggest list your canopy height used for SEBS in table 1. When you want to assess a model, all the parameters and input data (LST, air temperature, wind speed) should come from ground truth observation. Unfortunately, most of model users are not responsible for doing this evaluation. Satellite data are easily used to force model and conclude that the model has some problems when they compare remote sensing data based model result with ground flux measurement. To assess a model, we should first to remove bias in the forcing data. Otherwise, the result evaluation is confusing. The readers don`t understand where does the errors come from. If you want say the overestimation in SEBS is due to the model problem, It is better to do in-situ simulation with all ground measurement to check whether the conclusion is the same as what from using satellite data.

| Num | Sname | Lat | Lon | IGBP LCT | | $ET_d$ vs. EC (mm/d) | | | $Rn_d$ vs. EC (MJ/m$^2$d) | | |
|---|---|---|---|---|---|---|---|---|---|---|---|
| | | | | Site | MOD12 | RMSE | MB | r | RMSE | MB | r |
| 112 | Tonzi_Ranch | 38.4316 | -120.966 | SAV | SAV | 1 | 0.19 | 0.61 | NaN | NaN | NaN |
| 101 | Santa_Rita_Mesquite_Savanna | 31.8214 | -110.866 | SRB | SRB | 1.02 | -0.24 | 0.44 | 4.65 | -3.91 | 0.8 |

2. Please give the flux tower names in Figure 4 and 5. GRA, WSA, ENF is kind of mis-understanding information. One GRS site can not represent all 'GRS' sites.

3. I don`t agree the way you calculate kb_1. Z0m equation from Van der Kwast is empirical method. You have agreed this by saying that 'z0M was derived using a simple empirical relationship between the roughness length of momentum transfer, z0M, and NDVI, as suggested by Van der Kwast et al. (2009) [Page 9, Lines 23-24].'. Van der Kwast method cannot be used for forest or all kinds of canopy. Your using of z0h from Yang et al. 2002 also has some problem. Yang`s method cannot be used for canopy site, since his method has used a 0.003???? z0m initial value to calculate z0h. This problem is already discussed by Prof. Hotslag and other micrometeorologist. But remote sensing community rarely notice this. I think the code from Abouali, Mohammad also have this problem when he replace SEBS`s kb_1 with Yang` heat roughness scheme. The idea in Chen et al. 2013 is to merge Yang's method into SEBS's, due to Yang has a better performance than the Brutsaert method which is the soil part in SEBS kB_1:

$$kB_s^{-1} = 2.46(\mathrm{Re}_*)^{1/4} - \ln[7.4]$$

from Brusaert.

$$z_{0h} = (70\vartheta/u_*)\exp(-7.2u_*^{0.5}\theta_*^{0.25}) \quad \text{and} \quad \text{(10a)}$$

$$kB_s^{-1} = \log(z_{0m}/z_{0h}), \quad \text{(10b)}$$

from Chen et al. 2013

By fusing roughness schemes from both SEBS and Yang, Chen et al. 2013 can not only use the new kB_1 for bare soil but also canopy surface, which idea was already designed by Su 2002. In addition, no publication has tested Yang's scheme seriously over canopy covers. What is your argument for using Yang's method over forest or cropland? If you use yang's method how did you set the initial z0m value for this land covers?
The relationship between these roughness scheme and publications should be clarified in this paper.

4. About Fig. 13, the only solution is to calculate kb_1 from flux observation. You can use momentum and sensible heat flux to inversely calculate z0m and z0h, Then kb_1 can be calculated from 'observed' z0m and z0h. Otherwise the result in figure 13 is not trustable. In addition, I did not see any 0 values of kb_1, which is quite popular for forest sites. If the authors cannot derive kb_1 from observation, I suggest to remove this figure and about the analysis of kb_1. Or land cover could be taken as x-axis.

5. Figure 9,10 and 11, I need a plot of land covers, canopy height (its importance has be mentioned in one of your paper) to analyze the possible error source in SEBS, beside your using of Yang`s method.

6. I don't agree that you can use heat roughness to parameterize kB_1. This is contradict with designing of kb_1. kB_1 is intermedia variable which connect z0m and z0h, due to kb_1 cannot be used in MOST directly. Z0h should be deduced from kB_1 not the inverse way. Please revise this sentence 'The roughness height for heat transfer proposed by Yang et al. (2002), was used to parametrize kB-1.'.

And also the following sentences, as I said figure 13 has a problem:
z0M was derived using a simple empirical relationship between the roughness length of momentum transfer, z0M, and NDVI, as suggested by Van der Kwast et al. (2009). The roughness height for heat transfer proposed by Yang et al. (2002), was used to parametrize kB-1. This new parametrization of kB-1 was designed to improve the SEBS model performances on bare soil, low canopies, and snow surfaces as was proposed by Chen et al. (2013)."

7. Please rewrite these sentence:

The source codes for different sub-models within SEBS were either adapted or modified from Abouali et al. (2013).---->The SEBS codes in this study is adapted from Abouali et al. (2013), which is different from original version in Su 2002 and Chen et al. 2013.

8. Most of SEBS study found ET over ENF and DBF is overestimated, however, this is not reflected in your study. Please give some explanation.

---

## Author Response (AR2)

**Manuscript Title:** Regional evapotranspiration from image-based implementation of the Surface Temperature Initiated Closure (STIC1.2) model and its validation across an aridity gradient in the conterminous United States

**Authors:** Nishan Bhattarai, Kaniska Mallick, Nathaniel A. Brunsell, Ge Sun, Meha Jain
Correspondence to: Nishan Bhattarai (nbhattar@umich.edu)

Dear Dr. Bob Su,

Thank you for the opportunity to submit a revised manuscript. We would also like to thank all three referees for their comments and suggestions. Please find our responses to referee's comments (both round #1 and 2) and a revised manuscript with track changes attached in this document. In summary, we have made following key changes in the manuscript to address all comments from the referees.

1) More discussion on uncertainties associated with the use of a simplified version of surface roughness and $kB^{-1}$ parameter in SEBS (Reviewer # 1)
2) More discussion on the performance of the prior version of SEBS on different land cover types (Reviewer # 1)
3) Discussion on uncertainties associated with the use of 8-day aggregated data (Reviewer # 2)
4) Discussion on uncertainties associated with MOD16 (Reviewer # 2)
5) Results from a quick comparison of results with recently developed global SEBS monthly ET product (Reviewer # 3 and Editor)
6) Supplementary tables to address Reviewer #2's comments on implementations of ET models at instantaneous scale and 8-day scales.
7) Modifications to figure 4 and 5 (Reviewer # 1).

We hope you find the revised version of this manuscript to be greatly improved for publication in HESS.

Sincerely,

Nishan Bhattarai and all the coauthors

**Responses[1] to Referee # 1 (Round 2)**

*The authors shoud do more analysis on the model evaluation, especially when they are assessing three models. Any conclusion should be carefull, which might mis-leading reader who have not deep knowleged of the model physics.*

**Response:** We thank Referee # 1(R#1) for comments and suggestions on explicit evaluation.

We would like to mention that this manuscript does not intend to provide a detailed model inter-comparison of the three models, as there could be up to large uncertainties in the observed fluxes. We intended to demonstrate the regional ET mapping potential of the STIC1.2 model which is independent of any land surface parameterization of aerodynamic conductance. SEBS (thermal) and MOD16 (non-thermal) models were selected because of their widespread use and great potential for regional scale ET mapping. In the revised version, this statement is made explicitly (Page 11, Line 27-30; Page 18 Line 2-4). Further, to avoid potentially misleading information to readers about the tendency of SEBS to overestimate ET, we have stated that a previous version of the SEBS model was used to characterize surface roughness (Page 10, Line 14-16). In addition, a recent version of SEBS (SEBS$_{Chen}$ hereafter) global ET outputs (monthly and annual) is also presented and compared against STIC1.2 and observed annual ET. The relatively better performance of SEBS$_{Chen}$ (Fig. 12) highlights the importance of better aerodynamic resistance characterization in SEBS. In addition, the good correlation between monthly and annual ETs from SEBS$_{Chen}$ and STIC1.2 models provides an interesting insight for designing a multimodal-based thermal ET modeling framework in the future. Please find our responses to the comments below:

*Comment # 1. Below table is my result from global daily ET. I did not see overestimation, when I also use satellite data and SEBS model to calculate ET. You need be careful about your input data. I suggest list your canopy height used for SEBS in table 1. When you want to assess a model, all the parameters and input data (LST, air temperature, wind speed) should come from ground truth observation. Unfortunately, most of model users are not responsible for doing this evaluation. Satellite data are easily used to force model and conclude that the model has some problems when they compare remote sensing data based model result with ground flux measurement. To assess a model, we should first to remove bias in the forcing data. Otherwise, the result evaluation is confusing. The readers don`t understand where does the errors come from. If you want say the overestimation in SEBS is due to the model problem, It is better to do in-situ simulation with all ground measurement to check whether the conclusion is the same as what from using satellite data.*
* * *
[1] Page and Line numbers are referred to the clean version of the revised manuscript

| Num | Sname | Lat | Lon | IGBP LCT | | $ET_d$ vs. EC (mm/d) | | | $Rn_d$ vs. EC (MJ/m$^2$d) | | |
|---|---|---|---|---|---|---|---|---|---|---|---|
| | | | | Site | MOD12 | RMSE | MB | r | RMSE | MB | r |
| 112 | Tonzi_Ranch | 38.4316 | -120.966 | SAV | SAV | 1 | 0.19 | 0.61 | NaN | NaN | NaN |
| 101 | Santa_Rita_Mesquite_Savanna | 31.8214 | -110.866 | SRB | SRB | 1.02 | -0.24 | 0.44 | 4.65 | -3.91 | 0.8 |

**Response**: We appreciate R#1 for sharing the results. There could be a number of reasons for the difference in model performances with the most important ones being the use of different versions of SEBS that use $z_{OM}$ parameterizations derived from NDVI and the use of 8-day MODIS LST data and 8-day aggregated gridded data. In addition, we considered one dry, one wet, and one normal precipitation year for each flux site and because cloud free LST pixels are mostly available in dry days than wet days, the results could be positively biased, as the majority of the data points come from dry years (~40%). The use of coarse resolution gridded weather data and model implementation at 8-day temporal scales (please check our response for comment #2 from reviewer #2 and tables S2) are other sources of uncertainties. Prior studies using the same version of SEBS as used in this study have resulted in similar performances (RMSEs of 0.74 mm/day to 1.08 mm/day and PBIASs within 10%), as shown in this table, when implemented at several sites in Florida and Oklahoma (Bhattarai et al., 2016;Wagle et al., 2017). However, in those implementations, $z_{OM}$ was empirically derived from actual canopy height from tower and weather stations using the exponential relationship between remotely sensed NDVI and albedo and $z_{OM}$ (Allen et al., 2007). In this approach, the linear regression model (unique for each day) always forces $z_{OM}$ and canopy height to be close to actual values. In this study, we attempted to reduce such empirically derived parameters and rely on remotely sensed products. In addition, we relied on gridded products (NLDAS) for weather forcing to produce spatially distributed ET maps, as opposed to weather station data used in prior studies (Bhattarai et al., 2016;Wagle et al., 2017). NLDAS wind speed (used in SEBS but not in STIC1.2) was not found to be as reliable as their air temperature and relative humidity, when compared against weather station data in the western US (Lewis et al., 2014). We have explicitly stated these uncertainties in the discussion section (Page 18, Line 28-Page 19, Line 14).

We did not use a static value for canopy height (hc). We deduced canopy height from $z_{OM}$ (hc = $z_{OM}$ /0.13) (Brutsaert, 1982). While the point scale implementation using observed data could provide a true evaluation of the model, it cannot be used as a demonstrative tool for estimating ET maps at regional scales, as currently observed data is not possible to obtain for each pixel within an image. In addition, SEBS is one of the most widely used ET models within the thermal ET community. Studies (Vinukollu et al., 2011b; Su et al., 2005) suggest that while SEBS errors range from 1 to 15% when tower data are used, and the error could be significantly increased when remotely sensed and reanalysis data are used (McCabe et al., 2016a). Evaluation of SEBS at the instantaneous scale shows that SEBS ET estimates were within 17% of observed ET and use of 8-day MODIS products and 8-day average gridded products essentially added about 11% additional error (Table S3 and Page 18, Line 28-Page 19, Line 14 in the discussion). We believe

that most model errors associated with remote sensing based implementations of the SEBS model should come from uncertainties associated with input parameters and our study also revealed the same. In the revised version of this manuscript, we have explicitly stated the key uncertainties associated with SEBS implementation and also provided results using recently developed global SEBS ET products (Chen et al. 2013) that potentially overcome the uncertainties associated with surface roughness and $kB^{-1}$ parametrizations (Page 17, Line 30-Page 18 Line 4). Improved results with the latest SEBS global ET products does validate the need for better characterization of aerodynamic resistance through improved $kB^{-1}$ parametrization in the SEBS model.

In the revised version, we have also stated that the paper does not intend to provide a model inter-comparison, as there could be up to 30% uncertainties in the observed fluxes itself and other uncertainties associated with input data (Page 11, Line 27-30; Page 18 Line 1-2). We attempted to demonstrate the regional scale ET mapping potential of the STIC1.2 model using remotely sensed inputs (Page 11, Line 27-30).

*Comment # 2. Please give the flux tower names in Figure 4 and 5. GRA, WSA, ENF is kind of misunderstanding information. One GRS site can not represent all 'GRS' sites.*

**Response:** Figure 4 and 5 have been modified accordingly. A color code for each site has been used in all figures in the paper when flux sites are separated by colors.

*Comment # 3: I don`t agree the way you calculate kb_1. Z0m equation from Van der Kwast is empirical method. You have agreed this by saying that 'z0M was derived using a simple empirical relationship between the roughness length of momentum transfer, z0M, and NDVI, as suggested by Van der Kwast et al. (2009) [Page 9, Lines 23-24].'. Van der Kwast method cannot be used for forest or all kinds of canopy. Your using of z0h from Yang et al. 2002 also has some problem. Yang`s method cannot be used for canopy site, since his method has used a 0.003???? z0m initial value to calculate z0h. This problem is already discussed by Prof. Hotslag and other micrometeorologist. But remote sensing community rarely notice this. I think the code from Abouali, Mohammad also have this problem when he replace SEBS`s kb_1 with Yang` heat roughness scheme. The idea in Chen et al. 2013 is to merge Yang's method into SEBS's, due to Yang has a better performance than the Brutsaert method which is the soil part in SEBS kB_1:*

$$kB_s^{-1} = 2.46 \left( \text{Re}_* \right)^{1/4} - \ln[7.4]$$

from Brusaert.

$$z_{0h} = (70\vartheta/u_*)\exp(-7.2u_*^{0.5}\theta_*^{0.25}) \quad \text{and} \quad \text{(10a)}$$

$$kB_s^{-1} = \log(z_{0m}/z_{0h}), \quad \text{(10b)}$$

from Chen et al. 2013

*By fusing roughness schemes from both SEBS and Yang, Chen et al. 2013 can not only use the new kB_1 for bare soil but also canopy surface, which idea was already designed by Su 2002. In addition, no publication has tested Yang's scheme seriously over canopy covers. What is your argument for using Yang's method over forest or cropland? If you use yang's*

*method how did you set the initial z0m value for this land covers? The relationship between these roughness scheme and publications should be clarified in this paper.*

**Response:** We apologize for not better explaining how $kB^{-1}$, $z_{OH}$, and $z_{OM}$ were computed. Our rationale for using the van der Kwast et al. (2009) method was to provide a spatial representation of $z_{OM}$ across all study pixels and provide an estimate of canopy height (hc) and displacement height (d0) for use in the SEBS model. By using van der Kwast et al. (2009) method, we aim to remove the requirement for knowing canopy height for each pixel in the image. We have stated this on Page 10, Line 14-16. The work-flow for $kB^{-1}$ was as follows:

1. $kB^{-1}$ for Full canopy (kB1_c) using Choudhury and Monteith (1988)
2. $kB^{-1}$ for mixed canopy (kB1_m) using Choudhury and Monteith (1988)
3. $kB^{-1}$ for soil (kB1_s) using Brutsaert (1982)
4. $kB^{-1}$ = fvc$^2$ * kB1_c + 2 *(fvc *fs) * kB1_m + (fs$^2$) * kB1_s; where fvc and fs are fractional vegetation cover and fractional soil coverage, respectively.
5. Use extra parameterization for bare soil and snow, according to Yang et al. (2002)

We used Yang's method of $kB^{-1}$for pixels whose fractional vegetation cover was less than 1% or LAI was 0. The initial $z_{OM}$ value for bare soil was set as 0.005 and hence essentially this approach was not used for forest and crop covers. This would not affect our results, as our sites are mostly vegetated but could have worked better for potential bare soil pixels across the spatial domain (Fig. 2) used for deriving regional scale ET. Notably, we found SEBS to be working fine under crop and forest conditions.

*Comment # 4: About Fig. 13, the only solution is to calculate kb_1 from flux observation. You can use momentum and sensible heat flux to inversely calculate z0m and z0h, Then kb_1 can be calculated from 'observed' z0m and z0h. Otherwise the result in figure 13 is not trustable. In addition, I did not see any 0 values of kb_1, which is quite popular for forest sites. If the authors cannot derive kb_1 from observation, I suggest to remove this figure and about the analysis of kb_1. Or land cover could be taken as x-axis.*

**Response:** We do not agree with the reviewer and consider Fig. 13 to be an important part of the manuscript to assess the effects of different components of aerodynamic conductance on model agreements/disagreements. When comparing STIC1.2 with SEBS using remote sensing data it is important to assess the role of $kB^{-1}$and $z_{OM}$ on STIC1.2 versus SEBS ET difference where $kB^{-1}$ and $z_{OM}$ should be estimated using the same forcing datasets. Besides, for an inverse estimation of $kB^{-1}$ using observed sensible heat flux, we need observations of aerodynamic temperature. The aerodynamic temperature cannot be observed or estimated in the EC tower and using radiometric temperature instead of the aerodynamic temperature will result in the derivation of radiometric $kB^{-1}$.

To elaborate, through figure 13, we made an attempt to show how the difference between ET estimates from STIC1.2 and SEBS (and observed and SEBS) were associated with $kB^{-1}$ and $z_{OM}$ that are the essential parameters to define the aerodynamic resistance in the SEBS model. Hence, we were more interested in the relationship between the pattern of residual difference between models (STIC1.2 and SEBS) and observed (STIC1.2 and SEBS) than the tower based $kB^{-1}$values. We have added a sentence in the figure caption to suggest that $z_{OM}$ was derived using NDVI (van

der Kwast et al., 2009) and may not well represent tall canopies. Regarding the use of land cover based analysis, we made an explicit discussion in section 3 and 4 on how STIC1.2 and SEBS performances differ across different land cover types (Page 13, Line 4-25; Page 16, Line 23-Page 17, Line 15; Page 18, Line 2-4).

*Comment # 5: Figure 9,10 and 11, I need a plot of land covers, canopy height (its importance has be mentioned in one of your paper) to analyze the possible error source in SEBS, beside your using of Yang`s method.*

**Response:** The objective of the manuscript is to understand ET mapping potential of STIC1.2 across a wide variety of biome and hydrological regimes, and not to present the sources of errors in SEBS.

As stated in the response to comment # 1 (second paragraph), a static value of canopy height (hc) for each flux site (or the map) was not used in this study. Instead, it was deduced from $z_{OM}$ (hc = $z_{OM}$/0.13) (Brutsaert, 1982) that was derived from NDVI (van der Kwast et al., 2009) and hence hc would be different for each image used in the derivation of annual ET. We do believe that this could be one source of uncertainty, as the use of observed canopy height and $z_{OM}$ (0.13 * hc) produced much better results both at instantaneous and 8-day scales (Biases within 20% range). We have added this discussion in Page 18, Line 28-Page 19, Line 14 and results in the supplementary (Table S2). The sensitivity of the SEBS model to canopy height parametrization is already reported (Byun et al., 2014). However, such analysis is beyond the objectives and scope of this study.

Regarding the land cover maps in figures 9-11, we believe that it does not add much information to the discussion, as model performances under different land cover types are extensively discussed in sections 3 and 4. Instead, we have added them to the study area figure (Fig. 2), as we think it is more informative this way and could be used as a reference for ET maps in figures 8-11.

**Comment # 6. I don't agree that you can use heat roughness to parameterize kB_1. This is contradict with designing of kb_1. kB_1 is intermedia variable which connect z0m and z0h, due to kb_1 cannot be used in MOST directly. Z0h should be deduced from kB_1 not the inverse way. Please revise this sentence 'The roughness height for heat transfer proposed by Yang et al. (2002), was used to parametrize kB-1.'. And also the following sentences, as I said figure 13 has a problem: z0M was derived using a simple empirical relationship between the roughness length of momentum transfer, z0M, and NDVI, as suggested by Van der Kwast et al. (2009). The roughness height for heat transfer proposed by Yang et al. (2002), was used to parametrize kB-1. This new parametrization of kB-1 was designed to improve the SEBS model performances on bare soil, low canopies, and snow surfaces as was proposed by Chen et al. (2013)."**

**Response:** We apologize for this confusion. Indeed, $kB^{-1}$ was used to derive $z_{OH}$ (Equation 14). We revised this sentence as Yang et al. (2002) was used to parametrize $kB^{-1}$ for bare soil and $z_{OH}$ was estimated using Eq. 14 (Page 10, Line 14-16).

*Comment # 7. Please rewrite these sentence:*
*The source codes for different sub-models within SEBS were either adapted or modified from Abouali et al. (2013). The SEBS codes in this study is adapted from Abouali et al. (2013), which is different from original version in Su 2002 and Chen et al. 2013.*

**Response:** The change has been made (Page 10, Line 14-16).

*Comment # 8. Most of SEBS study found ET over ENF and DBF is overestimated, however, this is not reflected in your study. Please give some explanation.*

**Response:** In this study, we did find SEBS to overestimate ET from ENF and DBF by about 8% and 17%, respectively. However, we do not consider this as a poor performance given there are uncertainties associated with flux tower ET and input parameters. The overestimation of ET from ENF and DBF occurred during normal and dry years, as in the wet years ET estimates were within 5% (Supplementary Figs. S3-S5). While several studies (Michel et al., 2016;Ershadi et al., 2014) have reported SEBS to have overestimated ET in ENF and DBF, Ershadi et al. (2014) had also reported that the overestimation tendency of SEBS is not applicable everywhere (e.g. no overestimation of ET from site US-MMS in Ershadi et al., 2014).

**Responses to Referee # 3 (Round 2)**

*The authors have addressed the concerns of the previous reviews adequately.*
*However, given the fact that the authors have evaluated a previous version of the SEBS with a simple empirical parameterization of z0m with NDVI, it is suggested that a quick comparison is made with a recently completed global application of SEBS.*

*The data set can be obtained at:*
*http://en.tpedatabase.cn/portal/MetaDataInfo.jsp?MetaDataId=249454*
*This would be a timely contribution to the advances in remote sensing of land surface energy balance, heat fluxes and evaporation.*

**Response:** We thank Reviewer # 3 for the comments and suggestions. We have compared STIC1.2 results with the recently developed global monthly SEBS ET at monthly and yearly scales (Fig. 14). In addition, we also compared the annual ET from the flux tower with global SEBS ET (Fig. 12). Overall, we find that the global SEBS ET and STIC1.2 ET are well correlated at monthly ($R^2$=0.81) and yearly ($R^2$=0.58) scales. The new SEBS global annual ET (sum of all monthly ET) was within 14% of the observed annual flux ET during years and explained 56% of the variation in the observed flux annual ET. This performance was better than the old version of SEBS we used in this study and the other two models (SEBS and MOD16). We think the better characterization of surface roughness and treatment of canopy height significantly improved the performance of SEBS compared to the version used in our paper. Our main focus of the paper is to demonstrate a remote sensing based application of a new thermal-based ET model, and assessing its performance by comparing with SEBS and MOD16. Intercomparison of the SEBS models with different parametrizations of surface roughness and $kB^{-1}$ or with different forcing data (observed vs gridded data) is beyond objective and the scope of this study (Page 11, Line 27-30; Page 18 Line 2-4). However, this quick comparison does highlight the need for improved characterization of surface $z_{OM}$ and $kB^{-1}$ and careful selection of input forcing data in remote sensing based ET models.

**Responses to Referee # 1 (Round 1)**

*Comment: This paper has enlightened ET remote sensing community about the importance of aerodynamic conductance/resistance. Currently, most of the ET algorithms do not take into account the diurnal variation in this resistance. The authors have implemented STIC model at regional scale. STIC model integrates remote sensed surface temperature into Penman-Monteith equation to derive an analytical solution for the resistance and use the resolved resistance to calculate surface heat fluxes/ET. They also compare its performance with other two ET algorithms, SEBS and MOD16. SEBS model provide direct solution for surface and boundary layer conductance/resistance from momentum/heat roughness and stability. However, MOD16 uses a kind of constant resistance in its ET calculation, which explains its worst performance among the three methods. STIC use an energy balance and meteorological information to inversely retrieve the surface and boundary layer conductance. The results are sufficient to support their conclusions. The paper address very relevant scientific questions within the scope of HESS. Thus I suggest an acceptance for publication.*

**Response:** We thank the reviewer for appreciating our work and considering the manuscript to be interesting and relevant to the HESS community.

*Comment: Figure 5 shows that SEBS has a similar performance as STIC, and MOD16 for CRO, DBF, and ENF, but worse result at WSA and GRA. Please check if this is due to inaccuracy of satellite input data.*

**Response:** Fig. 5 suggests significant overestimation of SEBS ET in WSA and GRA sites. Specifically, at one of the WSA sites (e.g., US-Ton), all of the three models performed poorly, where observed ET values were extremely low (0.6 mm per day; Fig. 5). In this specific site, 10% of the observed LST pixels were within 2-3 K LST errors with surface emissivity errors of 0.01 to 0.03, based on the MODIS QA/QC data. Similarly, in another WSA site (i.e. US-SRM), 37% of observed LST pixels were within a similar error range. The literature also suggests high emissivity correction uncertainties and systematic underestimation of MODIS LST in arid and semiarid ecosystems (Wan and Li, 2008;Jin and Liang, 2006;Hulley et al., 2012). The significantly poor performance of SEBS in the US-SRM site could also be attributed to these uncertainties. We have added text in the discussion section about the performance of STIC1.2 at the US-Ton site (Page 16, Lines 7-10). It is also important to note that uncertainties also exist in the Bowen ratio energy balance closure correction of energy balance at the arid and semi-arid sites, which is also discussed on Page 16, Lines 14-19.

Regarding the poor performance of SEBS in the GRA sites, the errors are mostly due to large differences in SEBS ET and observed ET in the two semi-arid desert grasslands sites (US-SRG and US-Wkg) (Supplementary Table S1). In the two other grassland sites (US-Kon and US-KFS), SEBS performed relatively better and was comparable with STIC1.2 (Supplementary Table S1). In those two semi-arid desert grasslands sites (US-SRG and US-Wkg), the MODIS QC/QA bin suggested that 27% (US-Wkg) and 4% (US-SRG) of MODIS LST were within 2-3 K errors with emissivity errors within the 0.01 to 0.02 range, based on the MODIS QA/QC data. These errors are however more predominant in SEBS, and, as in the semi-arid and arid conditions, substantial differences exist between $T_R$ and $T_0$ due to strong soil water limitations. Such water-limited conditions may not have been properly characterized in the $kB^{-1}$ parameterization, which could lead to large differences between modeled and observed ET. We discussed these potential limitations on Page 16, Line 27-28; Page 16, Line 32 to Page 17, Line 15 and Page 16, Line 24-31. We added additional discussion on the performances of SEBS and other models in GRA and WSA sites and how the model performed differently in two GSA sites in two different climates in the revised paper (Page 16, Line 23-31).

*Comment: Fig. 8, 9, 10 shows that SEBS ET maps have higher ET than STIC and MOD16, this is due to sensible heat flux is low-estimated, because of high kB_1. Please check the reference of Chen et al. 2013.*

**Response:** This is a correct statement. The underestimation of sensible heat flux ($H$) by SEBS (nearly 41% underestimation) was mostly seen in arid and semi-arid sites, which eventually led to an overestimation of ET in these sites. Chen et al. (2013) provided an extensive discussion on how $H$ is underestimated by the original SEBS model and proposed an improved way of estimating roughness Yang et al. (2002) length for heat transfer through a new parametrization of $kB^{-1}$ adopted from Yang et al. (2002) for bare soil and snow surfaces. In this paper, we incorporated the same $kB^{-1}$ formulation from Yang et al. (2002) (source code obtained from Abouali et al., 2013). However, we adapted a simplified version of $z_{OM}$ parametrization (van der Kwast et al., 2009), which could have led to uncertainties in canopy heights. Fig. 13 suggests that ET biases (SEBS ET-Observed ET) were typically random and large when $kB^{-1}$ values were within 5 ($r$ = -0.03) and slightly negative when $kB^{-1}$ values were within 5 and 8 ($r$ = -0.16). However, for $kB^{-1}$ values > 8, a linear trend in ET bias was evident (underestimation of H) with an increase in $kB^{-1}$ ($r$ = 0.24). We added additional discussions on how uncertainties in $kB^{-1}$ parametrization could lead to biases in estimated fluxes in the revised version (Page 17, Line 29- Page 18, Line 4)

*Comment: Which method or model is used to calculate kB_1 and z0m in figure 13? Or kB_1 and z0m is derived from flux tower measurement?*

**Response:** Please refer to our responses for your comment #3 and 6 on the revised manuscript (Pages 4-7).

*Comment: Figure. 12, please have more discussion about the higher SEBS annual ET, is this due to the method in annual accumulation or SEBS model.*

**Response:** We have briefly discussed that this overestimation is mostly due to consistent overestimation of 8-day ET by the SEBS model (Page 15, lines 6-8). In addition, the 8-day average net radiation was also overestimated by 9% (Supplementary: Fig. 1), which could also add some positive biases by SEBS (and also STIC). In the two cropland sites (US-ARM and US-Ne1), SEBS annual ET estimates were within 2% of observed annual ET, which is better than the performance of STIC (22% underestimation) and MOD16 (49% underestimation). SEBS mostly overestimated annual ET from the arid and semi-arid sites (47%). We added these additional discussions in the revised manuscript (Page 15, Line 8- 14).

*Comment: Fig. 4 and 5 does not show SEBS ET has different performance over different land covers, at least does not always show high ET estimation.*

**Response:** Here we disagree. According to Fig. 4, RMSE and MAE of SEBS is similar to MOD16 across different precipitation conditions, but STIC1.2 performed better overall. According to Fig. 5, we noticed that SEBS performed best in CRO sites among all the models and its performance was similar to MOD16 in ENF sites compared to its performances in other biomes (GRA and WSA). We agree with the reviewer that SEBS does not overestimate ET all the time; as seen in Fig. 4 and 5, there are several instances when SEBS ET was lower than the observed ET. However, the overestimation tendency of ET by SEBS was predominant during the dry year (Fig. 4). The term "overestimation" refers to the mean ET observed at the flux sites. Notably, SEBS ET estimates were within 3%, 8%, and 17% of observed ET at croplands, ENF, and DBF sites, respectively, which were comparable or sometimes better than the other two models. We have briefly stated these in section 4 (Page 15, Line 8- 14; Page 16, Line 23-28; Page 16-Line 29- Page 17, Line 15; Page 17, Line 24-29).

*This manuscript provides an evaluation of STIC1.2 in estimating actual evapotranspiration at the spatial scale by combining the model with satellite remote sensing data. In addition, the authors also compare the performance of STIC1.2 with other two existing remote sensing algorithm (i.e., SEBS and MOD16). In general, the topic of this MS is of interest to the HESS' readership and the manuscript is well written. However, there are several major issues in this study, which introduce additional uncertainties and preclude a focused evaluation of the models themselves (as described below). In this light, a MAJOR REVISION is needed.*

We thank Referee #2 (R#2 hereafter) for finding our work interesting to the HESS community and providing useful criticism, comments, and suggestions. We also acknowledge R #2 for pointing out potential uncertainties associated with the use of input data. Uncertainties arising from the temporal and spatial mismatches of input datasets from different sources are common in remote sensing based studies. Acknowledging this fact, we made best possible efforts to minimize those. First, it is important to clarify that our study aims to provide a scientific basis for the operational use of STIC1.2 model towards estimating regional ET using remotely sensed data. Since the MODIS daily products suffer greatly due to cloud cover, the 8-day MODIS products are more applicable for regional ET model implementation and hence the validation was done at this temporal scale, which is a common practice (Ichii et al., 2009;Zhang et al., 2010;Yang et al., 2006;Xiong et al., 2015). The MOD11A2 product was considered to provide a better temporal and spatial representation (compared to the daily MODIS products) of regions (i.e. a wide range of biome and aridity conditions in the US). Nevertheless, we have now implemented STIC1.2 and SEBS at instantaneous (at point scale using daily MODIS and instantaneous weather data) and presented main validation results in this response. However, this implementation is not the core part of our paper and hence was only added to the supplementary (2 tables) and with a brief discussion in the main text (highlighted in red font).

In this response, we not only justify our approach (e.g. use of MODIS 8-day LST) with additional analysis and references but also show results from the evaluation of instantaneous ET from SEBS and STIC1.2 models using instantaneous/daily MODIS products (and weather inputs of the same period). Results suggest that the core findings and conclusions of this study will remain the same regardless of whether instantaneous or 8-day products are used. However, additional knowledge on the potential sensitivity of SEBS to input meteorological forcing and the applicability of the STIC1.2 model across different time scales were gained. In the revised version, we have added additional descriptions (Page 18, Line 28-Page 19, Line 14) on potential uncertainties associated with the application of models at 8-day scales, use of 8-day average (daytime) weather inputs, and the MODIS 16 ET products. We believe that the evidence we provided and detailed answers to your concerns with appropriate references will sufficiently address R #2's major concerns that are mostly associated with the use of input data and uncertainties.

*Major: 1. My largest criticism lies in the use of MOD11A2, where LST is reported as the average values of clear-sky LSTs during an 8-day period. As there is no information about which day (or days) out of each 8-days contributes to the final 8-day averages, this 8-day*

*average LST is highly likely to not correspond to the 8-day averages of meteorological variables (i.e., air temperature, VPD, etc). For example, the 8-day LST might only be a result of day-1 LST, or the average of day-1 and day-7 LSTs. Even they correspond well with each other, using 8-day averages may still lead to additional uncertainties due to differences in the temporal variability between, say, daily LST and air temperature. For example, H_day1+H_day2...+H_day8 does not equal to 1/8 \* (H_calculated using 8-day average LST and meteorological inputs), as all the responses are non-linear. To focus on evaluating the model itself, it is recommended to work on the instantaneous scale rather than 8-day averages.*

**Response:** The scientific basis for using MOD11A2 comes from an abundance of studies (for e.g. Ichii et al., 2009;Jin et al., 2011;Tian et al., 2013;Garcia et al., 2014;Xiong et al., 2015) that have also used this 8-day product in ET modeling. In our study, SEBS and STIC1.2 models were run at 8-day average scale corresponding to the MODIS daytime overpass time using MOD11A2 (and ancillary MODIS data; Table 2, Page 30) and 8-day average meteorological data corresponding at the MODIS Terra LST overview time (not the entire 8-day average; Page 10, Line 8-12 in the revised manuscript). The 8-day averaged meteorological data (that considers all hours) are only used in extrapolation of daytime ET to 8-day ET using a constant ET approach (Brutsaert and Sugita, 1992;Crago, 1996;Chávez et al., 2008). We evaluated 8-day cumulative ET to compare model performances against those from available MOD16 8-day ET products (Mu et al., 2011). We provide further justifications for using 8-day MODIS products as below:

**Comparison of 8-day vs. daily LST (or $T_R$), air temperature ($T_A$), and $T_R - T_A$**

We find that the 8-day average LST (or $T_R$), air temperature ($T_A$), and difference between $T_R$ and $T_A$ ($T_R - T_A$) were good representative of the corresponding instantaneous values during each of the 8-days within the corresponding MODIS 8-day period ($R^2$= 0.80-0.92, PBIAS within 3%) (Fig. AR1).

[Figure]

Fig AR1. Scatter plots of 8-day average LST vs. instantaneous LST, 8-day average daytime $T_A$ vs. instantaneous $T_A$, and 8-day average daytime $T_R - T_A$ vs. instantaneous $T_R - T_A$.

**Comparison of 8-day vs. daily ancillary meteorological variables**

When 8-day vs. daily ancillary meteorological variables were compared, solar radiation ($R^2$ = 0.82, PBIAS = -5%) and relative humidity ($R^2$ = 0.78, PBIAS= 6%) were also found to be in a

similar range (Table AR1). However, we noted that 8-day average daily wind speed, which is highly variable with time and space, was not well representative of daytime conditions ($R^2$ = 0.36). Note that this uncertainty in wind speed could affect instantaneous ET values from the SEBS model, which uses wind speed to determine sensible heat flux ($H$) and estimate latent heat flux ($\lambda E$) as a residual of surface energy balance. In addition, this would also slightly affect 8-day average net radiation that used the FAO-based Penman-Monteith (PM) equation (Allen et al., 1998;ASCE-EWRI, 2005) that is used to upscale instantaneous ET to 8-day cumulative ET. Hence, we think the uncertainties associated with differences in actual meteorological conditions will have more of an effect on SEBS than STIC1.2, which is also supported by results from model evaluation at the instantaneous scale (i.e., MODIS TERRA daytime overpass time).

Table AR1. Comparison of 8-day average daytime meteorological and radiative inputs vs. instantaneous inputs to assess how representative the 8-day average values were of each day within the 8-day period.

| Variable | $R^2$ | RMSE | MAE | PBIAS (%) |
|---|---|---|---|---|
| $T_R$ (K) | 0.92 | 3.53 | 2.74 | 0.1 |
| $T_A$ (K) | 0.900 | 3.04 | 2.34 | 0 |
| $T_R$ - $T_A$ (K) | 0.80 | 3.16 | 2.42 | 3.8 |
| RH (%) | 0.78 | 10 | 8 | 6 |
| Wind speed (m s$^{-1}$) | 0.36 | 1.61 | 1.19 | 2 |
| Incoming shortwave radiation (W m$^{-2}$) | 0.82 | 69 | 49 | -5% |

**Model implementation and validation at instantaneous scale showed similar results**

STIC1.2 and SEBS were implemented using all the available daily MODIS products (MOD11A1, MOD09GA, etc. and instantaneous weather data from NLDAS-2 forcing data) during the study years (Fig. 3, page 35 in the revised manuscript) and evaluated at the instantaneous scale. We noticed similar overestimation and slight underestimation tendencies of SEBS and STIC1.2, respectively (Table AR2 and Fig. AR2). Overall, model performances at the instantaneous scale during dry, normal, and wet years were also consistent with those observed when validation was done at the 8-day scale (see Fig. 4, Page 36 in the revised manuscript); there was a slight underestimation tendency of STIC1.2 and overestimation of SEBS (better under wetter conditions). The additional 11% overestimation of SEBS (17% vs. 28%) could be attributed to uncertainties associated with widely varied wind speed and other meteorological variables within the 8 days of a given 8-day period as well as the slightly overestimated 8-day average net radiation (Fig.AR3, also in the supplementary, Figure S1). The uncertainty associated with wind speed should not affect STIC1.2, which does not use wind speed; therefore, STIC1.2 performances were more consistent compared to SEBS. These results suggest that regardless of whether 8-day or daily MODIS products are used, the key findings of our research would largely remain the same.

Table AR2.Evaluation of instantaneous ET and 8-day cumulative ET (Table 3, Page 31) from STIC1.2 and SEBS against observed ET from thirteen core AmeriFlux sites in the US combining data from one dry, one wet, and one normal year. Note: the 8-day ET estimates are derived from 8-day MODIS products (MOD11A2, MOD09A2, etc.) and 8-day average weather data (both at the satellite overview time and the 8-day average values).

| Scales | Model | $R^2$ | RMSE (mm hr$^{-1}$ or mm 8-day$^{-1}$) | MAE (mm) (mm hr$^{-1}$ or mm 8-day$^{-1}$) | PBIAS (%) |
|---|---|---|---|---|---|
| Instantaneous | STIC1.2 | 0.61 | 0.12 | 0.09 | -5 |
| | SEBS | 0.53 | 0.14 | 0.10 | 17 |
| 8-day | STIC1.2 | 0.66 | 7.5 | 5.4 | -3 |
| | SEBS | 0.53 | 9.8 | 7.3 | 28 |

Fig. AR2. Evaluation of ET estimates from the STIC and SEBS models against flux data at the instantaneous scale using daily MODIS products (MOD11A1) during the dry, normal and wet years (see Fig. 4, Page 33).

**Good correlation between estimated and observed 8-day average available energy ($\phi$)**

The 8-day mean (considering all hours) available energy ($\phi = R_n$-G) that was used to upscale 8-day average daytime instantaneous ET to 8-day total ET (Eq. 17, page 10) was found to be strongly correlated with the observed 8-day mean $\phi$ at the flux towers. While the random noises in instantaneous $\phi$ (Figure AR3), which was also found to be well correlated with observed values ($R^2 = 0.65$), are removed in the 8-day average $\phi$, a small positive bias (i.e. overestimation of 9%) (Fig. AR3, also in the supplementary, Figure S1) was also added. The residual difference in estimated 8-day average $\phi$ and observed 8-day average $\phi$ was positively correlated with the residual difference in estimated and observed 8-day cumulative ET ($r = 0.27$ for SEBS and $r = 0.18$ for STIC1.2, $p$-value $< 0.001$). This positive bias in estimated 8-day mean $\phi$ could have reduced biases from STIC1.2 from -5% to -3% (positive shift) and led to further increases in SEBS biases (i.e. 28%, Table AR2).

[Figure]

Fig. AR3. Scatter plot of estimated vs. observed available energy ($\phi$) at instantaneous (using daily MODIS products) and 8-day averaged conditions (using 8-day products).

**Models performed better in areas that are mostly affected by clouds**

In the humid regions, all three models performed better than in the arid and semi-arid regions with similar accuracies despite high cloud cover (resulting in a fewer number of cloud-free days in each MODIS 8-day cycle). This is partly because vegetation (forests) is mostly energy-limited in this region and because estimated average 8-day $\phi$ from the meteorological inputs from the NLDAS-2 forcing (12.5°) were well correlated with observations ($R^2$=0.91) and with 9% error. In the arid sites, the estimated and observed $\phi$ relationship was also similar ($R^2 = 0.96$ and PBIAS within 6%). In our opinion, the effect of cloud cover is smaller in the arid and semi-arid regions compared to the humid and sub-humid regions and hence most of the differences in model performances could be attributed to the physical differences among the models.

**MOD11A2 and aggregated meteorological are commonly used in ET modeling**

The model implementation and validation scheme used in this study (i.e. use of MODIS aggregated datasets and *n*-day averaged meteorological inputs) have been applied in several other studies (Ichii et al., 2009;Senay et al., 2013;Wu et al., 2010;Xiong et al., 2016). In addition, comparison of daily ET, 8-day average ET or the 8-day with respective flux ET has become a common practice in ET model evaluation studies (Yang et al., 2006;Ichii et al., 2009;Senay et al., 2013;Biggs et al., 2016;Xiong et al., 2015;Ryu et al., 2012) and particularly those that use MODIS 8-day datasets.

**Changes in the revised manuscript**

While we have provided evidence and justifications for using MOD11A2 and aggregated weather information, we do acknowledge that there are uncertainties associated with the use of 8-day average LST and aggregated meteorological variables which should be mentioned in the manuscript. We have now added an extended discussion (Page 18, Line 28-Page 19, Line 14) in the revised manuscript and two tables in the supplementary (Table AR1-AR2).

*2. Page11 (Line 8-17) Again, validation should be carried out at an instantaneous scale but not daily (or 8-day averages), as upscaling can introduce additional uncertainties. In my opinion, upscaling from satellite overpass to longer time scales is another scientific question.*

**Response:** For the most part, please refer to our response to the previous comment. We have added few more points below.

**Should an ET model always be validated at the instantaneous scale only?**

While we agree that the validation of ET at the instantaneous scale could help reduce uncertainties associated with upscaling and overall meteorological representation, we also disagree that the validation should always be conducted this way. ET is a hydrological process and like precipitation and runoff, these processes (and the errors) are perceived better when reported at daily or seasonal scales (e.g. 0.01 mm hr$^{-1}$vs 1 mm/day or 1000 mm year$^{-1}$). For example, ET at daily or seasonal scales is more meaningful for hydrologists who manage water resources (Tang et al., 2015;Cammalleri et al., 2014;Colaizzi et al., 2006) and for comparing with accumulated precipitation (Baldocchi and Ryu, 2011). There are host of studies (for e.g.Fisher et al., 2008;Senay et al., 2013;Velpuri et al., 2013;Jiang and Ryu, 2016;Bunting et al., 2014) that have conducted ET validation at much larger temporal scales (e.g., 8-day, monthly, seasonal, annual) than the instantaneous scale.

The 8-day cumulative or mean ET corresponds with the cycle of MODIS global coverage (Masuoka et al., 1998) and one of the most widely used forms of ET model implementation and validation (Ryu et al., 2012). The 8-day or other multiday MODIS composites are designed to deal with cloud cover to provide a more routine measurement of Earth's surface than the daily MODIS data. The cloud effect on ET estimation or understanding other physical processes is greatly reduced when a composite 8-day LST product is used (Yang et al., 2013;Hu and Brunsell, 2013).

**STIC1.2 has already been validated at an instantaneous scale**

STIC1.2 model has been extensively validated at a half-hourly scale using flux tower data (Mallick et al., 2014;Mallick et al., 2015;Mallick et al., 2016), which is a better evaluation than using remotely sensed data, as any bias associated with the spatial and temporal mismatches between input meteorological and land surface variables are distinctly identified. The strength of the present study is to test the ET mapping potential of STIC1.2 at the regional scale using purely remote sensed data and gridded climate data. Therefore, we performed validation of the STIC1.2 model at the 8-day scale and compared our resulting ET estimates with readily available products such as MOD16 or the widely used SEBS model. In addition, our results are consistent with instantaneous scale validation (Table AR2 and Fig. AR2 in this response and Table 3, page 31 and Fig. 4, page 36 in the revised manuscript).

**Upscaling errors (instantaneous to 8-day ET) are minimized through reliable estimates of 8-day average $\phi$**

We agree that the upscaling of ET from satellite overpass time to longer time scales (e.g. 8-day, as done in this study) is a different scientific question, as there are several uncertainties

associated with it. The approach (Page 10, Eq. 17) we used in this paper is a well-established method (Allen et al., 2007;Colaizzi et al., 2006;Ryu et al., 2012;Shuttleworth et al., 1989;Gentine et al., 2007;Chávez et al., 2008). In addition, the estimated 8-day $\phi$ (the key driving force of ET) was strongly correlated with the 8-day average $\phi$ at flux towers ($R^2$ =0.89, RMSE = 20 W m$^{-2}$, PBIAS = 9%, Fig. AR3, also in the supplementary, Figure S1). Hence, the errors associated with model evaluation at the 8-day scale by upscaling of instantaneous to 8-day cumulative ET should be within 9%. If we had directly used the observed $\phi$ at the towers, PBIAS from SEBS would have reduced to 18% (from 28%) and STIC biases would have been -10% (from -3%) with increased $R^2$ (STIC = 0.69 and SEBS= 0.58) and slightly reduced RMSEs (STIC= 6.8 mm and SEBS = 8.6 mm) from both models. However, it should be noted that the evaporative fraction was derived during the image time obtained using the weather information from gridded data, not the flux tower data itself. Hence there are some uncertainties with the use of meteorological data from multiple sources during instantaneous and multiple scales, as data from the same source its typically used to extrapolate instantaneous to daily or other scales (Allen et al., 2007;Chávez et al., 2008;Allen et al., 1998). Overall, we find that the upscaling errors are within 10% for both models.

*Major Point 3. Page11 (L29-30): Any explanation of this model performance: overestimation in dry year and underestimation in wet years? Additionally, according to your Figure 6, it seems that this "overestimation in dry and underestimation in wet" pattern persists across sites (i.e., spatially). This may suggest some systematic uncertainty of the model. Given this, I do not agree with the statement given in Page 15 (Line 4-12). First, does any previous study support this wet/dry patches around the studied sites? If not, this is just your speculation. Second, the footprint issue could lead to random errors rather than a systematic underestimation. Finally, it is the authors' responsibility to ensure the footprint of a flux site corresponds (or encompasses) the MODIS footprint so that to eliminate data uncertainties and to allow a focused evaluation of the model.*

**Response:** Overestimation of SEBS could be due to uncertainties associated with the $kB^{-1}$ parameter (Chen et al., 2013), as well as the positive biases in 8-day average $\phi$ (as discussed earlier). Underestimation of ET from STIC1.2 could be due to an excessive moisture constraint applied during initialization of soil moisture availability ($M$) using $T_R$ and dew point temperature at source/sink and reference heights. In addition, in the dry years, overestimation errors of $\phi$ ($R^2$ = 0.88, RMSE = 22 W m$^{-2}$, PBIAS = 12%) was slightly more than in the wet year ($R^2$ = 0.91, RMSE = 18 W m$^{-2}$, PBIAS = 9%). At the instantaneous scale, we noticed that STIC1.2 did not overestimate ET during the dry year and the SEBS overestimation was within 24%(PBIAS = 24%), which could be due to uncertain conductance parameterization, as in case of 8-day evaluation. We have discussed potential uncertainties in detail in section 4 (Page 18, Line 28-Page 19, Line 14). The manuscript is about first ever regional scale implementation of the STIC1.2 model using remotely sensed data and hence have focused more on the initial validation as well as the comparison with two other commonly used models.

Please check figures AR4-AR7, where annual ET maps from three global ET products: 1) The Global Land Evaporation Amsterdam Model (GLEAM; 0.25 ° spatial resolution) (Martens et al.,

2017;Miralles et al., 2011); 2) MPI-BGC or Fluxnet: MTE (0.5° spatial resolution) (Jung et al., 2010;Jung et al., 2011)3) SSEBOp (1km spatial resolution; https://earlywarning.usgs.gov/fews/datadownloads) (Senay et al., 2013;Velpuri et al., 2013) are added.  Here, the annual ET map from one of three study years (the year when datasets from the other three models were also available) is shown. While the first two datasets (GLEAM and MPI) are at relatively coarser spatial resolution, most of these maps clearly show a similar spatial pattern of ET as STIC1.2. Hence, the spatial patterns of ET produced by our model seem to be reasonable and not linked with any systematic uncertainty of the model. In addition, scatter plots of estimated vs. observed ET (both instantaneous and 8-day cumulative; Figure AR2, Figure 3 and 4) show points spread uniformly around the 1:1 line.

[Figure]

Figure AR4. Annual ET map derived from STIC1.2, SEBS, MOD16, GLEAM, MTE, and SSEBop for the western (W) bounding box covering US-Ton and US-Me2 flux sites (Fig. 2, Page 31).

[Figure]

Figure AR5. Annual ET map derived from STIC1.2, SEBS, MOD16, GLEAM, MTE, and SSEBop for the mid-western 2 (MW2) bounding box covering US-ARM, US-SRG, US-Wkg, and US-NR1 flux sites (Fig. 2, Page 31).

[Figure]

Figure AR6. Annual ET map derived from STIC1.2, SEBS, MOD16, GLEAM, MTE, and SSEBop for mid-western 1 (MW1) bounding box covering US-Kon, US-KFS, US-ARM, US-Ne1, and US-MMS flux sites (Fig. 2, Page 31).

[Figure]

Figure AR7. Annual ET map derived from STIC1.2, SEBS, MOD16, GLEAM, MTE, and SSEBop for the eastern (E) bounding box covering US-NC1 and US-NC2 flux sites (Fig. 2, Page 31).

As demonstrated and discussed in Mallick et al. (2014), although towers are often installed in relatively homogenous terrain, rarely can this be assumed for heterogeneous landscapes in arid and semi-arid environment. The slope of the regression between the observed and estimated λE of individual biome category was significantly related to the average variance of LST surrounding the tower sites (Mallick et al., 2014). The slope of regression varied systematically with the landscape heterogeneity. Similar results are also shown by Stoy et al. (2013), who also found a systematic relationship between the surface energy balance closure, soil moisture variability, and landscape heterogeneity over 173 FLUXNET tower sites.

Currently, there is no consensus on which MODIS footprint size to use to represent fluxes from a flux site and hence any size or method used is subjected to debate. However, most flux sites (other than the arid and some semi-arid sites) used in this study cover vegetated area that is large enough for a $1 \times 1$ km$^2$ MODIS pixel to be represented as a homogenous pixel, which was also verified in Google Earth Engine. Typically, a pixel-to-footprint match is considered adequate if the vegetation and environmental characteristics within the footprint are good representatives of the surrounding area contained by the MODIS pixels (Yuan et al., 2010). The US-Ton site, however, may not be as homogenous as other sites in terms of vegetation type, as the site is dominated by deciduous blue oaks (*Quercusdouglasii sp.*) and the understory and open grassland are mainly cool-season C3 annual species (Ma et al., 2007). This could lead to dry and wet patches of LST, as briefly discussed on Page 16, Line 2-5. Blue oaks and grasses have distinct phenology and MODIS is not sensitive to understory canopy (Ma et al., 2007;Xiao et al., 2010). In addition, the US-Wkg site was classified as either open shrublands or grasslands in

different years on MCD12Q1 datasets and was not homogenous beyond a 3×3 neighborhood (i.e. one class was surrounded by pixels of another class).

Nonetheless, the sites considered in our study have been widely used in validation of ET as well as other land surface variables and is currently considered the state of the art observation datasets that can be used as benchmark to assess the performance of the remote sensing based models (Running et al., 2004;Yang et al., 2007;Jung et al., 2010) and several common approaches to extract representative MODIS pixel values include single tower pixel (Yuan et al., 2010;Ryu et al., 2011;Jiang and Ryu, 2016), 3×3 mean value with center pixel as the coordinates of flux towers (Sims et al., 2008;Xiao et al., 2008;Yang et al., 2013), and some foot print analysis (Vinukollu et al., 2011). In this study, we extracted ET values from a single tower pixel located closest to the MODIS pixel, but when a 3×3 mean value of estimated ET with the center pixel as the coordinates of flux towers was considered, only negligible changes in model performances was noticed (Table AR3). In addition, the mean values of 8-day cumulative ET from each model were not significantly different when a single pixel or a 3×3 neighborhood was considered ($p$-value > 0.75). Hence, we think that in this study, footprint uncertainties are minimized by selecting homogenous core AmeriFlux sites and this method is consistent with what has been done in previous studies.

**Table AR3.** Evaluation of 8-day cumulative ET (Table 3, Page 31) from STIC1.2 and SEBS against observed ET from thirteen core AmeriFlux sites in the US combining data from one dry, one wet, and one normal year when pixel values of estimated ET were taken as a mean of 3×3 neighborhood with center pixel as the coordinates of flux towers. No significant differences in model performance were noticed when a single tower pixel was considered (Table 3, Page 31).

| Model | $R^2$ | RMSE (mm 8-day$^{-1}$) | MAE (mm) (mm 8-day$^{-1}$) | PBIAS (%) |
|---|---|---|---|---|
| STIC1.2 | 0.66 | 7.3 | 5.3 | -4 |
| SEBS | 0.54 | 9.7 | 7.2 | 27 |
| MOD16 | 0.58 | 9 | 6.3 | -27 |

Minor: 4. Page7(L7): Delete "the" between "both" and "variables"

**Response:** Change has been made (Page 7, Line 14).

Minor: 5. Page9(L20): Please specify the equation for NDVI and/or provide references.

**Response:** We added the following reference for NDVI in section 2.4.2 (Page 9, Line 24).

Tucker, C. J. (1979). Red and photographic infrared linear combinations for monitoring vegetation. Remote sensing of Environment, 8(2), 127-150.

*Minor 6. Page 16 (Discussion on MOD16): It is worthwhile reading Yang et al. (2016, WRR; doi: 10.1002/2014WR015619) for a more physical explanation on the MOD16 uncertainty.*

**Response:** Thanks for referring this paper. Based on this literature, we added an extended discussion on MOD16 uncertainties in the revised manuscript (Page 18, Line 10-24).

[revised manuscript text omitted]

▲  **Core AmeriFlux study sites**
1    US-Me2 (Evergreen Needle Forest)
2    US-Ton (Woody Savanna)
3    US-SRM (Woody Savanna)
4    US-SRG (Semi Desert Grassland)
5    US-Wkg (Semi Desert Grassland)
6    US-NR1 (Evergreen Needle Forest)
7    US-Kon (Grassland)
8    US-KFS (Grassland)
9    US-ARM (Cropland)
10   US-Ne1 (Cropland)
11   US-MMS (Decidious Broadleaf Forest)
12   US-NC1 (Evergreen Needle Forest)
13   US-NC2 (Evergreen Needle Forest)

**30-year mean annual precipitation (mm)**

| | | | |
|---|---|---|---|
| < 400 | 400 - 700 | 700 - 1000 | 1200 - 1400 |
| 1400 - 1700 | 1700 - 2200 | 1000 - 1200 | |
| 2200 - 2800 | 2800 - 3600 | 3600 - 6000 | |

☐  Bounding Box for MODIS processing     ☐  US States

[revised manuscript text omitted]